



# Baltic Earth Assessment Report on the biogeochemistry of the Baltic Sea

Karol Kuliński[1], Gregor Rehder[2], Eero Asmala[3], Alena Bartosova[4], Jacob Carstensen[5], Bo Gustafsson[6], Per O.J. Hall[7], Christoph Humborg[6], Tom Jilbert[8], Klaus Jürgens[9], H. E. Markus Meier[10,4], Bärbel Müller-Karulis[6], Michael Naumann[10], Jørgen E. Olesen[11], Oleg Savchuk[6], Andreas Schramm[12], Caroline P. Slomp[13], Mikhail Sofiev[14], Anna Sobek[15], Beata Szymczycha[1], Emma Undeman[6]

[1] Department of Marine Chemistry and Biochemistry, Institute of Oceanology of the Polish Academy of Sciences, ul. Powstańców Warszawy 55, 81-712 Sopot, Poland

[2] Department of Marine Chemistry, Leibniz Institute for Baltic Sea Research Warnemünde, Rostock, 18119, Germany

[3] Tvärminne Zoological Station, University of Helsinki, J.A. Palménin tie 260, 10900 Hanko, Finland

[4] Research and Development Department, Swedish Meteorological and Hydrological Institute, Norrköping, 601 76, Sweden

[5] Department of Bioscience, Aarhus University, Frederiksborgvej399, DK-4000 Roskilde, Denmark

[6] Baltic Sea Centre, Baltic Nest Institute, Stockholm University, SE-106 91 Stockholm, Sweden

[7] Department of Marine Sciences, University of Gothenburg, Box 461, 405 30 Gothenburg, Sweden

[8] Ecosystems and Environment Research Program, Faculty of Biological and Environmental Sciences, PO Box 65, 00014 University of Helsinki, Finland

[9] Department of Biological Oceanography, Leibniz Institute for Baltic Sea Research Warnemünde, Rostock, 18119, Germany

[10] Department of Physical Oceanography and Instrumentation, Leibniz Institute for Baltic Sea Research Warnemünde, Rostock, 18119, Germany

[11] Department of Agroecology, Aarhus University, Blichers Allé 20, 8830 Tjele, Denmark

[12] Center for Electromicrobiology, Section for Microbiology, Department of Biology, Aarhus University, Ny Munkegade 114, 8000 Aarhus C, Denmark

[13] Department of Earth Sciences, Utrecht University, Princetonlaan 8A, 3584 CB, Utrecht, the Netherlands

[14] Finnish Meteorological Institute, Erik Palmenin Aukio, 1 00560, Helsinki, Finland

[15] Department of Environmental Science, Stockholm University, 10691 Stockholm, Sweden

*Correspondence to*: Karol Kuliński (kroll@iopan.pl)



## Abstract

Location, specific topography and hydrographic setting together with climate change and strong anthropogenic pressure are the main factors shaping the biogeochemical functioning and thus also the ecological status of the Baltic Sea. The recent decades have brought significant changes in the Baltic Sea. First, the rising nutrient loads from land in the second half of the 20th century led to eutrophication and spreading of hypoxic and anoxic areas, for which permanent stratification of the water column and limited ventilation of deep water layers made favourable conditions. Since the 1980s the nutrient loads to the Baltic Sea have been continuously decreasing. This, however, has so far not resulted in significant improvements in oxygen availability in the deep regions, which has revealed a slow response time of the system to the reduction of the land-derived nutrient loads. Responsible for that is the low burial efficiency of phosphorus at anoxic conditions and its remobilization from sediments when conditions change from oxic to anoxic. This results in a stoichiometric excess of phosphorus available for organic matter production, which promotes the growth of $N_2$-fixing cyanobacteria and in turn supports eutrophication.

This assessment reviews the available and published knowledge on the biogeochemical functioning of the Baltic Sea. In its content, the paper covers the aspects related to changes in carbon, nitrogen and phosphorus (C, N and P) external loads, their transformations in the coastal zone, changes in organic matter production (eutrophication) and remineralization (oxygen availability), and the role of sediments in burial and turnover of C, N and P. In addition to that, this paper focuses also on changes in the marine $CO_2$ system, structure and functioning of the microbial community and the role of contaminants for biogeochemical processes. This comprehensive assessment allowed also for identifying knowledge gaps and future research needs in the field of marine biogeochemistry in the Baltic Sea.

## 1. Introduction

The Baltic Sea (Fig. 1) is one of the most thoroughly studied marine ecosystems in the world. The long tradition of marine research is continued here within the framework of statutory activities of research institutes located around the Baltic Sea as well as in many national and international scientific projects. In recent years important contributions to the understanding of the Baltic Sea ecosystem have been made by the projects funded by BONUS - a funding mechanism dedicated for the Baltic Sea region. In addition to that, regular monitoring of physical, chemical, and biological variables is continuing under the auspices of HELCOM (Baltic Marine Environment Protection Commission) by the states surrounding the Baltic Sea. HELCOM has also been evaluating the contemporary state of the Baltic Sea ecosystem for almost 40 years (e.g. Melvasalo et al., 1981; HELCOM, 2018). For more than two decades the knowledge on the Baltic Sea ecosystem has also been systematically assessed, initially by BALTEX and since 2013 by its successor, Baltic Earth. As a result, two comprehensive assessments have been already released: BACC I (Assessment of Climate Change for the Baltic Sea Basin, 2008) and BACC II (2015). The present study is one of the thematic Baltic Earth Assessment Reports (BEARs), which comprise a series of review papers that summarize and assess the available published scientific knowledge on climatic, environmental and human-induced changes in the Baltic Sea region (including its catchment). As such, the series of BEARs constitutes a follow-up of



previous assessments: BACC I (2008) and BACC II (2015). BEARs are constructed around the major scientific topics (so-called Grand Challenges) Baltic Earth deals with. One of those topics, summarized in this study, addresses the biogeochemical functioning of the Baltic Sea.

Figure 1: The Baltic Sea and its catchment.



Marine biogeochemistry is a relatively new discipline, which deals with the transport and transformations of chemical elements
that are crucial for marine ecosystems, in particular: C, N, P, Si (silicon) and $O_2$ (oxygen). As such, it takes into account all
the physical, chemical, biological, and geological processes, which control the cycling of those elements in the marine
environment. On top of that, there are two overarching processes, namely: organic matter production and remineralization,
which link inorganic and organic pools of substances and drive their cycling in the ocean. In addition to the natural functioning
of marine ecosystems with its periodicity, anthropogenic and climatic pressures have the potential to change the
biogeochemical cycles on both global and regional scales. This directly links marine biogeochemistry with climate change and
its consequences, as well as with human-induced nutrient inputs, and causes that issues like deoxygenation, eutrophication or
ocean acidification and their development in a future warmer and high $CO_2$ world are in the centre of interests of the present-
day marine biogeochemistry. The case of the Baltic Sea shows that the biogeochemical functioning of the marine ecosystems
is especially complex in coastal and shelf seas, which are interlinkages between land, open ocean and the atmosphere and
where the anthropogenic drivers are most prominent.

The biogeochemical functioning of the Baltic Sea ecosystem is directly related to its location, specific topography, and
hydrographic setting (Schneider et al., 2015, 2017). The Baltic Sea is a semi-enclosed shelf sea located in northern Europe
(Fig. 1). Its catchment, being about four times larger than the sea surface itself and inhabited by about 85 million people, is
highly diverse. The Scandinavian Peninsula, being drained by many smaller rivers, is not that densely populated and covered
widely with boreal forests. This is opposite to the continental part, which is to a large extent used for agriculture, while
freshwater enters the Baltic Sea from this densely inhabited region through large riverine systems (Elken and Matthäus, 2008;
Snoeijs-Leijonmalm and Andrén, 2017). The Baltic Sea is connected to the North Sea (and thus to the North Atlantic Ocean)
through the narrow and shallow Danish Straits. Sporadic inflows of saline waters from the North Sea and large riverine runoff
imply that the Baltic Sea is one of the largest brackish water bodies on Earth. The salinity (S) gradient on the surface extends
along the southwest to northeast direction from more than S=20 in the Kattegat through about S=7 in the central Baltic Sea
(the so-called Baltic Proper) to almost freshwater conditions (S=2) in the northern part of the Gulf of Bothnia. In addition to
horizontal salinity gradients, the water column in the Baltic Sea is stratified with a permanent halocline located in the Baltic
Proper at 60-80 m water depth. It separates the surface brackish water layer (including the euphotic zone) from more saline
deeper waters, limiting ventilation of the latter (Elken and Matthäus, 2008). This, in combination with eutrophication, leads to
oxygen deficiency or even anoxia and/or hydrogen sulfide ($H_2S$) presence in the bottom waters in vast parts of the Baltic
Proper and large parts of the Gulf of Finland. Consequently, the redox alterations of N and P biogeochemical cycles give rise
to the "vicious circle" (see also 2.3), the positive feedback self-supporting eutrophication in the Baltic Sea (Vahtera et al.,
2007; Schneider et al., 2015, 2017; Savchuk, 2018).

Although the Baltic Sea can be considered a mesotrophic ecosystem, significant biogeochemical changes occurred over the
last decades. Increasing nutrient loads from rivers and atmospheric deposition, reaching their maxima in the 1980s, led to an
increase in ecosystem productivity. According to the conclusions made by Schneider et al. (2015) in the contribution to BACC



II (2015), based on data and scientific literature available at that time, the net ecosystem production increased by a factor of 2.5 since the 1920s/1930s, and winter nutrient concentrations increased ~3-fold. As a consequence, the hypoxic and anoxic areas in the Baltic Sea have also expanded (Carstensen et al. 2014). Since the 1980s the nutrient loads to the Baltic Sea have

been gradually decreasing. However, due to the long residence time of phosphorus in the system, extended by its liberation from sediments under anoxic conditions in the bottom waters, the expected decrease in ecosystem productivity and extent of anoxic/hypoxic areas have not been observed yet. In 2007, the Baltic Sea Action Plan (BSAP) was adopted by the HELCOM member states, in which target loads of N and P have been set. They assumed further reduction by 19% for N and 42% for P loads by 2021 compared to the period 1997-2003 (HELCOM, 2007). However, shortly after the BSAP was published, Eriksson

Hägg et al. (2010) reported that while there are options for further P loads reduction, problems are anticipated for the reduction of N loads. The authors particularly argued that an expected increase in livestock, protein consumption and agriculture development, especially in some eastern European countries, would counteract N load reduction measures.

In 2013, when the previous assessment report (BACC II, 2015) was prepared, the implementation of the BSAP was too short to conclude anything about its effectiveness based on observations, especially taking into account the long response time of

the system. Thus, most of the studies referred to model simulations, assuming different scenarios of nutrient loads and climate change (e.g. Meier et al., 2011, 2012; Neumann, 2010; Neumann et al., 2012; Omstedt et al., 2012). These studies showed that climate change will augment eutrophication effects, although the scale of those changes will depend largely on the nutrient loads scenario. According to those reports, keeping nutrient loads unchanged (business-as-usual scenario) will significantly increase the anoxic and hypoxic areas. On the other hand, the implementation of the BSAP was found to have the potential to

decrease the extent of hypoxic and anoxic areas despite the counteracting influence of climate change. Another highlight from those studies was that rising atmospheric $CO_2$ will lead to a pH decrease in the Baltic Sea surface waters, while the changes in ecosystem productivity will amplify the seasonal variability of pH with only small effects on the mean annual pH value.

Since the work on the last assessment (BACC II, 2015) was carried out, intensive research on the biogeochemical cycling in the Baltic Sea has been conducted, including studies on past, present and future changes. This paper not only summarizes the

results of these recent studies but comprehensively assesses currently available, published knowledge on the biogeochemical functioning of the Baltic Sea, while pointing out knowledge gaps and future research needs. The scope of this study extends from changes occurring in the catchment and their influence on C, N and P loads to the Baltic Sea, through biogeochemical transformations of those elements in the coastal zone and changes in organic matter production (eutrophication) and remineralization (oxygen availability) to burial and turnover of C, N, P in sediments. Additionally, the paper also directly

addresses the changes in the marine $CO_2$ system (including ocean acidification), the role of microorganisms in the biogeochemical functioning of the Baltic Sea and interactions between biogeochemical processes and chemical contaminants. Although the main focus of this assessment report addresses the cycling of C, N, P and O in the Baltic Sea, other substances are also considered, which take part as electron acceptors in redox processes playing important roles in organic matter remineralization under hypoxic and anoxic conditions.



This paper, apart from being a comprehensive assessment of the biogeochemical functioning of the Baltic Sea, is also a timely contribution and an important baseline for the discussion on future actions towards reaching a good environmental status of the Baltic Sea. As of 2021, when this assessment is concluded, the first time frame for the BSAP is coming to an end, while the discussion is continuing on the selection of new measures and actions for an updated BSAP.

## 2. The current state of knowledge

### 2.1 Changes in the catchment and the inputs to the Baltic Sea

### 2.1.1 Changes in external drivers

The biogeochemistry of the Baltic Sea is largely fuelled by external loads of nutrients. Changes in nutrient loads are driven by human activities in the catchment and modified by climatic conditions (primarily temperature and precipitation). There have been dramatic changes in these factors and their drivers over the past decades, and these are expected to change further into

the future as affected by the socio-economic and climatic change (Pihlainen et al., 2020). For the factors affecting N and P loads to the Baltic Sea basin, modelling shows that changes in societal factors have the potential to outweigh the effects of changes in the climate (Bartosova et al., 2019; Pihlainen et al., 2020).

Anthropogenic activities drive much of the nutrient inputs to the Baltic Sea, either through stream discharges or atmospheric deposition. Stream discharges of N and P are mainly driven by land-use activities and wastewater discharges from urban areas.

These wastewater discharges depend on the population, their diets and the efficiency of wastewater treatments (Van Puijenbroek et al., 2015). The land-use activities that drive N and P loads are mainly associated with agricultural activities (Reusch et al., 2018), and these activities vary greatly across the Baltic Sea basin with much greater N inputs and losses in the south compared to the northern part of the drainage basin (Andersen et al., 2016). Therefore, regulating these activities may have the greatest impact on the N loads to the Baltic Sea (Olesen et al., 2019), although there may also be effects of changes

in land use in other parts of the basin (Bartosova et al., 2019). The Baltic Sea is also affected by N (and sulfur (S) deposition from the atmosphere, and these originate from many different sources, including ammonia from primarily agricultural activities in the region and beyond, and nitrogen oxide (NOx) emissions from on-land combustion and shipping in the Baltic Sea (Karl et al., 2019).

An updated architecture for future scenarios related to climate change was developed in support of the Intergovernmental Panel

on Climate Change (IPCC) process (Ebi et al., 2014). This approach distinguishes scenarios of climate change from those of socioeconomic developments. The climate change scenarios were simplified into four Representative Concentration Pathways (RCP), representing typical developments in radiative forcing during the 21st century and beyond. There are four core RCPs (RCP2.6, RCP4.5, RCP6.5 and RCP8.5) that represent key pathways for global warming, where the numbers refer to the additional radiative forcing in 2100 in $W\ m^{-2}$. These RCPs have been applied with global climate models (GCM) to project

changes in future climate, and results of the Coupled Model Intercomparison Project (CMIP5) study that contain projections of many different GCMs have been widely used in impact studies (Knutti and Sedlacek, 2012). These GCM projections need





to be downscaled for use in impact models in particular to resolve regional biases in the climate models. The results generally show a greater variation among climate models than between RCPs for projections until the mid-21st century, but greater variation among RCPs towards the end of the century.

Projections of climate change in the Baltic Sea region by 2050 compared to the late 20th century show temperature increases of 1 to 5°C with an average of 2.5°C and annual precipitation increases of 0 to 20% with an average of 10% (Bartosova et al., 2019). The projected changes show greater temperature increases in the northern parts of the Baltic Sea Basin than in southern parts, in particular during winter (EEA, 2017). There are no clear spatial patterns of changes in projected precipitation across the region.

The revised IPCC scenario approach also includes Shared Socioeconomic Pathways (SSPs) that reflect how different policies within climate change mitigation and adaptation interact with other sustainable development policies and pathways (Ebi et al., 2014). There are five core SSPs (SSP1-SSP5) that span a matrix of challenges for adapting to climate change and challenges for mitigating climate change. Zandersen et al. (2019) adapted this concept to the environmental problems for the Baltic Sea Basin so that the SSPs span a matrix of challenges for adapting respectively mitigating Baltic Sea environmental problems.

This resulted in narratives that allow quantification of changes in the drivers of emissions of N and P to the Baltic Sea through modelling (Bartosova et al., 2019; Pihalainen et al., 2020). Of the SSPs, the most contrasting in terms of nutrient loads to the Baltic Sea are SSP1 (sustainable development) and SSP5 (fossil-fueled development). The policies in SSP1 focus greatly on mitigating environmental issues leading to reductions in agricultural land use and use of technologies that lower all emissions, whereas agricultural land use expands in SSP5 with some adoption of more efficient technologies. Results of scenario analyses

with this approach show that the targets of the Baltic Sea Action Plan can only be achieved following the trajectories of SSP1 (Pihlainen et al., 2020). Other scenario analyses have focused on the impact of existing policies, and this has for instance shown that the European Union (EU) Agricultural Policy does not contribute to lowering nutrient emissions from agricultural activities (Jansson et al., 2019).

### 2.1.2 Hydrological regime

A large amount of matter enters the Baltic Sea with riverine flows, playing a significant role in the biogeochemical conditions of the marine ecosystem. Changes in runoff can thus significantly impact inflows of nutrients and organic matter into the Baltic Sea. Several studies indicate an overall increase in mean discharge (Bartosova et al., 2019; Donnelly et al., 2014; Hesse et al., 2015) to the Baltic Sea projected for future climate scenarios. Although the studies agreed on the significance and the direction of the change, the magnitude of the increase varied with the selected climate model (GCM/RCM), hydrological simulation

model, or bias adjustment method.

Both the projected magnitude of the increase and the confidence in the change vary spatially across the drainage basin. The largest relative increase was projected for the northern part of the Baltic Sea Drainage Basin (BSDB). The projected change in the southern part was more uncertain and likely of lower magnitude. The projected increases in freshwater inflow to the



Baltic Sea can affect the surface sea salinity with potentially negative effects on biotic communities in the Baltic Sea
(Kniebusch et al., 2019).

Most studies again agree on a decrease in flows associated with snowmelt and increase of winter flows. The change of summer
flows is then uncertain, varying from a decrease (Donnelly et al 2014) to a smaller increase (Hesse et al., 2015). The difference
may be associated with a larger uncertainty in future evapotranspiration as calculated by the different models with different
underlying assumptions.

Aside from the changes in magnitude, spatial, and temporal distribution of discharge, other characteristics are also affected.
Klavins et al. (2009) pointed out e.g. the ongoing reduction in ice cover and time shift in the ice break-up to earlier periods in
all rivers in the Baltic Region except the most southern and most northern rivers. The ice cover duration was declining by 2.8
to 6.3 days per decade during the past 30 years, having been strongly influenced by the North Atlantic Oscillation index.

### 2.1.3 Nutrient legacy pools

Over the last century, developments in agriculture have added significant amounts of N and P to agricultural land in the form
of fertilizer and manure. This considerably exceeds the amounts removed by the harvest, and the current nutrient use efficiency
is slightly above 50% for both N and P integrated over the whole agricultural land of the Baltic Sea catchment (McCrackin et
al., 2018a). Therefore, large amounts of N and P have accumulated in agricultural soils. This led to hotspots of agricultural
nutrient losses especially in the southern part of the catchment where N root zone leakage is significant (Andersen et al., 2016)
and constitutes the major pathway (> 50%) of N emissions into the Baltic Sea. However, whereas catchment-wide dynamic
modelling of agricultural losses does exist for N, for P mainly direct observations (Pengerud et al., 2015) and empirical
approaches have been developed, addressing hot spots and risk areas (Djodjic and Markensten, 2019) through empirical
relationships. Further, the long-term legacy of these inputs may have greater impacts through their effects on the land-based
nutrient pools than the accumulated legacy nutrient pools in the Baltic Sea (McCrackin et al., 2018b). In this study, the authors
developed a three-parameter box model approach and estimated that more than 44 Tg P have accumulated in agricultural soils
in the entire catchment over the last century of which 17 and 27 Tg P have accumulated in a mobile and stable storage pool,
respectively. Presently, losses from this mobile pool contribute nearly half of the riverine P loads. The model suggests an
overall residence time of P in the mobile pool of some 30 years and that riverine loads could decrease by as much as 10% by
2021 and 15% as a result of recent measures by 2050, even if there were no further reductions in P inputs.

### 220 2.1.4 Weathering and trends in alkalinity and TOC

Carbon is entering the Baltic Sea mainly as total inorganic (TIC) and organic (TOC) C. According to Kuliński and Pempkowiak
(2011), riverine C input amounts to 10.9 Tg C yr$^{-1}$ of which 37.5 % has been estimated as TOC. Most terrestrial derived TOC
is respired in the Baltic Sea (Fransner et al., 2016, 2019) and therefore exerts positive feedback to atmospheric $CO_2$
concentrations. In contrast, dissolved inorganic carbon (DIC) and alkalinity production via silicate and carbonate weathering
constitute a $CO_2$ sink, because atmospheric $CO_2$ is consumed during the various weathering reactions between minerals and



carbonic acid supplied by precipitation that form DIC and alkalinity (Berner, 1991). In general, it is assumed that both TOC mobilization and weathering increase with temperature rise due to increased biomass turnover and faster chemical reaction rates. In fact, TOC concentrations have increased from 12 to 15.1 mg l$^{-1}$ corresponding to an increased riverine input of 0.28 Tg C yr$^{-1}$ between 1993 and 2017 in the northern boreal watersheds of the Baltic Sea catchment (Asmala et al., 2019).

Similarly, weathering fluxes (as expressed as total dissolved solids, TDS, which include DIC and alkalinity) have increased by 10-20% over the last 40 years (Sun et al., 2017). The increase in TOC can be related to increasing trends in water discharge and pH whereas weathering fluxes could be related to precipitation only. However, $CO_2$ consumption rates by weathering are estimated with about 3 g C m$^{-2}$ yr$^{-1}$, which correspond to 3-30% of the net ecosystem carbon exchange in the boreal part of the Baltic Sea catchment (Sun et al., 2017). Overall, river chemistry data are more available and reliable for the boreal part of the

Baltic catchment compared to the southern river catchments and therefore overall carbon trends in river loads and related climate feedback processes are still uncertain. This is also the reason why the potential dampening effect of increased alkalinity loads on Baltic Sea acidification still needs to be better quantified (Gustafsson et al., 2019, see also Chapter 2.6).

### 2.1.5 Nutrient loads under changing climate

Changes in nutrient loads from the Baltic Sea Drainage Basin to the Baltic Sea due to changing climate or anthropogenic

influences were studied using a number of different approaches in recent years, including modelling, trend analyses, and functional relationships.

Bartosova et al. (2019) projected an increase in nitrogen and phosphorus loads to the Baltic Sea from a mini-ensemble of climate projections using the hydrological model E-HYPE. Hesse et al. (2015) reported decreasing trends for nitrate, ammonia, and phosphate loads on average to the Vistula lagoon using the hydrological model SWIM. However, a wide range of impact

projections was reported for individual ensemble members. Hägg et al. (2014) also projected an increase in nutrient loads using a split model approach to project changes in total nitrogen (TN) and total phosphorus (TP) loads into Baltic Sea sub-basins (BSB). They combined discharges modelled with CSIM for a climate projection ensemble with a statistical approach for nutrients using population and projected population changes. Huttunen et al. (2015) in a study of Finnish basins draining to the BSB used a national nutrient load model (VEMALA) with a mini-ensemble for climate impacts. Even here the results

suggest an increase of total nitrogen and total phosphorus loads.

Øygarden et al. (2014) used measurements in several small agricultural catchments to establish functional relationships between precipitation, runoff, and N losses from agricultural land, and qualitatively related their findings to projected precipitation change pattern across the BSDB under climate change scenarios, as well as mitigation measures to counter the climate-driven effects. While such data-driven approaches avoid uncertainties related to impact model chains, it is limited

spatially by the availability of measurements and by assuming these relationships will stay unaffected under changing conditions.



Several studies compared the relative importance of the changing climate and changing socioeconomic conditions or adaptations scenarios, agreeing that the socioeconomic factors play a significant role (Bartosova et al., 2019; Hägg et al., 2014; Huttunen et al., 2015; Pihlainen et al., 2020) and may in some cases outweigh or even reverse the climate impacts (Bartosova et al., 2019; Hägg, 2014; Pihlainen et al., 2020). Impacts of socioeconomic adaptation choices on nutrient loads to the BSB in the same magnitude range as climate impacts indicate the importance and potential of effective mitigation strategies in the region.

### 2.1.6 Atmospheric pathway of the nutrient input to the Baltic Sea

The driving mechanisms of the atmospheric input are the dry deposition and scavenging with precipitation of a variety of gaseous and particulate nutrient species. Among the considered nutrients, the most significant input from the atmosphere is for the nitrogen compounds (~220 kt or >20% of the total input), which is related to the strong emissions of both oxidized and reduced nitrogen into the atmosphere over Europe. Phosphorus input via the atmospheric pathway is uncertain but estimated to be roughly at the level of 2 kt or 5% of the total load. (Svendsen et al., 2015).

During the last two decades, the nitrogen supply via the atmospheric pathway has been noticeably reduced, owing to overall reductions of European emissions (Gauss et al., 2017, 2021). The largest reductions have been for oxidized nitrogen – since 1995 it has lowered by over 40%, whereas reduced nitrogen (ammonium and ammonia) has declined by ~10%. The total nitrogen deposition on the Baltic Sea surface has therefore decreased by about 30%.

The atmospheric nitrogen supply to the Baltic Sea has strong geographical variation and a south-to-north decreasing tendency because the majority of the nitrogen sources are located south of the sea. There are also several mechanisms controlling the deposition patterns. Oxidized nitrogen comes into the atmosphere mainly in the form of NO and $NO_2$, which are very poorly soluble gases. The formation of secondary pollutants, such as nitric acid and nitrate aerosols requires a certain time and favourable environmental conditions (for ammonium nitrate formation – also the presence of ammonia in the air). As a result, near-source deposition is quite limited and the long-range transport of the pollutants plays a key role in the final deposition pattern (Hongisto, 2011). The episodic character of the transport and deposition events leads to the high load variability, even at the annual level (Bartnicki et al., 2011). Therefore, a "normalized" deposition was introduced using the EMEP source-receptor matrices for reducing the meteorology-induced inter-annual variability – see e.g. Annex D in Bartnicki et al. (2017). The above-mentioned reduction trends have been estimated using this noise-reduction approach. Interestingly, over 50% of these reductions were achieved in recent years (Bartnicki et al., 2011; Gauss et al., 2017, 2021), despite the lowering pace of emission reduction. However, Gauss et al. (2017, 2021) show that the driving factors for the faster decrease in recent years were of meteorological origin – and the normalized deposition exhibits a practically constant decreasing trend.

The nitrogen deposition pattern, apart from the south-north gradient, also reflects regional differences and different trends in the Western and Eastern parts of Europe. In particular, the transformation of the economy and environmental practices in



Eastern Europe resulted in the growth of NOx and NHx emission in several countries during the 1990s with subsequent reduction in the 2000s (Bartnicki et al., 2018).

Shipping in the Baltic Sea is a significant source to nitrogen deposition: in some regions and seasons, ships can contribute more than 50 % to the total load (Stipa et al., 2008). On average, about 17% of the total NOx load originate from ship exhausts with an increasing tendency (Jonson et al., 2015). If no measures are taken, ship emissions are estimated to reach 25% by 2030. With the currently planned Nitrogen Emission Control Area (NECA) in the Baltic and North Seas, growth in shipborne N-emissions is still expected but confined within < 20% (Jonson et al., 2015; Karl et al., 2019).

**2.1.7 Nutrient inputs from the catchments**

Humans have for a very long time impacted nutrient inputs to the Baltic Sea through agriculture and deforestation. Potentially already during the Medieval era, anthropogenic nutrient inputs caused significant eutrophication effects in the Baltic Sea (Zillen et al., 2008). With industrialization, human and industrial waste sources also started to influence the nutrient inputs to the sea.

Several studies have investigated nutrient inputs around 1900. These studies are primarily constrained to certain countries and driven by the Water Framework Directive (WFD) requirements to evaluate the state of the environment in relation to a reference state (e.g. Hirt et al., 2013; Rosenstrand Poulsen et al., 2017), but there are also a few studies on a pan-Baltic scale (e.g. Savchuk et al., 2008; Schernewski and Neumann, 2005).

The first comprehensive time series of riverine nutrient inputs to the Baltic Sea was constructed by Stålnacke et al. (1999).
This study compiled data to a complete time series covering the period 1970-1993, although data from several major rivers were lacking for the first decade, e.g. the major Polish rivers, and had to be reconstructed. This data set was later extended to 2006 by Savchuk et al. (2012a) using among other sources data from HELCOM pollution load compilations (HELCOM PLC). Savchuk et al. (2012a) also added estimates of coastal direct point source loads. Savchuk et al. (2012b) reconstructed nutrient loads for 1850-1970. For riverine loads, they used established loads at 1900 as one fix point (Savchuk et al., 2008) and a two-
step linear increase with a breakpoint at 1950. Coastal direct point source changes were estimated using population changes in major cities as a driver, but also here linear changes between a few specific years were assumed. But only relatively recently, in 1995, HELCOM PLC started to annually publish comprehensive, quality-controlled time series of both riverine and coastal direct point source loads (HELCOM, 2019).

In Figure 2, the data sets from Savchuk et al., (2012a, 2012b) are shown together with the data from HELCOM PLC. It is
evident that we know very little about the temporal development of nutrient inputs before 1970, but there is clear evidence that much of the riverine load increase happened after 1950 as was assumed in the reconstruction. Differences between the loads from Savchuk et al. (2012a) and the HELCOM PLC time-series are relatively small in the overlapping period and some of the



difference in concentrations might be due to differences in the river runoff data used. In the Gulf of Bothnia, riverine nitrogen

loads (Fig 2-A) have been relatively stable during the monitored period (1970-2017), but for phosphorus (Fig 2-B) it seems that loads dropped to a lower level after 2000. Still, average flow-weighted concentrations in the Gulf of Bothnia rivers are about 25% (nitrogen) and 100% (phosphorus) higher than pre-industrial concentrations. In the Baltic Proper (including the Gulfs of Finland and Riga), riverine nitrogen loads (Fig 2-C) are weakly declining after 2000 from a long period (1970-2000) of relatively constant loads. However, for the Baltic Proper, we have to bear in mind the lack of data from major rivers already

before 1980. The riverine loads of phosphorus (Fig 2-D) are much lower today than during the maximum period (ca. 1975-2000) and concentrations dropped by about 40%. Here, the two data sets differ quite a bit in the period 1995-1999. In the Baltic Entrance area, both riverine nitrogen and phosphorus loads decreased (Fig 2-E and 2-F). Flow-weighted concentrations are at times misleading in this basin since a large proportion of the water flow is supplied by the Göta River that drains an area to the northeast of the remaining catchment of the Baltic Entrance area with low agricultural land use.

The temporal development of the nitrogen and phosphorus coastal point source loads (Figure 2-G and 2-H) have had a significant influence on the temporal development of the total loads, in particular for phosphorus.

In summary, riverine nutrient loads to the Baltic Sea generally decreased, in particular since about 2000. Today, coastal point sources contribute relatively small amounts of nutrients compared to the rivers, but they have been very large contributors to eutrophication in the past. Both in the Baltic Proper and the Entrance area, it is estimated that today's coastal point sources

contribute fewer nutrients than they did in 1900.





*Figure 2: Nitrogen (panels A-C and G) and phosphorus (panels D-F and H) nutrient load time-series. Total and inorganic riverine loads of nitrogen and phosphorus, respectively, are shown as bars and flow-weighted concentrations of total nutrients*

*calculated from the annual load and flow are shown as lines. The riverine loads to the Gulf of Bothnia are shown in panels A and D, Baltic Proper plus Gulfs of Riga and Finland in panels B and E, and the Baltic Entrance Area (Kattegat, the Sound and Belt Sea) in panels C and F. In panels G and H are the loads from coastal direct point sources drawn for total nitrogen and total phosphorus, respectively. Loads are assessed from two sources (Savchuk et al., 2012a and 2012b) and the HELCOM PLC (HELCOM, 2019 for total nutrients and directly from the HELCOM PLC-water database for the inorganic nutrients).*



## 2.2 Transformations of C, N, P in the coastal zone

### 2.2.1 Functioning of the coastal filter

The coastal zone is the link between land and the open sea with a diverse range of habitats. These complex coastal ecosystems are important for the cycling of elements on a global, regional and local scale. Inputs of organic matter (OM) and nutrients (N and P) from land are bypassed, transformed, retained and removed on their passage to the open sea through the coastal zone, removal being the only process permanently directing nutrients and OM outside the aquatic ecosystems (Asmala et al., 2017). The efficiency of the coastal filter is highly variable, depending on its hydromorphology and biological configuration (Carstensen et al., 2020; McGlathery et al., 2007). Coastal ecosystems harbour diverse biological communities and the strong benthic-pelagic coupling in shallow coastal ecosystems plays an important role in the functioning of the coastal filter. However, reduced functional biodiversity from nutrient over-enrichment and hypoxia, particularly the loss of deep-burrowing benthic macrofauna, hampers the coastal filter significantly (Carstensen et al., 2020; Conley et al., 2009; Norkko et al., 2012).

The most important processes for permanent nutrient removal in the coastal zone are denitrification and phosphorus burial (Fig. 3), and both these processes are strongly modulated by oxygen conditions. The Baltic Sea coastal zone removes approximately 16% of total N and 53% of total P inputs from land (Asmala et al., 2017), whereas less is known about the removal of OM in the coastal filter.

Denitrification removes N by reducing nitrate to dinitrogen that escapes to the atmosphere, and this process is regulated by the availability of nitrate and labile organic carbon as well as temperature and oxygen concentrations (Piña-Ochoa and Álvarez-Cobelas, 2006). Denitrification typically occurs at the oxic-anoxic boundary in sediments, with nitrate supplied through nitrification of ammonia to nitrate or diffusive transport from the nitrate-rich overlying water. However, denitrification can be limited by nitrate availability, particularly with reduced nitrification under hypoxic conditions. Nitrification of ammonia to nitrate is inhibited by low oxygen in the sediments and loss of bio-irrigating macrofauna reduces transport of nitrate across the sediment-water interface. Denitrification rates vary by two orders of magnitude across Baltic coastal ecosystems (Fig. 3; Asmala et al., 2017).

Phosphorus burial occurs in fine-grained sediments in three general forms: 1) organic P, 2) iron (Fe)-oxide bound P, and 3) authigenic minerals. The latter form is generally thought to be dominated by apatite (Ca-P form), but in low salinity areas such as the Baltic Sea vivianite (Fe(II)-P form) can constitute a major sink for P (Slomp, 2011). However, these burial forms have different stability and phosphate is liberated from Fe-oxides under anoxic conditions. On the other hand, increasing Fe inputs from land and decreasing salinity promote vivianite formation. P burial rates are tightly coupled to sedimentation rates and vary by an order of magnitude across coastal ecosystems (Fig. 3; Asmala et al., 2017).

Organic matter is removed through heterotrophic consumption, photochemical degradation, flocculation and burial. More than half of the organic carbon consumed by bacteria is respired as $CO_2$ (Asmala et al., 2013), suggesting that this pathway is an important process for OM removal. Bacterial consumption is further stimulated by photodegradation, transforming biologically





recalcitrant OM into more labile constituents (Moran et al., 2000). OM inputs from land are characterized by large molecules with humic properties, which are susceptible to flocculation when freshwater mixes with saltwater (Asmala et al., 2014). Flocculation contributes to OM burial, together with the autochthonous production of particulate OM. The biogeochemical

processing of organic carbon is complex and variable across coastal ecosystems, with profound changes in the quantity and quality of OM inputs from land to sea.

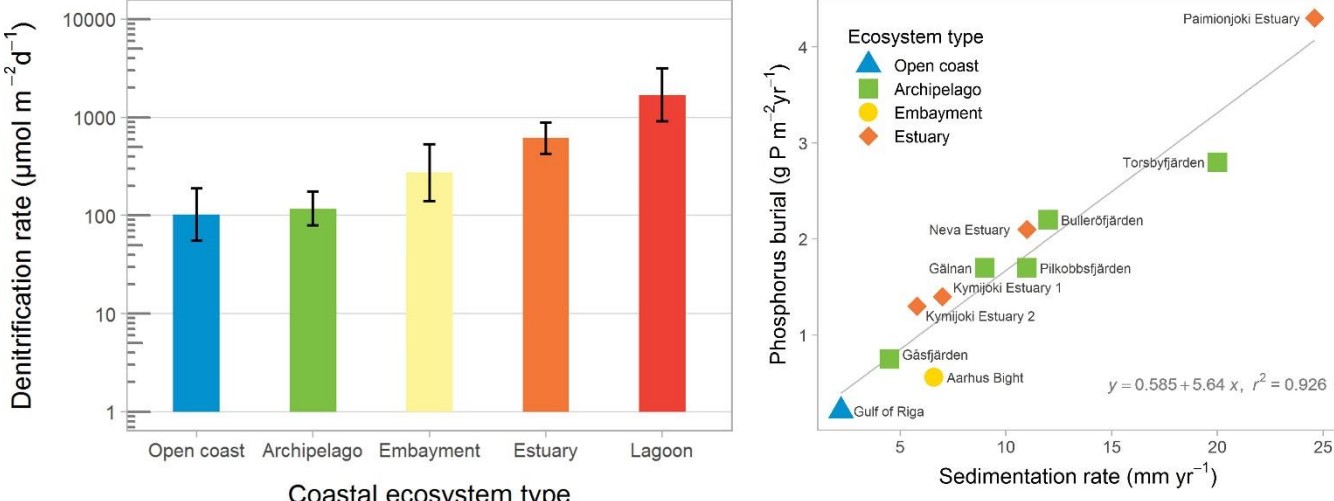

*Figure 3: Mean denitrification rates in coastal ecosystem types in the Baltic Sea (left panel). Error bars show the 95% confidence intervals for the mean estimates. Relationship between sedimentation accumulation rate and phosphorus burial in*

*11 study sites across the Baltic Sea (right panel). Redrawn from Asmala et al. (2017)*

Coastal ecosystems of the Baltic Sea are hydromorphologically diverse, ranging from lagoons, archipelagos, river-dominated estuaries, and embayments to open coastal stretches, but they also vary broadly in their physical-chemical conditions (Carstensen et al., 2020). Particularly, hypoxia is widespread in the coastal zone due to stratification and low ventilation of bottom waters combined with high inputs of nutrients and organic matter (Conley et al., 2011). The increase in coastal hypoxia

over the last century has therefore reduced the "filter function" of the coastal zone, enhancing nutrient enrichment of the open Baltic Sea. Despite that nutrient inputs to the Baltic Sea have been curbed, hypoxia remains prevalent in many coastal ecosystems and even continues to increase in some because of warming (Conley et al., 2011). Increasing temperature promotes hypoxia by reducing the solubility of oxygen in surface water and enhancing respiration. However, warming also prolongs the productive period, when nutrient and organic matter inputs from land are intercepted and processed by coastal organisms,

thereby enhancing the filter function. The diversity of coastal ecosystems and watersheds around the Baltic Sea has resulted in different trends in the past regarding biogeochemical functioning, and will likely experience different trajectories in the future.





### 2.2.2 Coastal filter processes across the different coastal ecosystems

**2.2.2.1 Archipelagos**

Archipelagos are found mainly along the Swedish east coast and in southern Finland. Nutrient inputs are generally low from the boreal watersheds, dominated by forests, draining into the archipelagos. Except for archipelagos receiving large inputs of nutrients and OM from point sources (e.g. urban areas and pulp and paper industry), these coastal ecosystems are not severely affected by eutrophication. However, due to the complex bathymetry and restricted ventilation of bottom waters, some

archipelagos are naturally prone to locally low oxygen conditions. Increasing inputs of nutrients and OM have disrupted the subtle balance between oxygen supply and consumption, causing hypoxia to develop in many locations (Conley et al., 2011). Archipelagos also have sheltered sedimentation basins due to the complex bathymetry, which promotes particle trapping and subsequently elevated sedimentation. Consequently, archipelagos are important for the burial of P and particulate organic matter (POM) (Jilbert et al., 2018). However, due to the low terrestrial inputs of nitrate and labile organic matter, denitrification

rates are low and primarily fuelled by autochthonous carbon from the spring bloom (Hellemann et al., 2017). In contrast, archipelagos with longer residence times that receive large inputs of N and OM constitute efficient filters for C, N and P. For example, Almroth-Rossell et al. (2016) estimated that 72% and 65% of N and P inputs from the land were removed in the Stockholm Archipelago.

**2.2.2.2 Estuaries**

Estuaries are found mainly in the south-western Baltic Sea, where population density in the catchments is higher and land use dominated by agriculture. Consequently, most of these estuaries have suffered from eutrophication for a long time after receiving substantial inputs of nutrients and organic matter, although efforts to reduce these inputs have been successful over the past 2-3 decades (Riemann et al., 2016). Estuaries are typically stratified in the deeper parts, but the renewal of bottom water is dynamic, driven by changes in freshwater inputs and winds. Many estuaries experience seasonal hypoxia in summer

and early autumn when oxygen consumption outpaces oxygen supply. Despite nutrient reductions, oxygen conditions have not improved, as these efforts have been counteracted by increasing temperatures (Carstensen and Conley, 2019; Conley et al., 2007). Estuaries harbour rich biological communities, stimulating removal processes and thereby increasing the coastal filter efficiency (Carstensen et al., 2020). Estuaries with longer residence times have higher sedimentation rates, enhancing the burial of phosphorus and OM as well as denitrification (Seitzinger et al., 2006). Due to the high productivity and degradation/removal

of OM inputs from land, the characteristic of the OM pool rapidly changes from terrestrial to marine (Asmala et al., 2018).

**2.2.2.3 Lagoons**

Coastal erosion and sediment transport have formed lagoons along the southern coastline. These lagoons receive nutrients and OM through rivers of variable sizes, draining a watershed dominated by agriculture. Despite significant freshwater input, residence times can be long in these lagoons due to the restricted connection with the open Baltic Sea. The lagoons are mostly

shallow with low burial rates, but the high inputs of nitrate and labile OM result in high rates of denitrification (Asmala et al.,





2017). Remineralisation and denitrification rates are further stimulated by abundant benthic microalgae and chironomids (Benelli et al., 2018). However, the relatively high N removal in lagoons is counteracted by nitrogen fixation during summers favourable to cyanobacteria blooms (Vybernaite-Lubiene et al., 2017). Recently, it has also been shown that high productivity in shallow lagoons can drive the already carbonate-rich waters from the southern drainage basins to extreme values of calcium carbonate ($CaCO_3$) supersaturation and even trigger $CaCO_3$ mineral precipitation (Stokowski et al., 2020).

#### 2.2.2.4 Large river plumes

A number of large rivers discharge directly into the open Baltic Sea, with physical mixing of river and seawater dominating conditions in the plume. In the southern Baltic Sea, the large rivers deliver high inorganic nutrient inputs that sustain high productivity and sedimentation in the plume. The sedimenting particles are partly buried, partly remineralised and partly shuttled towards the deeper Baltic Sea (Nilsson et al. 2021). In contrast, the large rivers in the northern Baltic Sea have low nutrient but high dissolved organic matter (DOM) concentrations, and the considerable OM load partially flocculates in the plume and settles onto the seafloor. Overall, the large river plumes are mostly conduits of nutrients and OM and the coastal filter efficiency is low.

#### 2.2.2.5 Open coasts

Large stretches of the south-eastern Baltic Sea coastline are dominated by open, sandy shores, where local inputs from land are small and exchange with the open sea is significant. Due to the strong and variable hydrodynamics sedimentation and resuspension occur intermittently, whereas permanent burial is small. Denitrification rates are also small relative to estuaries and lagoon (Sundbäck et al., 2006). Consequently, open coasts are primarily conveyors of C, N and P, i.e. having a low coastal filter effect. The role of open coasts in the biogeochemical processing of nutrients and OM will most likely not be altered substantially with the expected climate changes.

#### 2.2.3 Efficiency of the coastal filter in the future

The Baltic Sea region is expected to warm even further and precipitation is expected to increase in the future. This will have large consequences for the biogeochemical processing of nutrients and organic matter in the coastal zone. Hypoxia is an important factor governing the efficiency of the coastal filter, and hypoxia will increase in a warmer climate due to reduced oxygen solubility and enhanced respiration, and this will consequently reduce coupled nitrification-denitrification and iron-bound P burial, which are two important processes of the coastal filter. The negative effects of developing hypoxia on the coastal filter will be most pronounced in archipelagos and estuaries.

Higher temperatures enhance remineralization processes and thereby ammonia production in sediments, which could stimulate coupled nitrification-denitrification, as long as nitrification is not inhibited by low oxygen. However, higher remineralisation under oxic conditions also reduces the availability of labile organic matter for denitrification. Most likely, denitrification will be stimulated in lagoons and estuaries where the availability of labile OM is high, whereas denitrification will be reduced in



archipelagos that receive low inputs of nutrients and OM. Thus, the resulting outcome of temperature increase on denitrification varies among coastal ecosystems and the seasonality in nutrient and organic matter inputs and processing (Bartl et al., 2019).

The productive period is prolonged with increasing temperatures, already signified by the occurrence of earlier spring and later
autumn phytoplankton blooms (Wasmund et al., 2019). This implies that more inorganic nutrients are intercepted, thereby enhancing the coastal filter efficiency. This effect is particularly pronounced in estuaries, lagoons and large river plumes, where most of the nutrients are discharged during winter and spring. Warming also stimulates nitrogen-fixing cyanobacteria, particularly in brackish lagoons, which counters the coastal filter effect by adding nitrogen to the coastal ecosystem.

Increasing precipitation may also involve increasing freshwater discharge, which reduces estuarine residence time and
increases stratification. The effect on oxygen supply below the pycnocline is highly site-specific and thus, oxygen conditions may improve, deteriorate or remain unaltered with associated consequences for removal of nutrients and OM. Moreover, increasing freshwater discharge will extend the large river plume zone further and enhance the direct transport of nutrients and OM into the open Baltic Sea. Enhanced export of Fe from land and decreasing salinity from freshening in the coastal zone can promote the burial of P in more stable forms, such as vivianite, in low-salinity archipelagos in the Gulf of Bothnia (Lenstra et
al., 2018).

Inputs of organic matter from land are also expected to increase in the future in response to increasing precipitation and warming. These changes are expected to be most pronounced in boreal watersheds. It is therefore possible that this expected increase can alleviate carbon limitation of denitrification, which occurs in northern coastal ecosystems (Hellemann et al., 2017). However, the lability of this organic carbon source is considered low and may not significantly enhance denitrification.

In summary, there is no uniform response of the coastal filter to climate change for either C, N or P removal. The complexity of the biogeochemical processes as well as how these are modulated by coastal organisms mean that the future outcome of the coastal filter in a given ecosystem depends on several oppositely directed processes. Consequently, predicted changes in the coastal filter can only be resolved through coupled system-specific hydrodynamic-biogeochemical-biological coupled models with an improved parameterization of the key processes.

**2.3 Changes in organic matter production (eutrophication)**

This section focuses mostly on the effects of nitrogen and phosphorus on organic matter production, as the primary reason for increasing organic matter production in the Baltic Sea has been a relatively fast enrichment of its ecosystem with limiting nutrients, i.e. man-made eutrophication (e.g. Chislock et al., 2013; Hutchinson, 1973; Smith et al., 2006).

**2.3.1 Baltic Sea nitrogen and phosphorus budgets**

Rapid load changes, in particular between the 1950s and 1980s, have disturbed the balance between nutrient inputs, their biogeochemical sinks, and their export from the Baltic Sea. As a result, combined nitrogen and phosphorus loads from land and atmosphere during the 1980s were about 3 and 4 times higher than in 1900 (Gustafsson et al., 2012). Pelagic and



sedimentary nutrient pools followed the load increase with a delay and, after loads declined significantly starting from the mid-1980s, are close to a balance with present-day nutrient loads (Gustafsson et al., 2012). Because denitrification provides an

efficient sink for nitrogen, 87% of the annual nitrogen load is removed by biogeochemical processes, compared to only 69% of the annual phosphorus load (Tab. 1, Fig. 4). In total, about 99% of the annual nitrogen and 96% of the annual phosphorus load are lost by exports and biogeochemical processes such as denitrification and burial, the remaining 1% and 4% accumulate in water column and sediments. For comparison, the pelagic pools of nitrogen and phosphorus alone hold about 5 times the annual nitrogen and 11 times the annual phosphorus load (Savchuk, 2018). Between 1950-1980, during the peak increase in

Baltic nutrient loads, about 1% of the annual nitrogen inputs and 12% of the phosphorus load accumulated in the water column alone; adding sediment storage, as much as 12% and 43% of the annual inputs accumulated in total (calculated from Gustafsson et al., 2012).

*Table 1: Nitrogen and phosphorus budgets for the entire Baltic Sea, both based on observations with water fluxes reconstructed from salt budgets, as well as on model results. Fluxes show loads from land and atmosphere, the net export to*

*the North Sea, and the biogeochemical sinks within the Baltic Sea, indicating the basin with the largest sink in brackets (BP=Baltic Proper, BS=Bothnian Sea). The sink for nitrogen is calculated as the net result of nitrogen fixation and biogeochemical sinks.*

| Source | Period | Method | N flux, ktons year-1 | | | P flux, ktons year-1 | | |
|---|---|---|---|---|---|---|---|---|
| | | | load | net sink | export | load | sink | export |
| **Liu et al. (2017)[a]** | 1971-1999 | simulated | 1102 | 1077 (BP) | 42 | 56 | 52 (BP) | 0.7 |
| **Gustafsson et al. (2017)** | 1980-2014 | simulated | 1099 | 890 (BP) | 199 | 51 | 31 (BP) | 11.6 |
| **Savchuk (2005)** | 1991-1999 | empirical | 1045 | 842 (BP) | 202 | 44 | 27 (BS) | 17 |
| **Schneider et al. (2017)** | 1985-2005 | empirical | 938 | 806 | 133 | 47 | 37 | 11 |
| **Savchuk and Wulff (2007)[b]** | 1997-2003 | simulated | 1015 | 846 (BP) | 169 | 42 | 26 (BP) | 16 |
| **Gustafsson et al. (2012)[c]** | 1997-2006 | simulated | 770 | 746 | 12 | 45 | 35 | 5 |
| **Savchuk (2018)** | 2005-2018 | empirical | 954 | 795 (BP) | 135 | 37 | 19 (BS) | 18.8 |

a) simulated with data assimilation; excludes Kattegat and transports are calculated as net flux between model grid points along the system boundary; b) simulated steady-state budget; c) includes only bioavailable nitrogen fraction



Organic compounds make up 81% of the pelagic nitrogen, but only 30% of the pelagic phosphorus pool in the Baltic Sea (Savchuk, 2018). Their long-term dynamics are poorly understood and over the past three decades (1970-2016, calculated from Savchuk (2018)) organic and inorganic nitrogen pools correlated weakly with each other (Bothnian Sea and Bay, Gulf of Finland, deep Baltic Proper, $p<0.05$, $R^2_{adj}$ 0.08-0.20). Organic and inorganic phosphorus pools correlated only in the Baltic

Proper above the halocline ($p<0.05$, $R^2_{adj}$ 0.11). The share of labile compounds that are degradable by bacteria within weeks, is generally higher for dissolved organic nitrogen (DON) and, especially, dissolved organic phosphorus (DOP) than for dissolved organic carbon (DOC) (Hoikkala et al., 2015). DON and DOP feed into biogeochemical processes via direct phytoplankton uptake, or mineralization to ammonium and phosphate via photodegradation, bacterial mineralization, or excretion by heterotrophs within the microbial loop (Hoikkala et al., 2015 and sources therein). A modelling study estimates

that 71% (89%) of phytoplankton nitrogen and phosphorus uptake in the Baltic Proper were channelled through dissolved organic pools (Kreus et al., 2015).

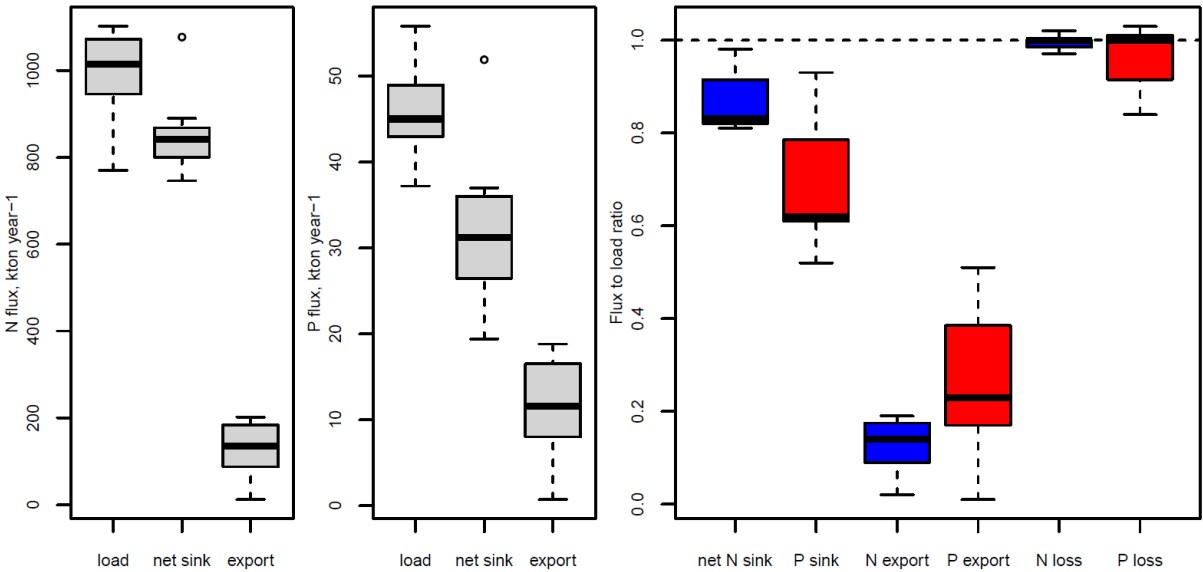

*Figure 4: Nitrogen (left) and phosphorus (middle) fluxes corresponding to nutrient budgets published for 1971-2018 (Gustafsson et al., 2012, 2017; Liu et al., 2017; Savchuk, 2005, 2018; Savchuk and Wulff, 2007; Schneider et al., 2017). The*

*right panel shows ratios between sinks, exports, and total loss as a sum of sinks and export, to external inputs. Box and whiskers show the distribution of the seven flux estimates. Boxes enclose 25%-75% percentiles, whiskers the non-outlier range.*

Nitrogen and phosphorus are exchanged intensely between different parts of the Baltic Sea. Both nutrients are transported westward from the Baltic Proper towards the Danish Straits, while phosphorus is also transported northwards because its

concentrations are successively and significantly declining from the Baltic Proper to the Bothnian Sea and the Bothnian Bay



(e.g. Savchuk, 2018 and references therein). In fact, most budgets estimate that the highest phosphorus removal takes place in the Bothnian Sea. Nitrogen, in contrast, is transported southward from the Bothnian Bay and the Bothnian Sea into the Baltic Proper, where most of the nitrogen removal takes place (see Table 1 and references therein). The net nutrient exchange between the Baltic Proper and the Gulf of Finland and the Baltic Proper and the Gulf of Riga is directed towards or away from the
Baltic Proper, depending on budget calculation method and period covered.

Nutrient pools in the Baltic Proper, which contain 61% of the entire pelagic nitrogen and 78% of the phosphorus pool (Savchuk, 2018), depend on deep water oxygen conditions. The pelagic nitrogen pool declines when deep water hypoxia expands, whereas the phosphorus pool increases (Conley et al., 2002, 2009; Savchuk, 2018; Vahtera et al., 2007). Therefore, both declining loads, as well as fluctuating bottom water oxygen conditions, have contributed to the decrease in Baltic Proper winter
dissolved inorganic nitrogen (DIN) concentrations since the 1980s, while in contrast winter dissolved inorganic phosphorus (DIP) concentrations remained stable at a high level (Andersen et al., 2017; HELCOM, 2018).

### 2.3.2 Primary production and nutrient limitation

Phytoplankton primary production is the main source of organic matter in Baltic Sea food webs. Despite its importance, temporal and spatial dynamics are difficult to describe and data coverage is poor.

Currently, primary production is excluded from the coordinated HELCOM monitoring programme (https://helcom.fi/action-areas/monitoring-and-assessment/monitoring-manual/introduction/), but measurements are included in some national programs and are also monitored for research. Most commonly measured by isotope $^{14}$C incubations, data reflect a continuum between gross and net primary production, i.e. between the carbon uptake rate into phytoplankton cells and the net effect of carbon uptake and release due to cellular respiration and exudation (Milligan et al., 2015; Sakshaug et al., 1997; Spilling et al.,
2019). Primary production is rarely measured in-situ but mostly by incubating samples at selected light levels and temperatures, which introduces methodological problems (ICES Working Group on Primary Production, 1987; Platt and Sathyendranath, 1993; Sakshaug et al., 1997). Further, water samples also contain bacteria, microzooplankton and detrital organic carbon, which all affect the fate of $^{14}$C in water samples (Spilling et al., 2019). Attempts to automate primary production measurements have focused on Fast Repetition Rate Fluorometry (FREF), an automated in-situ technique based on light absorption by
photosystem II (Houliez et al., 2017). Deriving carbon fixation from FREF is hampered by conversion efficiencies that depend on species and growth conditions (Hughes et al., 2018; Lawrenz et al., 2013), and by wavelength adaptations to capture cyanobacteria  (Houliez et al., 2017; Simis et al., 2012). Other automated measurements based on the air-sea $CO_2$ exchange describe the changes in total inorganic carbon, i.e. depict the net effect of autotroph and heterotroph processes (e.g. Schneider and Müller, 2018).

The sparse primary production measurements that cover the entire seasonal cycle show that productivity declines northward from the Baltic Proper to the Bothnian Sea and Bothnian Bay and that productivity is somewhat higher in shallow areas like the Danish Straits and the Gulf of Riga (Tab. 2). Temporal trends are best described for the Kattegat and the Danish Straits,





where Rydberg et al. (2006) found that primary production doubled between 1950 and 1980 and its seasonality had changed from uniform rates throughout the growing season to pronounced spring and autumn blooms.

*Table 2: Annual primary production (g C m$^{-2}$ year$^{-1}$) in different areas of the Baltic Sea. Updated from Savchuk et al. (2012)*

| Period | Kattegat | Danish Straits | Baltic Proper | Bothnian Sea | Bothnian Bay | Gulf of Finland | Gulf of Riga |
|---|---|---|---|---|---|---|---|
| **1970-1982** | 90-125[a] | 100-195[a] | 91-135[f] | 50-70[d] | 12-20[d] | 70-100[d] | 90-125[a] |
| **1991** | | | 100-200[c] | | | | |
| **1994-2006** | 116[b] -165[a] | 185-200[a] | 65[e]-200[d] 172[j] | 32[e]-52[d] | 16-17[d] | 80[d]-130[g] | 200[h]-250[d] |
| **2011-2012** | | | | | | | 353-376[i] |

*a) Rydberg et al. (2006), b) Carstensen et al. (2003), c) Kaczmarek et al. (1997), d) Wasmund et al. (2001) and references therein, e) Larsson et al. (2010), f) Renk (1990), g) Raateoja et al. (2004), h) Savchuk (2002) and references therein, i) Purina et al. (2018), oxygen method, j) recalculated from Gustafsson et al. (2013), Landsort Deep.*

The phytoplankton spring bloom is nitrogen-limited in the Kattegat, Danish Straits, Baltic Proper and the Gulf of Finland, N/P
co-limited in the Gulf of Riga, and phosphorus limited in the Bothnian Bay (see Schneider et al., 2015, 2017). The Bothnian Sea has gone through a change from alternating N/P limitation to N limitation of the spring bloom since the 1990s, in particular in its southern part because inflows of phosphorus-rich water from the Baltic Proper shifted the nutrient balance and probably also increased production and sedimentation (Ahlgren et al., 2017; Rolff and Elfwing, 2015). The increasing phosphorus concentrations without a matching nitrogen increase have also led to higher cyanobacteria abundance in the Bothnian Sea
(Kahru and Elmgren, 2014; Kuosa et al., 2017), and might have induced nitrogen fixation in the range of external N inputs (Olofsson et al., 2020).

Since the late 1980s, phytoplankton seasonality changed, with an earlier spring bloom and a delayed autumn bloom in coastal (Wasmund et al., 2019) and open areas of the Baltic Proper (Kahru et al., 2016). The length of the growing season roughly doubled between 1998 and 2014 and the biomass maximum shifted from spring to summer (Kahru et al., 2016). Spring bloom
intensity correlated with an index of winter DIN and DIP concentrations (Baltic Proper, Fleming and Kaitala, 2006; Groetsch et al., 2016), or winter DIN concentrations alone (Arkona and Bornholm basins, Raateoja et al., 2018). Declining nutrient levels since the end of the 1990s led to a slight drop in peak spring biomass (Groetsch et al., 2016), while higher water temperature, more intense solar radiation and low wind speed caused longer blooms and a faster transition between spring and summer communities (Groetsch et al., 2016).



However, it is currently unclear whether changes in phytoplankton seasonality also affect phytoplankton nutrient uptake and total primary production. In the global ocean, increased stratification is expected to lead to a decrease in primary production because of a reduced nutrient supply from deeper water layers (IPCC Special Report on the Ocean and Cryosphere in a Changing Climate, 2019). In the Baltic Sea, both temperature and salinity stratification have strengthened since the early 1980s (Liblik and Lips, 2019). However, in the Baltic Sea increased stratification reduces bottom oxygen concentrations (Meier et

al., 2018a) and will thus affect nitrogen and phosphorus turnover.

### 2.3.3 Nitrogen fixation

Nitrogen fixation by cyanobacteria is a substantial source of nitrogen to the Baltic Sea, with inputs comparable to riverine and atmospheric loads (see e.g. Schneider et al., 2015, 2017). Warm (Jaanus et al., 2011; Kaiser et al., 2020; Kanoshina et al., 2003; Laamanen and Kuosa, 2005; Lips and Lips, 2008; Mazur-Marzec et al., 2006) and calm conditions (Kanoshina et al.,

2003; Mazur-Marzec et al., 2006) seem to favour cyanobacteria blooms. At time scales longer than five years, surface accumulations are related to the total amount of phosphorus in the Baltic Sea, bottom water hypoxia and temperature (Kahru et al., 2020), at shorter time scales cyanobacteria blooms follow pronounced three-year oscillations, probably caused by biological feedback mechanisms (Kahru et al., 2018).

The dinitrogen fixed during cyanobacterial blooms becomes readily available to other primary producers and stimulates

summer production in the entire food web, from zooplankton and benthos to fish (e.g. Karlson et al., 2015; Motwani et al., 2018; Svedén et al., 2016). Diazotrophs thus relieve ecosystem production from nitrogen limitation and enable communities to make use of the phosphorus pool left in the water column at the end of the spring bloom (Nausch et al., 2008; Raateoja et al., 2011; Rahm et al., 2000; Schneider et al., 2017; Wasmund, 1997).

Nitrogen fixation is therefore an important link in the vicious circle of Baltic Sea eutrophication (Vahtera et al., 2007). When

the oxygen demand for organic matter mineralization exceeds the limited oxygen supply to deep-water layers and sediments, oxidation of organic matter below the halocline gives rise to hypoxia/anoxia with corresponding redox alterations of the nutrient cycles (cf. Sections 2.5-6). In the vicious circle, the inorganic nitrogen removal due to denitrification in the hypoxic zone and the release of phosphates from iron-humic complexes in the anoxic zone results in a Redfield excess of phosphorus. Cyanobacteria channel the phosphorus excess into biotic cycling via nitrogen fixation, thus increasing primary production,

sedimentation and decomposition of organic matter, which, in turn, leads to further expansion of hypoxic and anoxic zones with increased denitrification and DIP release (Savchuk, 2018 and references therein; Vahtera et al., 2007). In the Baltic Proper, cyanobacteria blooms followed the increasing phosphorus loads with a lag of 20 years (Kaiser et al., 2020). The large-scale manifestation of the vicious circle has been empirically supported by the significant correlations between satellite-detected cyanobacteria surface accumulations, water temperature, and integral phosphorus pool (Kahru et al., 2020; Savchuk, 2018). It

has been suggested that in recent years, this self-sustaining positive feedback is further reinforced by increased oxygen





consumption in saline inflowing waters, making inflows less efficient in aerating the water below the halocline (Meier et al., 2018b).

### 2.3.4. Expected future changes

Given present-day nutrient loads, Baltic Sea biogeochemical models show that nutrient turnover and productivity will increase
in warmer climates (Meier et al., 2012a, 2012b, 2012c, 2014, 2018a). Still, load reductions will affect Baltic Sea nutrient concentrations and productivity more than climate change, as seen by a large ensemble of six coupled physical-biogeochemical models and 58 transient simulations (Meier et al., 2018a). Expected changes differ between the more eutrophic southern areas and the northern basins. In the Baltic Proper, climate change will increase primary production because warming and reduced bottom-water oxygen levels will intensify nutrient turnover. Higher pelagic regeneration will benefit phytoplankton, whereas
benthic production depending on export from the euphotic zone, will decline (Ehrnsten et al., 2020). In the more oligotrophic Bothnian Sea and Bothnian Bay primary production might decrease because higher DOC inputs reduce transparency and favour heterotrophic bacteria (Andersson et al., 2015).

Stratification and future bottom water oxygen conditions will play a major role in future productivity and nutrient concentrations in the Central Baltic Sea. With present nutrient loads, bottom water oxygen is expected to decline because of
increasing stratification (Meier et al., 2018a). Warmer inflows take up less oxygen from the atmosphere and carry less dissolved oxygen into bottom waters (Meier et al., 2011; Skogen et al., 2014), where hypoxia, in turn, intensifies phosphorus cycling (Meier et al., 2011). In high-warming scenarios, also global mean sea level rise will start to contribute to increased salt-water inflows and stratification (Meier et al., 2017). Stratification changes are uncertain in sub-basins that receive saltwater intrusions from the Baltic Proper, in particular the Gulf of Finland (Meier et al., 2019a), where lateral intrusions drive stratification
(Vankevich et al., 2016), depending on halocline position in the Baltic Proper (Meier et al., 2019a). Both the Gulf of Finland and the Bothnian Sea might become less stratified in the future, with higher bottom water oxygen concentrations and reduced phosphorus turnover (Meier et al., 2018a).

Future nutrient loads, warming, and changes in stratification and oxygen conditions will also shift the balance between nitrogen removal via denitrification and nitrogen fixation, which determines Baltic Sea nitrogen levels (Skogen et al., 2014). Future
warmer climates will have longer periods suitable for cyanobacteria growth (Hense et al., 2013; Neumann et al., 2012). Simulations suggest that denitrification can counteract nitrogen fixation and external inputs except in load increase scenarios (Skogen et al., 2014), where the fraction of nitrogen inputs removed by denitrification starts to decline (Meier et al., 2012b).

The Baltic Sea, especially its entrance area, will also be affected by changes in nutrient concentrations in the inflowing North Sea water. These are highly uncertain and mostly determined by water exchange with the North Atlantic (Meier et al., 2019a;
Skogen et al., 2014). Assuming that changes at the North Sea – Atlantic boundary would halve Skagerrak nitrogen and phosphorus concentrations, primary production would drop by 40% in the Kattegat and 10% in the Gotland Sea (Meier et al., 2019a).



In the more oligotrophic northern Baltic Sea, climate change effects will be modulated by increasing DOM inputs (Andersson et al., 2015). DOM from the northern part of the drainage basin contains high amounts of colored dissolved organic matter (CDOM), which is conservatively mixed over a large salinity range (Harvey et al., 2015). Mesocosm experiments (Paczkowska et al., 2020) and field studies in estuaries (Andersson et al., 2018) show that the deteriorating light climate at high DOC inputs can counteract nutrient effects on primary production. In field studies this effect is found in estuaries with high CDOM combined with low nutrient input, leading to a decline in primary production and an increase in bacterial production (Andersson et al., 2018). Suppressed phytoplankton growth and food-web shifts to microbial loop dominance are seen when DOC concentrations increase by 25%-30% on top of background levels (Andersson et al., 2013; Lefébure et al., 2013), probably modulated by nutrient competition between phytoplankton and bacteria (Meunier et al., 2017)

### 2.4 Changes in organic matter remineralization and oxygen availability

### 2.4.1 Oxygen supply

Dissolved oxygen concentration in the water column is controlled by the supply of oxygen by vertical and lateral transports and by oxygen consumption in the water column and sediment (e.g. Savchuk, 2018). In the eutrophied Baltic Sea, water below the permanent pycnocline is oxygen-depleted because both large but sporadic barotropic inflows of oxygenated saline water from the North Sea, so-called Major Baltic Inflows (MBIs) (e.g. Mohrholz et al., 2015) and smaller inflows preferably ventilating the halocline but also sometimes deeper layers (e.g. Feistel et al., 2003; Meier et al., 2004; Neumann et al., 2017), do not always compensate for the oxygen consumption due to organic matter remineralization after the spring and summer blooms. Three narrow straits and shallow sills in the western Baltic Sea constitute natural obstacles, constraining the free water exchange with the world ocean (Matthäus et al., 2008). As a result, the deep-water layer of the central basins is prone to hypoxic or even euxinic conditions.

Although MBIs explain only about 20% of the total salt input (Mohrholz, 2018), they are the only mechanism that can ventilate the deeper parts of the Baltic Sea. In this respect, the Baltic Sea is special compared to other coastal seas because it is characterized by a largely varying topography with the deepest areas in the central-eastern (Gotland Deep, ~250 m) and northwestern (Landsort Deep, ~459m) Gotland Basin and long water residence time. For the period 1887-2017, MBIs do not show a systematic trend, but a pronounced multi-decadal variability of about 25-20 years (Mohrholz, 2018). On average MBIs occur once per year. However, there are longer periods without any MBI, so-called stagnation periods, e.g. during 1983-1992. According to model simulations, such periods without MBIs and with decreasing salinity are part of the natural variability of the system and occur once per century on average (Schimanke and Meier, 2016).

During periods with lower average salinity, deeper halocline and weaker vertical stratification, vertical fluxes of oxygen are larger and capable of ventilating the bottom water along the rim of the sub-basins with permanent halocline such as the Bornholm and Gotland basins (Väli et al., 2013). Hence, during stagnation periods without MBIs basin-wide hypoxic areas



are smaller compared to periods with many MBIs although the very deep areas of the Baltic Sea suffer from oxygen depletion
(Conley et al., 2002, 2009; Meier et al., 2017).

Another important process is the entrainment of ambient water into the inflowing gravity-driven saltwater plumes (Kõuts and
Omstedt, 1993; Meier et al., 2018a; Neumann et al., 2017). Most of the oxygen arriving with MBIs in the bottom water at the
Gotland Deep is oxygen from the Baltic Sea interior and not from the Baltic Sea entrance area (Neumann et al., 2017). Due to
a strong internal vertical re-circulation, the entrainment of inflowing water is considerable. From 20-year long records of
observations, Kõuts and Omstedt (1993) showed that the inflowing dense water is diluted by surface water and that the flow
increased by a factor of four on its way between the Kattegat into the Landsort Deep.

Further, no trend was found in the impact of wind-induced vertical turbulent mixing on the multi-decadal variations in salinity
(Radtke et al., 2020).

In summary, the ventilation of the Baltic Sea deep water is very intermittent, given by the frequency of MBIs and shows
pronounced multi-decadal variations with improved bottom oxygen conditions during stagnation periods. Due to the large
interannual to multi-decadal variability, systematic trends in deep water ventilation on the centennial time scale are difficult
to detect. However, long-term changes in oxygen consumption are pronounced (Carstensen et al., 2014; Gustafsson et al.,
2012; Meier et al., 2018b, 2019a, 2019b) and will be discussed in the following.

**2.4.2 Organic matter remineralization**

The predominant sink for oxygen and other oxidants are bacterially mediated degradation processes of organic matter. These
processes take place in the water column as well as in the marine sediments. Fluxes between water and sediment connect both
environments and thus sedimentary oxidant sinks impact the water column and vice versa. The oxygen flux between the
atmosphere and ocean keeps the oxygen concentration in the mixed surface layer close to equilibrium with atmospheric
oxygen. Therefore, oxygen deficiency becomes important below a temporal or permanent pycnocline, which hampers an
oxygen flux from the surface. In the Baltic Sea, both regions with temporal and permanent oxygen deficiencies exist.

Carstensen et al. (2014) reported an about 10-fold increase in hypoxic area within the last 115 years. Their analysis is based
on oxygen profiles and estimated total apparent oxygen utilization. As the primary cause for the increase, they detected
increased nutrient loads from land, but also climate warming. Similar results were found using modelling (Gustafsson et al.,
2012; Meier et al., 2019b, 2019c).

For the stagnation period 2004-2014, Schneider and Otto (2019) did not find an interannual variability of oxygen consumption
in the eastern Gotland Basin. Furthermore, the mineralization rate was relatively constant at 2 mol carbon m$^{-2}$ year$^{-1}$. The
detected mineralization rate is independent of the redox conditions in the water column and were based on observed total
inorganic carbon dynamics.



The oxygen consumption rates after Major Baltic Inflows have increased since the 1970s (Meier et al., 2018b). This trend was
detected from observations as well as from model simulations. The explanation, derived from the model study, is the increased
abundance of POM, especially zooplankton, which is mixed into the inflowing water. The higher POM concentration in the
inflowing bottom water apparently accelerates oxygen consumption due to respiration and mineralization.

An additional explanation was provided in a model simulation study by Neumann et al. (2017). The focus on MBIs only is not
sufficient to quantify the oxygen supply to the deep Baltic Sea waters. For the MBIs 2003 and 2014, it could be shown that
the 2003 event was accompanied by several smaller events altogether exceeding the stronger 2014 MBI 1.5-fold. A difference
in oxygen consumption rates immediately after the MBIs in 2003 and 2014 could not be shown.

An example for hypoxia dynamics in the Gulf of Finland was given in an observational study by Stoicescu et al. (2019). This
region shows a strong seasonality due to a temporal pycnocline. The oxygen consumption during an established pycnocline
was estimated to 0.31-0.82 mg l$^{-1}$ month$^{-1}$ for 2016 and 2017, respectively.

**2.4.3 Changes in oxygen concentration**

The individual sub-basins of the Baltic Sea are characterized by very different hydrodynamics, which is strongly reflected in
the ventilation and resulting oxygen content of the deep waters. Bottom water hypoxia and anoxia with different degrees of
deoxygenation and frequencies of occurrence makes the Baltic Sea an ideal study region for hypoxia-related processes. The
availability of long-term data series, uninterrupted for at least the last 70 years, is an additional valuable asset in this regard.
Figure 5 shows the development of dissolved oxygen and hydrogen sulphide concentrations in the near-bottom layer of key
stations in the center of three of these basins.

The Arkona Basin in the southwest is  located close to the Danish straits, the sills and narrow straits connecting the Baltic to
the Kattegat. It is characterized by highly dynamic oxygen concentrations in the deep water, which are rarely hypoxic) and
influenced even by weak and short inflow pulses via the Danish straits, which occur at a high frequency. The long-term mean
of 6.3 ml/l (141 µmol/l) is well above the hypoxic limit, with no sign of reduction over time.

In the Bornholm Basin, located east of the Arkona Basin, with a water depths of ~90 m, bottom oxygen concentrations are
lower, and saltwater intrusions less frequent. Only more intense inflow events, occurring within annual to multiannual
intervals, ventilate the bottom waters of this basin, leading to alternating oxic and hypoxic deep-water conditions. The long-
term mean of 1.7 ml/l (38 µmol/l) indicates mostly hypoxic bottom waters, with alternating oxic phases. From 1980 to 2010
even euxinic conditions with occurrence of H$_2$S were observed, but these conditions have not reoccurred during the last decade.

In the central sub-basins, conditions are more stagnant and deep-water renewals/ventilations are rare and linked to major Baltic
inflows (Mohrholz, 2015). As an example, the conditions in the Gotland Deep (Eastern Gotland Basin) show the persistence
of hydrogen sulphide during the last decades with a mean of -0.9 ml/l (i.e. -20.2 µmol/l oxygen equivalents or 10.1 µmol/l
hydrogen sulfide) over time (Fig. 5), and three clearly discernible ventilation events caused by MBIs since 1990.





*Figure 5: Long-term variations of dissolved oxygen concentrations (blue points) and hydrogen sulphide (red points, converted to negative oxygen equivalents) in the near-bottom layer at the key stations Arkona Basin (BY1, TF0113, 45m), Bornholm Deep (BY5, TF0213, 92m) and Gotland Deep (BY15, TF0271, 245m) (source: IOW-DB, ICES).*

A lateral view of the recent oxygen (and $H_2S$) distribution in the water column is displayed along a cross-section from the western to the central Baltic Sea for July 2020 (Fig. 6). After the latest period of intense inflow activity during 2014 to 2017 and several ventilation events of the central deep water (Neumann et al., 2017), the environmental status switched back to stagnant conditions, resulting in large hypoxic to euxinic water volumes below 70 m water depth. Hansson et al. (2019) calculated a hypoxic area of 82.000 $km^2$ (32 % of the Baltic Proper) and a hypoxic volume of 3100 $km^3$ (22 % of the Baltic Proper water volume) for late summer 2019. Such levels were already reached sporadically in the 1970s, but have become


more persistent during the recent two decades (see also Fig. 5). Furthermore, hydrogen sulphide concentrations are higher and
more persistent in the last two decades compared to earlier times. Worsening of the oxygen deficit by increasing concentrations
of hydrogen sulphide due to amplified oxygen consumption rates over the last decades was shown by model simulations
comparing the last 150 years in more detail (Meier et al. 2018b).

Under recent oceanographic and climatic conditions, with high freshwater discharge and strong water column stratification,
hypoxia nearly attains the maximum spatial extent possible for this ecosystem.

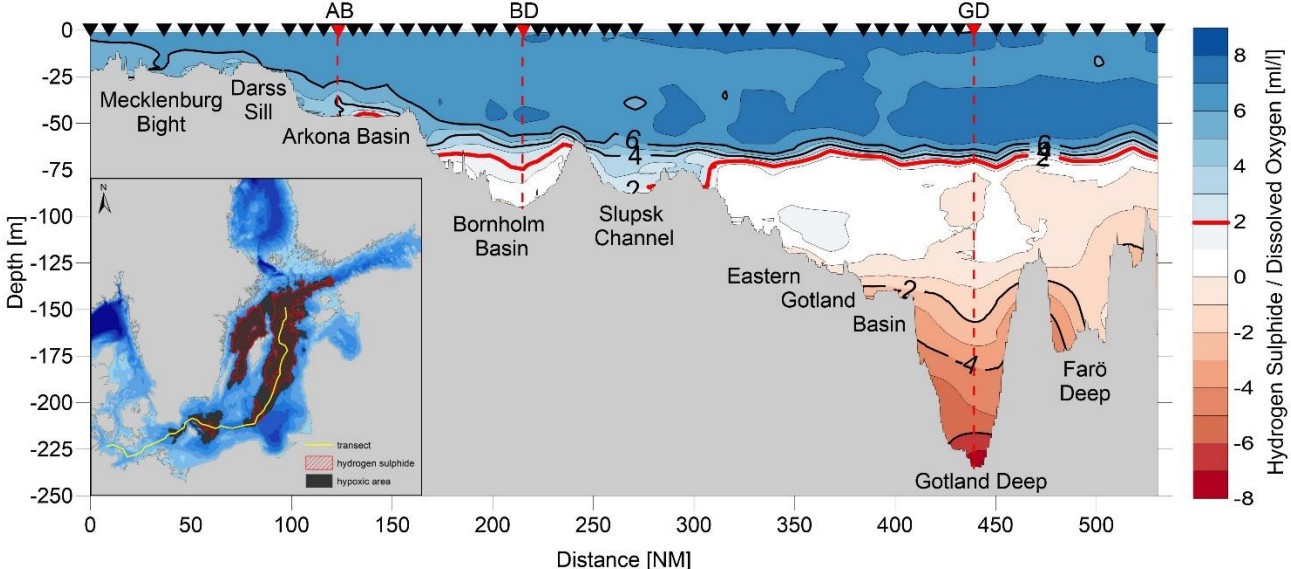

*Figure 6: Transect from the Great Belt to Baltic Proper, showing the recent state of dissolved oxygen and hydrogen sulphide*
*(converted to negative oxygen equivalents) concentrations along the pathway of saltwater inflows from the western Baltic Sea*
*to the deep basins (July 2020, unpublished data of cruise EMB242, IOW). Locations of CTD measurements and water sampling*
*are marked by black triangles along the track. Red triangles mark the positions of the key stations Arkona Basin (AB),*
*Bornholm Deep (BD) and Gotland Deep (GD) adressed also in Fig. 5. The map (left lower corner) shows the position of the*
*transect (yellow line) and the hypoxic and euxinic area (dark grey and red hatch)*

**2.5 Burial and turnover of C, N, P in the Baltic Sea sediments**

**2.5.1 Spatial variability in the deposition of $C_{org}$ at the sediment surface**

Aphotic seafloor sediments constitute an essential compartment within the marine biogeochemical cycles of C, N and P (Aller,
2014; Burdige, 2006). Settling of reactive organic matter at the base of the water column allows sediments to host a multitude
of microbially-mediated and abiotic reactions involving organic matter and the products of its degradation (Berner, 1980;
Schulz and Zabel, 2006). The rate of deposition of organic carbon ($C_{org}$)at the seafloor is a key variable determining rates of
sedimentary carbon remineralization, as well as the rate of sedimentary carbon burial. The Baltic Sea today is characterized





by a strong N–S gradient from oligotrophic through mesotrophic to eutrophic conditions (Section 2.3), which exerts a first-order control on potential $C_{org}$ deposition due to vertical settling of autochthonous organic matter (Section 2.4, Tamelander et al., 2017). Deposition of this material is modulated by transformations in the water column, including aggregation and degradation processes, and impacted by transport through currents and seafloor morphology (Leipe et al., 2011). A key process affecting net $C_{org}$ deposition in the Baltic Sea is wind-wave driven resuspension and lateral transport of fine-grained sediments

away from shallower areas (Almroth-Rosell et al., 2011; Leipe et al., 2000). This leads to the accumulation of fine-grained sediments in the central deep basins, with $C_{org}$ contents of 12–16 wt%, (Leipe et al., 2011). Terrestrial organic matter ($C_{org-T}$) delivered by rivers mainly enters the Baltic Sea in dissolved form (Gustafsson et al., 2014; Mattsson et al., 2005), but undergoes flocculation and settling along the salinity gradient, leading to accumulation in sediments (Asmala et al., 2014; Jilbert et al., 2018). Estimates based on lignin biomarker analyses suggest that 10–30% of sedimentary $C_{org}$ in the Baltic Sea is $C_{org-T}$, with

clear inter-basin differences (Miltner and Emeis, 2001). Erosion of earlier-deposited marine sediments due to glacio-isostatic uplift is important in northern regions (Virtasalo and Kotilainen, 2008) and may constitute an additional source of $C_{org}$ to modern accumulation areas.

Deposition rates of $C_{org}$ in the Baltic Sea have been estimated from both sediment traps (e.g., Gustafsson et al., 2013; Heiskanen and Tallberg, 1999; Lehtonen and Andersin, 1998; Struck et al., 2004) and surface sediment data (e.g. Leipe et al., 2011;

Nilsson et al., 2019; Winogradow and Pempkowiak, 2014). Of these approaches, the latter may be considered preferable due to the integration of deposition over a longer period (e.g. 2–10 yr in a sediment slice of 0–2 cm). An important consideration, as outlined by Nilsson et al. (2019), is that estimates for $C_{org}$ accumulation based on surface sediment $C_{org}$ contents and local mass accumulation rates represent deposition rates, rather than ultimate burial rates in sediments. Remineralization of $C_{org}$ persists throughout the uppermost decimeters of the sediment column, modulating burial. Nilsson et al. (2019) therefore

adopted a closed-sum budget approach, based on remineralization rates from DIC effluxes and true burial rates estimated from deeper sediment intervals, to calculate a net rate of $C_{org}$ deposition of 22.8 ± 7.76 Tg C/yr for the entire Baltic Sea. The deposition is higher in deep accumulation areas than shallower erosion and transport areas, and varies per basin, with the highest values observed in the Baltic Proper (Nilsson et al., 2019).

**2.5.2 Cycling of $C_{org}$ in upper sediment column: controls on primary redox reactions**

Oxygen has the highest energy yield of any of the electron acceptors (EA) used in $C_{org}$ remineralization, hence spatial and temporal variability in bottom-water oxygen supply is a key factor determining the vertical zonation of diagenetic reactions in Baltic Sea sediments (e.g., Lehtoranta et al., 2009). In regions of oxic bottom waters, diffusive penetration of oxygen into muddy sediments is typically in the order 1–5 mm (Bonaglia et al., 2013; Hietanen and Kuparinen, 2008). Below this depth, sediments are characterized by a light-coloured *suboxic* zone in which reduction of nitrate (i.e. *denitrification*), and manganese

(Mn) and Fe oxides mostly dominate (Fig. 7). When these EAs are exhausted, sulfate reduction becomes the dominant remineralization pathway and the sediment acquires a darker colour due to the associated formation of iron sulfides. In regions where $C_{org}$ deposition is sufficiently high, sulfate too may be completely exhausted within the uppermost decimeters of the



sediment column, leaving methanogenesis as the only remaining remineralization pathway. In persistently anoxic areas of the deep basins, the entire oxic and suboxic zone is usually absent from the sediment column (Fig. 7) and remineralization is

dominated by sulfate reduction and methanogenesis. Transitional states between the end-member examples in Fig. 7 exist around the margins of the deep basins and may be observed transiently following major Baltic Inflows and the associated re-oxygenation of deep waters (Dellwig et al., 2018; Hermans et al., 2019). The dissolved reduced products of redox reactions in the sediment column, including methane, ammonium and divalent metal cations, can diffuse vertically and can consume EAs through secondary redox reactions, further influencing the diagenetic zonation.

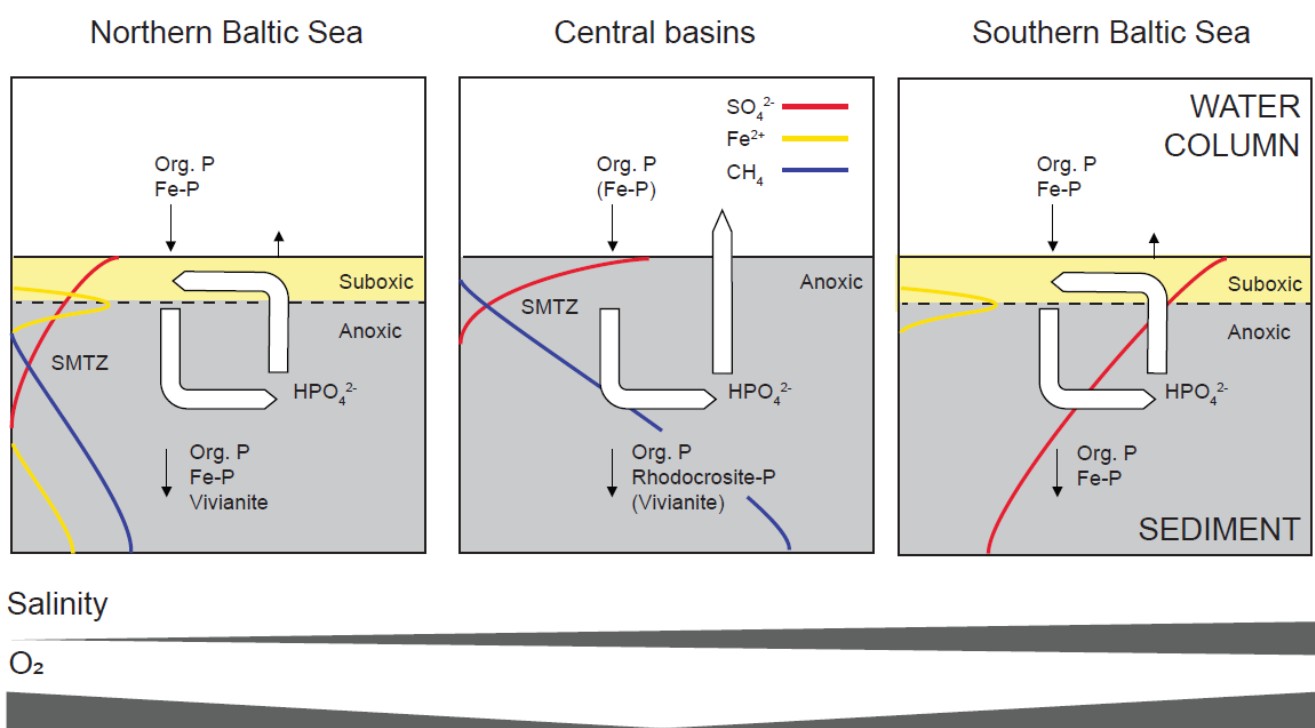


*Figure 7: Schematic of vertical biogeochemical zonation of Baltic Sea sediments from a perspective of P cycling (partially redrawn from Lehtoranta et al., 2009). Coloured lines indicate pore water concentrations of key species (zero to left). In areas with oxic bottom waters, remineralization of OM in the suboxic zone proceeds by denitrification and metal oxide reduction. Remineralization proceeds by sulfate reduction and methanogenesis in the anoxic zone, above and below the sulfate-methane*

*transition zone (SMTZ), respectively. Processes impacting on regeneration and burial of P are shown with annotated arrows. Remineralization produces orthophosphate ($HPO_4^{2-}$), which may diffuse to the bottom waters, become trapped in surface-sediment Fe oxides, or incorporated into authigenic minerals, depending on the depositional environment (see text). Note that exceptions to these schematic representations can occur (e.g. a shallow SMTZ was observed in the Gdansk Basin, Southern Baltic, due to high rates of $C_{org}$ deposition and sulfate reduction, Brodecka et al., (2013)).*





In addition to determining the zonation of primary redox reactions, oxygen availability also, directly and indirectly, influences
the overall rates of microbial $C_{org}$ remineralization. Organic matter degradation through sulfate reduction and methanogenesis
have slow reaction kinetics (Moodley et al., 2005) and are typically assumed to proceed at half the rate of the higher-energy
yield reactions in diagenetic models of Baltic Sea sediments (Radtke et al., 2019; Reed et al., 2011). Hence, respiration of
organic matter proceeds more slowly in the sediments of the deep anoxic basins. Oxygen also strongly controls the distribution

of benthic fauna in the Baltic Sea (Carman and Cederwall, 2001; Gogina et al., 2016; Laine, 2003). Within oxic areas, benthic
fauna may directly ingest and respire $C_{org}$ (Ehrnsten et al., 2019) as well as modify rates of microbial processes through
bioturbation and bioirrigation (e.g., Kristensen et al., 2011). As a rule, the presence of fauna enhances $C_{org}$ remineralization,
for example through enhancing the supply of EAs in the upper sediments (Arndt et al., 2013). However, benthic community
composition has an important role in determining the exact mechanisms. In a comparative study, Kristensen et al. (2011)

showed that the invasive polychaete *Marenzellaria viridis* enhanced net sediment respiration by stimulation of microbial
sulfate reduction in its I- and J-shaped >20 cm deep burrows. In contrast, the native *Nereis diversicolor* enhanced aerobic
remineralization through well-flushed U-shaped burrows in the shallow sediments.

The North-South salinity gradient of the Baltic Sea exerts an important secondary control on the availability of two key EAs,
namely sulfate and Fe oxides. The low salinity of the Bothnian Bay and the Bothnian Sea leads to bottom water sulfate

concentrations < 5 mmol/L, limiting the potential reservoir for microbial sulfate reduction. Under present-day conditions of
relatively high $C_{org}$ deposition, sulfate penetration in the sediment column of the Bothnian Sea is < 10 cm (Egger et al., 2015a;
Slomp et al., 2013). Consequently, the sulfate-methane transition (SMTZ) and underlying methanogenic zone are encountered
at a relatively shallow depth in the sediment column (Fig. 7), increasing the likelihood that a significant fraction of accumulated
$C_{org}$ will degrade via methanogenesis. Similar porewater profiles have been observed in other areas of the northern Baltic Sea

(Jilbert et al., 2018; Myllykangas et al., 2020a; Sawicka and Bruchert, 2017) indicating a key role for methanogenesis in low-
salinity, eutrophied regions. In contrast to sulfate, the availability of sedimentary Fe oxides is generally greater in the low
salinity northern regions, as a result of high supply rates of Fe from peatland-rich catchments (Sarkkola et al., 2013), limited
loss to flocculation due to the weak coastal salinity gradient (Jilbert et al., 2018), and less sulfidization of Fe in surface
sediments.

**2.5.3 Quantifying remineralization and burial rates of $C_{org}$**

Using DIC fluxes in benthic chamber *in situ* experiments, Nilsson et al. (2019) estimated net sedimentary $C_{org}$ remineralization
of 21.8 ± 7.76 Tg C/yr for the entire Baltic Sea. This value is on average equivalent to 96% of the estimated depositional flux,
implying that the vast majority of $C_{org}$ deposited at the sediment surface is recycled to the water column as DIC. It is also
considerably higher than the rate of 1.04 Tg C/yr estimated in an earlier budget (Kuliński and Pempkowiak, 2011), and

approximately twice the value simulated in a modelling study by Gustafsson et al. (2017).



Attempts to quantify relative rates of primary redox reactions within the sediment column at a given location have typically been made by diagenetic modelling. Reed et al. (2011) showed that in a muddy sediment location in the Arkona Basin, oxygen accounted for >50% of total $C_{org}$ remineralization throughout the year, even under hypoxic (<63 µmol/L $O_2$) bottom water conditions during late summer. Reduction of nitrate, Fe oxides and sulfate each accounted for 10–20%, depending on the
season, while methanogenesis was insignificant in shallow-sediment diagenesis at this southerly location. Conversely, a similar model estimated that up to 40% of total $C_{org}$ remineralization at a muddy deep site in the Bothnian Sea proceeds via methanogenesis, due to the shallow penetration of porewater sulfate at this location (Rooze et al., 2016). In the deep anoxic basins of the Baltic Proper, remineralization in muddy sediments is dominated by sulfate reduction and methanogenesis. However, as shown by Reed et al. (2016), temporal fluctuations in shelf-to-basin transport of Fe oxides (cf. Lenz et al., 2015)
or reoxygenation events associated with MBIs (Matthaus and Franck, 1992) may lead to a transient role for Fe oxides in $C_{org}$ remineralization in the deep basins. Until recently, estimates of process rates in permeable sandy sediments have been lacking. However, new results from the south-western Baltic Sea region show that both oxygen uptake and sulfate reduction rates are similar in sandy sediments compared to nearby muds (Lipka et al., 2018). Similarly, Bartl et al. (2019) detected comparable rates of denitrification between permeable and non-permeable sediments offshore from the Vistula estuary. Extrapolation of
location-specific process rate estimates to the basin-scale remains fraught with difficulty, but advances in high-resolution sediment typology (e.g. Tauber, 2014) and coupled water column–sediment biogeochemical modelling (Radke et al., 2019) offer high potential for improvement in this area.

Nilsson et al. (2019) estimate a modern $C_{org}$ burial rate of 0.98 + 0.31 Tg C/yr for the entire Baltic Sea, based on $C_{org}$ contents below the zone of upper-sediment diagenesis and estimated mass accumulation rates from 30 locations. As first indicated by
Emeis et al. (2000), such values are greatly over the pre-industrial background, confirming that eutrophication of the Baltic Sea has enhanced the rate of carbon burial. The value estimated by Nilsson et al. (2019) also corresponds well with the 0.91 Tg C/yr simulated in the modelling study of Gustafsson et al. (2017). Both approaches effectively consider burial to occur only in deep basin accumulation areas, for example as defined by Håkanson and Jansson (1983) and Carman and Cederwall (2001). However, high caesium isotope ($^{137}$Cs)-based sediment mass accumulation rates have also been observed in many coastal
locations traditionally considered to be erosion and transport areas (Mattila et al., 2006). Using these data and sediment $C_{org}$ contents, Leipe et al. (2011) showed that significant rates of $C_{org}$ burial may occur in shallow settings. Therefore, it is likely that the true rate of carbon burial in the Baltic Sea as a whole is greater than the estimate of Nilsson et al. (2019). Future studies should try to identify locations ("depocenters") with high sediment accumulation rates within the Baltic erosion and transport bottom areas to get an improved comprehensive estimate of the integrated $C_{org}$ burial rate.

**2.5.4 Specific aspects of sedimentary nitrogen and phosphorus cycling**

Organic nitrogen and phosphorus are deposited, remineralized and buried in sediments together with $C_{org}$, hence several aspects of the sedimentary cycling of these elements follow what has already been described for carbon. However, N and P cycling is also impacted by multiple further reactions in sediments, which influence their ultimate fate. Sedimentary nitrogen cycling is





most dynamic in areas of oxic bottom waters overlying organic-rich sediments (Carstensen et al., 2014). Under these
conditions, ammonium released during remineralization may be nitrified, facilitating nitrate reduction processes in the upper
sediment column (principally denitrification, leading to loss of fixed N as $N_2$ gas; and dissimilatory nitrate reduction to
ammonium (DNRA), which retains fixed N). Denitrification dominates in Baltic Sea sediments (van Helmond et al. 2020;
Hietanen and Kuparinen, 2008; Jäntti et al., 2011) although DNRA may be more active under certain conditions (Bonaglia et
al., 2017; Jäntti and Hietanen, 2012). Anaerobic ammonium oxidation (*anammox*), leading to loss of fixed N as $N_2$ gas, has
been found to be insignificant in Baltic Proper sediments (Hylén et al., unpublished results). Microbial denitrification rates
have been shown experimentally to be accelerated by the presence of bioturbating meiofauna, especially of the phylum
*Nematoda* (Bonaglia et al., 2014). In the deep anoxic basins, ammonium released during remineralization diffuses out of the
sediments and accumulates in the sub-halocline water mass, from where it may be oxidized upon entrainment into the halocline
(Dalsgaard et al., 2013) or during inflow events (e.g., Myllykangas et al., 2017).

Similarly to ammonium, orthophosphate released during remineralization is most likely to participate in complex further
cycling in sediments underlying oxic bottom waters. The association of P with sedimentary Fe oxides in the Baltic Sea is well
established (e.g. Lehtoranta et al., 2009; Mort et al. 2010). Phosphorus is adsorbed or co-precipitated with oxide minerals when
these are present in surface sediments, limiting the P efflux to bottom waters (Fig. 7). Conversely, when these oxides are
reduced, the P efflux increases. Seasonal and multiannual variations in the size of the hypoxic area in the central Baltic,
therefore, induce periods of net release or retention of P from sediments around the margins of the deep basins (Conley et al.
2002; Reed et al., 2011). Shelf-to-basin shuttling of oxide-bound P also generates a maximum in P efflux as this material
crosses the hypoxic transition zone at the halocline (Almroth-Rosell et al., 2015). High P effluxes in this zone could be related
to transient release from sulfur-oxidizing bacteria such as *Beggiatoa*, which form large mats at the sediment surface and have
been shown to accumulate P in the form of polyphosphates in other marine systems (Noffke et al., 2016). In the deep anoxic
basins, orthophosphate diffuses freely across the sediment-water interface in the absence of surface sediment Fe oxides (Emeis
et al., 2000; Jilbert et al., 2011; Mort et al., 2010; Viktorsson et al., 2012, 2013). Sediment $C_{org}/P_{org}$ ratios show that the relative
rate of P regeneration from organic matter under anoxic conditions is elevated with respect to C (Jilbert et al., 2011), as
confirmed by creating sub-Redfield C/P ratios in the efflux at the sediment-water interface (Viktorsson et al., 2012, 2013).
This phenomenon is related to the low P demand of the sediment microbial community (Steenbergh et al., 2011).

Despite efficient remineralization of P in the sediment column, the efflux of orthophosphate across the sediment-water
interface in the Baltic Sea is modulated by the formation of authigenic P-bearing minerals, which enhance solid-phase P burial
rates. Three principal groups of minerals have been identified in Baltic Sea sediments: calcium (Ca)-phosphates such as
carbonate fluorapatite (CFA), Mn-Ca carbonates such as rhodocrosite (Jilbert and Slomp, 2013); and Fe (II) phosphates such
as vivianite (Dijkstra et al., 2016; Egger et al., 2015b). Of these, CFA formation is relatively unimportant in the Baltic in
comparison with open ocean margin sediments (Mort et al., 2010), while the others are characteristic of the Baltic Sea and
may constitute important burial phases of P in certain locations. Both rhodocrosite-bound P and Fe (II) phosphates have been





shown to accumulate in deep basin sediments due to shelf-to basin shuttling of precursor Fe and Mn oxide minerals (Dijkstra et al., 2016; Jilbert and Slomp, 2013; Reed et al., 2016). Fe (II) phosphates are also observed below the depth of sulfate penetration in northern regions such as the Bothnian Sea (Egger et al., 2015b), where reduction of deep-buried Fe oxides in
the absence of hydrogen sulfide leads to high porewater Fe concentrations and supersaturation with respect to vivianite (Fig. 7). A modelling study by Lenstra et al. (2018) showed that fluctuations in inputs of Fe oxides, as well as salinity, regulate the importance of vivianite as a P burial sink in Bothnian Sea sediments. The study showed that vivianite may account for over 50% of total P burial in low-salinity, high Fe environments such as estuaries.

### 2.6 Changes in the marine $CO_2$ system and Ocean Acidification

**2.6.1 The role of the inorganic carbon cycle in Baltic Sea ecosystem research**

Carbon is the main component of organic matter and thus, primary production and mineralization inevitably are connected to the fixation or liberation of carbon. Other than for nutrients like nitrate or phosphate, the bioavailable pool for carbon in seawater as well as brackish water (e.g. the Baltic Sea), due to its inorganic carbon system, is large and usually not limiting. Carbon removal from the upper layer of the sea is thus a direct measure of net production and a quantitative measure of
eutrophication (Section 2.3). As the bulk of oxygen production during photosynthesis and oxygen consumption during mineralization of organic matter is a consequence of reduction and oxidation of carbon, carbon transformation processes are the main driver of oxygen depletion (Section 2.4). As the main component of the acid-base system of seawater, the inorganic carbon system is the dominant buffering system for an increase of atmospheric $CO_2$ penetrating the marine system, leading to a decrease in pH, a process referred to as ocean acidification (OA). The imbalance of the fugacities of $CO_2$ in the atmosphere
and at the sea surface determines the flux of $CO_2$ at the air-sea interface. Investigations of the carbon cycle thus allow insights into some of the processes of main interest for today's chemical oceanography and ecosystem research in the Baltic Sea, including the function as source/sink for atmospheric carbon, ocean acidification, and the coupling of ecosystem production and formation of hypoxia.

**2.6.2 Main components of the inorganic carbon system**

While the acid-base system controlling seawater pH is regulated by a number of substances that can act as a proton donor or acceptor (Dickson et al., 2007), the dominating species are bound to the inorganic carbonate system. The inorganic carbon system is comprised of the different forms of carbonic acid and its dissociation products in seawater, including $CO_2$ solvated in water (including a very small fraction of undissociated carbonic acid) referred to as $CO_2^*$, and the two deprotonated forms bicarbonate ($HCO_3^-$) and carbonate ($CO_3^{2-}$). These are connected by two equilibrium constants (e.g. Millero, 2010). As a
convention, the equilibrium reactions of the carbonate system are usually formulated referring to concentrations rather than activities. Consequently, the equilibrium constants are functions not only of temperature and pressure, but also salinity, as they include the salinity dependence of the activity coefficients of the individual carbon system species (Müller, 2018).





The concentration of $CO_2^*$ is related to the fugacity of $CO_2$ ($fCO_2$) by the solubility constant, and the direction of $CO_2$ flux is determined by the difference of the $fCO_2$ in surface seawater and the surface-near air. Lastly, the concentration of $CO_3^{2-}$ is related to the solubility product of calcium carbonate, either in the form of calcite or aragonite.

*Main variables used to describe the inorganic carbon system*

Analytically, the inorganic $CO_2$ system can be described by four parameters, which can be directly measured (Dickson et al.,

2007); the total inorganic carbon content ($C_T$), the total alkalinity ($A_T$), the partial pressure of $CO_2$ ($pCO_2$) and the seawater pH, where

$$C_T = [CO_2^*] + [HCO_3^-] + [CO_3^{2-}]$$

represents the sum of all inorganic carbon forms in seawater;

$A_T = [HCO_3^-] + 2\,[CO_3^{2-}] + [B(OH)_4^-] + [OH^-] + [HPO_4^{2-}] + 2\,[PO_4^{3-}] + [SiO(OH)_3^-] + [NH_3] + [HS^-] - [H^+]_F - [HF] -$

$[HSO_4^-] - [H_3PO_4] + [\text{minor bases} - \text{minor acids}]$

is defined as the excess of proton acceptors over donors, which is dominated by the carbonate and bicarbonate species, and mostly determines the buffer capacity of seawater (Middelburg et al., 2020);

$pCO_2$, the partial pressure of $CO_2$, is very closely related to the fugacity (e.g. Pfeill, 2013), and determines the flux direction of $CO_2$ at the sea-air interface;

and pH is the negative decadic logarithm of the $H^+$ activity, and thus the variable most closely related to the seawater "acidity".

Due to thermodynamic equilibria, it is usually (open ocean) sufficient to determine two of the four variables (and pressure, temperature and salinity) to fully describe the inorganic $CO_2$ system. Similarly, in the open ocean, OA is predictable (e.g. Doney et al., 2009), where only the rise in atmospheric concentrations and change in surface temperature have been considered. Neither of these holds for the Baltic Sea due to peculiarities in the acid-base and carbon system.

**2.6.3 Status of carbon system measurements in the Baltic Sea**

To date, there exists no harmonized long-term effort to monitor the inorganic carbon system in the Baltic Sea. Yet, apart from campaign and research project-based data, some long-term data series exist. $A_T$ has been measured continuously as part of the Swedish monitoring program and temporarily by other countries, allowing robust trend analysis since the mid-1990s (Müller et al., 2016). pH has been measured mostly as a side parameter to biological monitoring, mostly based on glass electrodes with

some restrictions on a system with variable salinity, such as the Baltic Sea. Yet, some long-term trends could be depicted from these data (Carstensen and Duarte, 2019). Recent technological and chemical breakthroughs will allow the use of traceable precise spectrophotometric pH measurements in the Baltic (Müller and Rehder 2018; Müller et al., 2018a, 2018b). $pCO_2$ has been measured continuously aboard commercial ships between Helsinki and Lübeck since 2003 (Schneider and Müller, 2018),





and have been used extensively to describe and quantify productivity patterns in the Baltic Sea, as well as to constrain
biogeochemical models for the Baltic Sea. $C_T$ measurements, though the most direct variable for the transport of carbon, have
so far only been continuously monitored on one station in the Central Baltic Sea since 2003 and used to quantify carbon
mineralization in the deep basin during stagnation periods (Schneider and Otto, 2019).

### 2.6.4 Peculiarities of the inorganic carbon system in the Baltic Sea

#### 2.6.4.1 Alkalinity-salinity relations and main different end members

The alkalinity of the Baltic Sea is governed by a complex interplay of different waters entering the Baltic Sea, unusual
contributions to alkalinity complicating carbon system calculations, and long-term trends. The surface alkalinity can be best
described by considering the Central Baltic Sea as a mixing bowl with salinity around 6-7 and receiving water with higher
salinity and alkalinity from the North Sea, freshwater surplus with lower alkalinity from the North and East (i.e. Gulf of Bothnia
and Finland), and freshwater sources with higher alkalinity from the southern part of the drainage basins through the Gulf of
Riga, or runoff from the Odra or Vistula Rivers (Fig. 8) (Beldowski et al., 2010; Hjalmarsson et al., 2008; Kuliński et al., 2014,
2017; Stokowski et al., 2020). The robustness of AT-S relations in the Baltic is also reflecting a lack of calcifying primary
producers in the Baltic Sea east of the Kattegat (Tyrell et al., 2008), mostly suppressing a change of $A_T$ during primary
production and oxic mineralization. The range of $A_T$ in the Baltic from $< 700$ µmol kg$^{-1}$ to $> 3000$ µmol kg$^{-1}$ leads to a large
range of pH under equilibrium with the atmosphere (Gustafsson and Gustafsson 2020; Kuliński et al., 2017, Omstedt et al.,
995   2010).

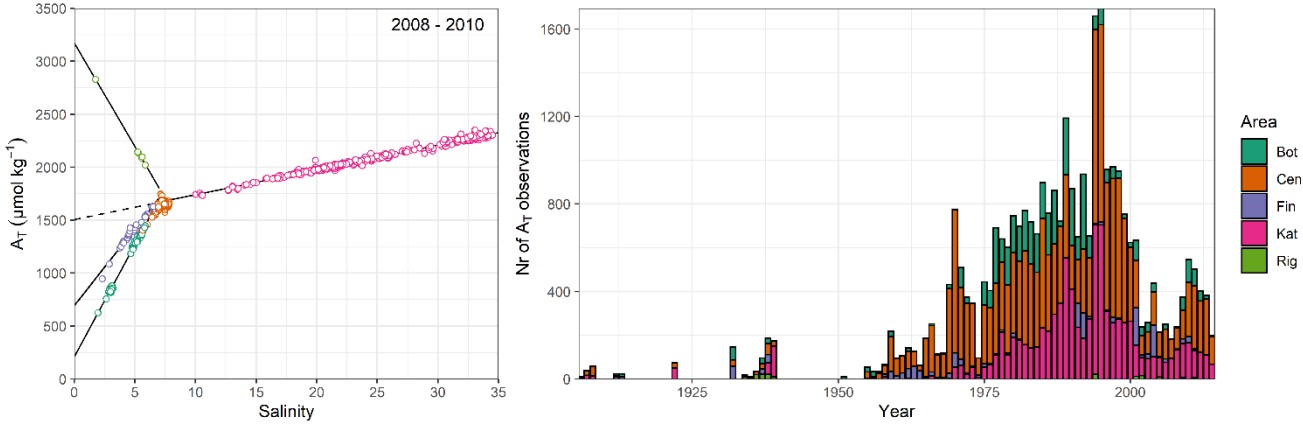

*Figure 8: Different $A_T$ vs. S regimes in the Baltic Sea derived from data from 2008 to 2010, and $A_T$- data density for the
different basins as a function of time (modified after Müller et al., 2016).*

#### 2.6.4.2 Impacts of organic alkalinity and boron anomalies on carbon system calculations



The impact of organic matter, which is far more abundant in Baltic Sea waters than in the open ocean, has been shown to have a non-negligible impact on the internal $A_T$ distribution through changing the contribution of different components to $A_T$, despite not directly changing $A_T$ itself (Kuliński et al., 2014; Ulfsbo et al., 2015). This leads to considerable differences (up to 50 μmol kg$^{-1}$; Hammer et al., 2017) between measured $A_T$ and $A_T$ derived from $C_T$ and either pCO$_2$ or pH, which can lead to significant errors when using $A_T$ in carbon system calculations, in particular in the classical combination with measured $C_T$. Kuliński et

al. (2018) extended earlier data on total boron (TB), derived a Baltic specific TB-S relationship and inferred small errors when calculating pH or pCO$_2$ from $A_T$ and $C_T$, contributing to the uncertainty in the carbon system determination.

**2.6.4.3 Trends in alkalinity over the last decades and potential implications on acidification**

The Baltic Sea receives alkalinity from various sources, including the exchange with the North Sea, riverine alkalinity inputs, and internal alkalinity generation due to organic matter cycling, including anaerobic diagenetic processes in the sediments (e.g.

Gustafsson et al., 2014, 2019a, 2019b). Indications of increasing riverine alkalinity inputs (Hjalmarsson et al., 2008), and an increase of $A_T$ in the central Gotland Sea over the last century (Schneider et al., 2015) were confirmed by a statistical analysis of all available surface alkalinity data (< 20m water depth) until 2015 (Müller et al., 2016). The data suggest a ubiquitous increase in the $A_T$/S-relations in the Baltic Sea between 1995 and 2015, with an increase of 3.4 μmol kg$^{-1}$ yr$^{-1}$ in the central Gotland Basin, and up to 7.4 μmol kg$^{-1}$ yr$^{-1}$ in the Bothnian Bay at a salinity of 3, corresponding to an increase of 70 μmol kg$^{-}$

$^1$ (~5%) or 140 μmol kg$^{-1}$ (~20%) in the two basins over two decades, respectively.

The reasons for this increase in alkalinity are not completely clarified. Müller et al. 2016 suggested a major contribution of weathering-induced external (riverine) input. Sun et al. (2017) support this interpretation by reporting 10-20% increased weathering rates in the pristine northern drainage basin based on a 40-year record of riverine water chemistry data. Gustafsson et al. (2019a, 2019b) report on an overall increase in flow-normalized $A_T$ loads to the Baltic Sea from Swedish rivers by

approximately 21% over the period 1985–2012 (Gustafsson et al., 2019a, 2019b). Gustafsson et al. (2014b), based on model-derived mass balance considerations, suggested that about one-third of the alkalinity sources to the Baltic Sea should originate from internal sources, of which only a small fraction could be accounted for in the model (mainly denitrification), and suggested sulphate reduction with subsequent pyrite formation and silicate weathering as main underestimated sources of alkalinity. Yet, Gustafsson et al. (2019a) estimate that a maximum of 18% of the unaccounted alkalinity can be generated by

pyrite formation, and suggest that both internal and external sources contribute to the unknown alkalinity contribution. The implication of trends in alkalinity on coastal acidification will be discussed below.

For the Northern Basins, the alkalinity of riverine endmembers derived from (a) chemical characterization of riverine waters and (b) extrapolation of $A_T$-S relationships in the basins are in excellent agreement (Gustafsson et al., 2014b, 2019a). A peculiar process leading to a considerable loss of alkalinity has been recently reported for the Odra estuary (Stokowski et al., 2020).

Highly alkaline riverine waters during the productive period reach extreme carbonate oversaturation as a consequence of high biological productivity in the Szczecin Lagoon, which triggers inorganic carbonate precipitation. The authors calculated that





the process led to a reduction of the alkalinity exported to the Baltic Sea of >800 mmol/kg (~30%) during the spring productive phase in the lagoon. This recent work highlights the importance of the Baltic Sea with its carbon system peculiarities as a natural lab for coastal processes.

### 2.6.5 pH-trends in the Baltic Sea on decadal timescales

Ocean acidification, i.e. the decrease of pH in seawater as a consequence of rising atmospheric $CO_2$ levels, also referred to as "the other $CO_2$ problem" is one of the major concerns related to ocean chemistry and its interplay with ecosystem functioning. In the open ocean, OA is predictable (e.g. Doney et al., 2009), where only the rise in atmospheric concentrations and change in surface temperature have to be considered. Trends in pH (and other carbon system parameters) in coastal areas are way more dynamic and less predictable, due to possible changes in the catchment area, including precipitation, weathering, liming, nutrient and organic matter supply (eutrophication), and timing and magnitude of production/respiration patterns occurring on similar time scales (e.g. Carstensen and Duarte, 2019). This is particularly true for stratified water bodies like the Baltic Sea, where production and respiration can be vertically separated, leading to different and potentially contrasting trends in surface and deep waters. Though pH measurements have been performed as part of the Swedish, Finnish and Danish monitoring programs for decades, measurement methodology and frequency would impede detecting the "climatologic" acidification signal caused by increasing atmospheric $CO_2$ levels (Almén et al., 2017; Carstensen and Duarte, 2019), and new methods allowing high precision pH measurements with the required high frequency have only recently become available (Müller et al., 2018a, 2018b; Müller and Rehder, 2018). Yet, clear pH trends on decadal scales have been reported for some of the Danish coastal systems (Carstensen et al., 2018), and the surface waters of most of the major Baltic Sea basins. The weakly buffered (low alkalinity) Northern Basins are amongst the coastal regions with the highest seasonal and interannual pH variability (Carstensen and Duarte, 2019; see Fig. 9). A common feature in the surface waters of all basins is an increase in pH until the early to mid-80[th] as a consequence of eutrophication, which in the central and western basins is followed by a decrease in pH, an expected trend due to the combined effects of oligotrophication and rising atmospheric $CO_2$ levels. In the Bothnian Bay and the Bothnian Sea, however, further increase in pH since the late 90[th] coincides with the reported strong increase in alkalinity (Müller et al., 2016; see above). In a recent sensitivity study, Gustafsson and Gustafsson (2020) used the Baltic Sea Long-Term Large Scale Eutrophication Model (BALTSEM) to address the isolated effect of different drivers on pH in the Baltic Sea basins, including atmospheric $CO_2$ and temperature increase, effects of eutro- and oligotrophication, as well changes in runoff and weathering (i.e. alkalinity input). They conclude that in the long run, the increase in atmospheric $CO_2$ is likely to dominate surface pH trends. An important observation is that although summertime and annual mean surface water pH are enhanced in a eutrophied Baltic Sea with increasing hypoxia, the wintertime pH – and with this, the annual pH minimum – is lower than under lower nutrient loads and better oxygen conditions, as e.g. under the BSAP scenario. It has to be emphasized that in a scenario of climate warming and atm. $CO_2$ increase under effective restoration (i.e. oligotrophication) measures, various pH-reducing drivers work in the same direction, with natural weathering potentially counteracting. Despite large seasonal amplitude in pH and huge differences in the mean annual pH between the different basins due to the large alkalinity





gradients, an observational strategy to allow the determination of pH trends, including the reaction to ecosystem management

actions, is an important task for the future.

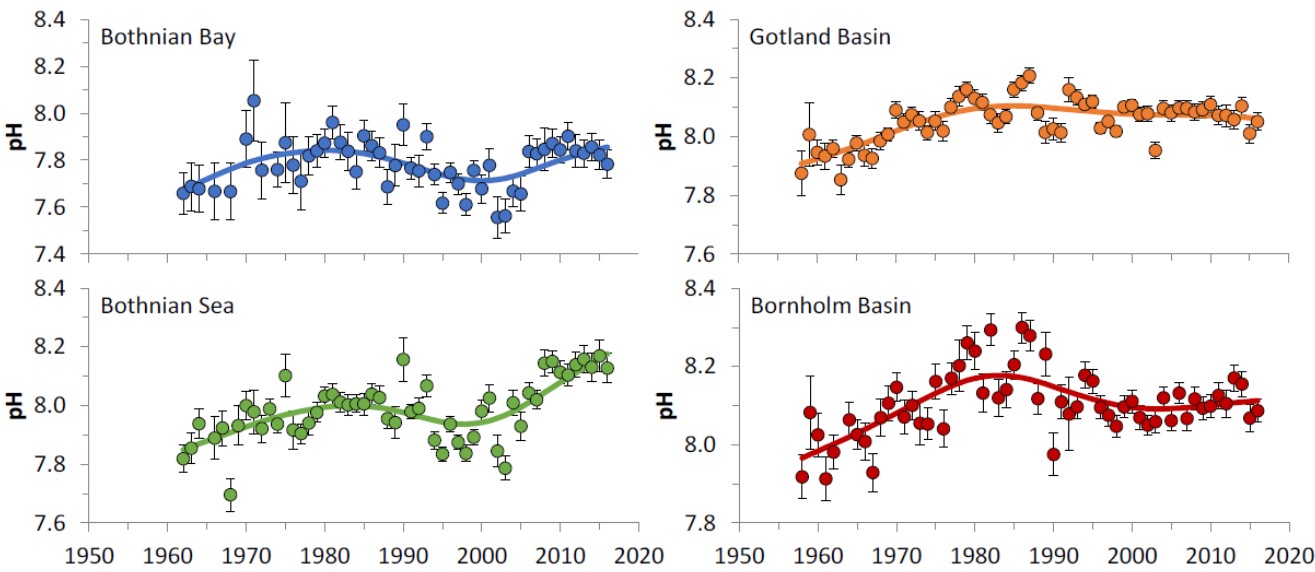

*Figure 9: Trends in pH$_T$ (recalculated from NBS) for four major Baltic Sea Basins, adopted from Carstensen and Duarte, 2019, Suppl. Information.*

**2.6.6 Assessment of biogeochemical processes from observations of the inorganic carbon system**

While loads of bioavailable nitrogen and phosphorus are considered as the most important driver of eutrophication (Section 2.3), the largest part of the biomass formed during primary production and decomposed during remineralization involves the cycling of carbon. Particularly, the link of enhanced organic matter remineralization and oxygen demand is dominated by the oxidation of carbon. Observations of pCO$_2$ in the surface waters, which have been collected in the Baltic Sea with the high

spatiotemporal resolution, have been used to quantitatively address the annual cycle of net community production (NCP), i.e. the balance of primary production and heterotrophic remineralization (Schneider and Müller, 2018). For this approach, the lack of calcifying organisms in the Baltic Sea except for the Kattegat (Tyrell et al., 2008) allows calculating changes in C$_T$ from pCO$_2$ changes and a rough estimate of the alkalinity, which can be derived from regional A$_T$-S relations (Schneider et al., 2009). In the past, surface carbon observations have been used to assess the air-sea exchange of carbon dioxide (Schneider

et al., 2018 & 2014; Wesslander et al., 2010), to quantify nitrogen fixation in the central Baltic Sea (Schneider et al, 2009), and to describe interannual variability in production and remineralization cycles (Schneider et al., 2014 & 2015).

One of the major outcomes of the combined observation of the inorganic carbon and nutrient pools has been the recognition of a high degree of variability in the stoichiometry of nutrient and carbon removal during the summer bloom, but also during parts of the spring bloom. This led to the development of non-Redfieldish parameterizations for organic matter production in



the Baltic Proper (Kuznetsov and Neumann, 2013, Kreus et al. 2015), and recently also the Gulf of Bothnia (Franssner et al., 2019). The proper description of C:N:P ratios during production and vertical transport is a key to link the two main groups of eutrophication indicators for the Baltic Sea: nutrient loads and primary production at the one side, oxygen-debt and hypoxic area at the other.

## 2.7 Role of specific microorganisms in Baltic Sea biogeochemistry

Prokaryotes (Archaea, Bacteria) have a pivotal role in the majority of biogeochemical processes in all ecosystems and similar principles as in other aquatic systems are also relevant for the Baltic Sea. However, some Baltic Sea-specific features deviate from many other coastal and marginal seas and have to be considered for understanding the link between microbial diversity and the biogeochemistry of the Baltic Sea. The most important ones are (1) the reduced circulation in the deeper basins and the resulting hypoxic to sulfidic conditions, (2) the salinity gradient from full marine to nearly limnic conditions and a large

brackish area in the Baltic proper, and (3) constant inflow-borne supply of freshwater and terrestrial bacteria together with terrestrial organic matter.

### 2.7.1 New insights into Baltic Sea microbiomes

The application of molecular techniques to study natural microbial communities has resulted in major new insights into the diversity, biogeography and functionality of Baltic Sea microbial communities within the last decade. High-throughput

sequencing, using the 16S rRNA gene, provided the first comprehensive picture of bacterioplankton diversity along the entire salinity gradient (Herlemann et al., 2011). This approach also revealed, among other findings, strong seasonal changes in bacterial composition in surface waters (Andersson et al., 2009) and pronounced differences between free-living and particle-associated microbial assemblages (Rieck et al 2015). Single-cell genomics of sediment microbes was pioneered in the Baltic Sea and has given insight into the metabolic potential and function of uncultured Archaea (Lloyd et al., 2013) and Bacteria

(Marshall et al., 2017), e.g., in protein degradation in the seafloor, but was also limited by the incompleteness of the single amplified genomes (SAGs).

Metagenomics illuminates the functional potential of complex microbial communities and has been applied in the Baltic Sea, for example, to reconstruct metabolic pathways and functional adaptations along with an oxic-anoxic transition (Thureborn et al., 2013) and along the salinity gradient (Dupont et al., 2014), and to assess the response to environmental perturbations

(Markussen et al., 2018). Metagenomic analyses were also applied to examine the functional potential of specific bacterial groups, revealing their ecological niches in the Baltic Sea (Hugerth et al., 2015). A Baltic Sea Reference Metagenome (BARM) is available for the taxonomic and functional annotation of genes (Alneberg et al., 2018). Nearly complete genomes from metagenomes (MAGS) enabled the physiological analysis of newly discovered prokaryotic groups such as Lokiarchaeota from Baltic Sea sediments (Cáceres et al., 2020), and allowed to predict ecological niches for dominant planktonic prokaryotes

(Alneberg et al., 2020). Metatranscriptomics is a powerful approach to assess the complete set of expressed functions of communities, thereby indicating locations for specific transformations or microbial responses to external factors. It has been



applied, for example, to examine the response to small-scale mixing in the oxic-anoxic transition zone (Beier et al., 2019), and to examine the physiology and performance of cyanobacterial isolates (Teikari et al., 2018).

### 2.7.2 Microbially-mediated transformations in oxygen-deficient waters

The strong stratification in the deep basins of the Baltic Sea, due to a permanent halocline, results in oxygen depletion and euxinic conditions in the bottom waters, only interrupted by major Baltic inflow events which cause ventilation and extensive deepwater renewal (last ones in 1993, 2003, 2014) (Mohrholz et al., 2015). The oxic-anoxic transition zones (pelagic redoxclines) are sites of important transformations within the C-, N-, S-cycles and have been well- studied concerning the prokaryotic key players. Particularly the two deepest basins, the Landsort and the Gotland Deeps, served as model systems for

the study of microbial communities in oxygen-deficient water columns. Overall, there is a high similarity in microbial composition to other marine euxinic systems, but also to oceanic oxygen minimum zones (Jürgens and Taylor, 2018; Wright et al., 2012).

It seems that relatively few taxa control the major inorganic biogeochemical transformations at these pelagic redoxclines. Epsilonproteobacteria of the genus *Sulfurimonas* dominate at the oxic-anoxic interface and in the upper sulfidic zone, are major

contributors to chemoautotrophic production (Jost et al., 2008) and use nitrate to oxidize reduced sulfur compounds (Bruckner et al., 2013; Grote et al., 2012). This process of chemoautotrophic denitrification in the central basins can be considered as a major N loss process for the Baltic Sea, being equivalent to sedimentary denitrification (Dalsgaard et al., 2013). Depending on the actual sulfide concentration, an important side product of this denitrification process is nitrous oxide ($N_2O$) (Dalsgaard et al., 2013). There is a close coupling of nitrification and denitrification around the oxic-anoxic interface. Interestingly, the

process of ammonia oxidation is entirely dominated by Archaea (Thaumarchaeota related to the genus *Nitrosopumilus*) (Labrenz et al., 2010), probably partly due to their adaptation to cope with frequent exposure to sulfidic conditions (Berg et al., 2015). Ammonia-oxidizing bacteria are insignificant here but seem to play a more important role in coastal waters with high nutrient loading (Happel et al., 2018). Nitrite oxidation is conducted by the globally distributed marine nitrite oxidizer phylum Nitrospinae (Beier et al., 2019). Other important chemoautotrophic sulfur oxidizers in the redoxcline are

Gammaproteobacteria of the SUP05 clade (Glaubitz et al., 2013), a group known also from oceanic oxygen minimum zones (Wright et al., 2012).

Methane oxidation in the pelagic redoxcline has been shown to play a dominant role to prevent transport from the methane-enriched deeper anoxic waters to the upper water column and atmosphere. As for other key microbiological processes, methanotrophy is mainly controlled by one single phylotype of aerobic type I methanotrophic bacteria (Schmale et al., 2012,

Jakobs et al., 2013). It has been shown that methane oxidation rates can quickly adapt to increased vertical transport by enhancing microbiological abundance and cell-specific turnover (Jakobs et al., 2014).

Anammox does not seem to be an important process in the redoxcline (Hietanen et al., 2012), which is due to the fact that, unlike in the Black Sea, an extended anoxic and sulfide-free zone, the typical habitat for anammox bacteria, is missing in the



Baltic Sea. Instead, the proximity of the nitrate and sulfide in the vertical profiles favour chemoautotrophic denitrification. An
interesting exception constitutes the situation after MBIs and the re-establishment of anoxia in bottom waters. As observed
after the 2003 inflow event (Hannig et al., 2007), total (reduced and oxidized) manganese concentrations in a compressed
anoxic zone were considerably increased and the enhanced downward flux of oxidized, particulate Mn was probably used for
bacterial $H_2S$ oxidation, thus creating an anoxic sulfide-free zone, suitable for anammox (Hannig et al., 2007). After the MBI
in 2003, the occurrence of anammox bacteria (*Candidatus* Scalindua) and significant anammox rates created a "Black Sea-
like" situation for the Gotland basin (Hannig et al., 2007). Bacteria of the genus *Sulfurimonas* probably also play an important
role in manganese oxide-driven sulfide oxidation, as shown for the Black Sea (Henkel et al., 2019). Inflow events also result
in the temporal displacement of water masses with anoxic microbial communities (and the processes triggered by them) to
shallower water depths (Bergen et al., 2018).

In contrast to the inorganic transformations, the decomposition of organic material in the deeper anoxic basins, where also
DOM composition differs from the oxic layers (Seidel et al., 2017), has been much less studied. Sulfate-reducing bacteria
(mainly Deltaproteobacteria) become a dominating group here (Herlemann et al., 2011), and sulfate reduction is probably a
major organic matter decomposition process. However, neither for this nor for other anaerobic metabolic pathways (e.g.
different fermentations), field data from the central Baltic Sea exist.

**2.7.3 Pelagic microbial communities within the salinity gradient**

Meanwhile, we have a fairly good picture of the salinity-related Baltic Sea bacterial biogeography (Dupont et al. 2014;
Herlemann et al., 2011). At a broad phylogenetic level, the relative abundance of Gamma- and Alphaproteobacteria increases
with salinity, whereas an opposite trend is exhibited by Actinobacteria and Betaproteobacteria. Further, Verrucomicrobia,
mainly represented by one taxon affiliated with the Spartobacteriaceae, dominate large areas of mesohaline waters in the central
Baltic Sea (Herlemann et al., 2011). The reconstruction of the genome of this taxon by metagenomics revealed many genes
potentially involved in the processing of polysaccharides, which are produced by phytoplankton, especially cyanobacteria
(Herlemann et al., 2013). Shifts in bacterial composition with declining salinity also occur on a finer phylogenetic level, e.g.,
within the globally dominating marine clade SAR11 (Herlemann et al., 2014). A challenging question is whether the strong
shift in bacterial composition along the salinity gradient also involves functional changes, e.g., in the decomposition efficiency
of different compounds. Some evidence for this was derived from mesocosm experiments in which bacterial communities
from the Northern Bothnian Sea seemed to be better adapted to utilize river-born terrestrial DOC (tDOC) (Herlemann et al.,
2017), and where DOM degradation differed between different communities (Logue et al. 2016). This is consistent with
substantial remineralization rates of tDOC and $CO_2$ supersaturation measured for this area (Fransner et al., 2019). Overall,
freshwater bacteria can successfully migrate into the brackish Baltic Sea, where they might gain a selective advantage when
riverine DOC is the main carbon source (Kisand et al., 2005; Riemann et al., 2008).

**2.7.4 Role of benthic microbial communities**



Like any sediment microbial community, also Baltic Sea communities exhibit a strong vertical stratification driven by organic carbon availability, redox zonation (Edlund et al., 2008), and environmental filtering (Marshall et al., 2019; Starnawski et al., 2017), where deep sediment communities assemble close to the sediment surface (Petro et al. 2019), at the bottom of the bioturbation zone (Chen et al., 2017). In the Western Baltic Sea, sulfate-reducing microorganisms (SRM) peak in abundance

and richness just below the bioturbation zone, and their numbers strongly decline after sulphate is depleted in the SMTZ (Marshall et al., 2019). When oxygen is absent from the bottom waters, the SRM peak is shifted up to the sediment surface, and also the overall community at the sediment surface differs between sites with permanently oxic (bioturbated), permanently anoxic (not bioturbated), and seasonally hypoxic bottom water. Microbial diversity is highest at oxic and lowest at anoxic sites (Broman et al., 2017; Sinkko et al., 2019), and oxygen availability thus largely controls, via bioturbation, benthic microbial

community structure (Deng et al., 2020). The depth of the SMTZ on the other hand is controlled by the availability of organic matter (and thus sedimentation rate) and the concentrations of sulfate and other electron acceptors (and thus salinity and bottom water conditions). This implies that SRM should peak closer to the surface and decline faster with depth in the oligohaline, hypoxic areas of the Eastern Baltic Sea, for which, however, only few data are available (Reyes et al., 2016; Sinkko et al. 2011). Methanogenesis and anaerobic methane oxidation (AMO) are highest in the SMTZ, where methanogenic

*Methanosarcina* may form syntrophic associations with acetate oxidisers, e.g., *Geobacter* (Rotaru et al., 2018) and ANME-1 Archaea may conduct both methanogenesis and AMO (Beulig et al., 2019); in other sites, ANME-1 and/or ANME-2 Archaea have been implicated in AMO (Myllykangas et al., 2020b; Shubenkova et al., 2010; Treude et al., 2005). For a recent review on biogeochemical processes and microbial life in Baltic Sea sediments, including the deep biosphere, see Jørgensen et al. (2020).

Similar to the water column, salinity exerts an overarching control on benthic bacterial communities, with typical marine and typical freshwater taxa towards the extremes, and a broad overlap at mesohaline conditions (Klier et al. 2018). In coastal sediments, seasonal changes in community composition follow the major inflow of freshwater in spring and phytoplankton sedimentation after the spring bloom (Sinkko et al., 2013; Vetterli et al., 2015).

The important role of chemolithotrophs and metal-cycling microbes seen in the water column also applies to the sediment

communities but with distinct taxa: nitrification (at oxic sites) appears dominated by ammonia-oxidizing bacteria and nitrite oxidizers of the genera *Nitrospina*, *Nitrospira*, and *Nitrobacter* (Reyes et al., 2017) and is closely coupled to sulfide-dependant denitrification and DNRA, e.g., by *Beggiatoa* and *Thiothrix* (Klier et al., 2018; Reyes et al., 2017). The role of epsilonproteobacterial *Sulfurimonas* and *Sulfurovum* remains unclear (Broman et al., 2017). In the absence of internal nitrate production, e.g. in permanently or periodically hypoxic basins, nitrate reduction may be fuelled by settling pelagic diatoms

after phytoplankton blooms, who link pelagic and benthic N cycling by transporting intracellular nitrate to the oxygen-deficient seafloor (Kamp et al., 2018).

A coupling of N and Fe cycles is implicated by the detection of Fe(II)-oxidizing nitrate reducers (Laufer et al., 2016b), tentatively identified as e.g., *Thiobacillus, Hoeflea, Dechloromonas* in Danish fjord sediments (Laufer et al., 2016a, Otte et



al., 2018). Also, microaerophilic Fe/Mn oxidizers (*Mariprofundus, Gallionella*) and phototrophic Fe-oxidizers (e.g.,

*Rhodobacter, Chlorobium*) were found at several sites (Otte et al., 2018; Reyes et al., 2016), and metal-oxidizers were enriched from iron-manganese concretions in the Gulf of Finland (Yli-Hemminki et al., 2014). Fe- and Mn-reduction appears to be coupled to organic carbon content (Laufer et al., 2016a), and members of *Arcobacter, Colwellia* and Oceanospirillaceae were involved in Mn reduction at Mn-oxide-rich marine sites of the Baltic Sea (Vandieken et al., 2012). However, identification of metal-cycling key players in Baltic Sea sediment remains often tentative, when known metal reducers (*Geobacter, Shewanella*)

are rare, while versatile sulfate (and potentially metal-) reducers (Desulfobulbaceae, Desulfuromonadaceae, and Pelobacteraceae) appear associated with zones of metal reduction (Otte et al., 2018; Reyes et al., 2016).

The permanently or seasonally hypoxic sediments of the Baltic Sea are prone to the release of free sulfide. Euxinia can be counteracted by benthic microbial sulfide oxidation as long as a minimum of oxygen (or nitrate) in the bottom water is available. This is seen in the hypoxic transition zone of the Eastern Gotland Basin, which is covered by dense mats of *Beggiatoa*

(Noffke et al., 2016), which have been estimated to consume up to 70% of the sulfide flux towards the sediment/water interface (Yücel et al., 2017). *Beggiatoa* is also common at many other sites of the Baltic Sea (Klier et al., 2018; Reyes et al., 2017). Alternatively, filamentous cable bacteria (originally discovered in the Baltic Sea; Pfeffer et al. 2012) can link the oxidation of sulfide in deeper sediment horizons to the reduction of oxygen or nitrate by conducting electrons over centimetre distances. Cable bacteria belong to the deltaproteobacterial family Desulfobulbaceae, with the marine/brackish genus *Ca*. Electrothrix

and the freshwater/oligohaline *Ca*. Eletronema (Trojan et al., 2016). They are widespread across all salinities in Baltic Sea sediments (Klier et al., 2018; Marzocci et al., 2018; Otte et al., 2018), show the highest densities at seasonally hypoxic sites with high sulfate reduction rates (Hermans et al., 2019), and have been hypothesized to interact with Fe-cycling microbes (Otte et al., 2018). The metabolic activity of cable bacteria does not only remove sulfide but generates a suboxic zone with a pH minimum, which promotes the dissolution of iron sulfide (FeS) and the formation of Fe and Mn oxides (Risgaard-Petersen et

al., 2012). These can act as a "firewall" against euxinia long after the bottom waters have become hypoxic (Sejtaj et al., 2015). Hermans et al. (2019) proposed this cable bacterial firewall to explain why bottom waters in the highly eutrophic Gulf of Finland rarely contain sulfide in summer. The niche partitioning between *Beggiatoa* and cable bacteria, and the extent of their respective euxinia protection, are currently unresolved.

### 2.7.5 Anticipated future development of microbial communities and activities

Since more comprehensive data on microbial communities have been gathered only in recent years and do not yet cover adequately the whole Baltic Sea, it is currently not possible to anticipate how these communities will respond to predicted future environmental changes of the Baltic Sea. From experimental and field studies it is clear that higher sea surface temperature (SST) and increased river runoff will fuel the activity of heterotrophic microorganisms, result in higher carbon remineralisation and shift pelagic food webs towards the microbial components (Wikner and Andersson, 2012). Bacterial

communities respond quickly to changes in environmental factors, with the appearance of well-adapted taxa, probably due to the presence of a large seed bank both in the sediment and the water column. This has been experimentally investigated, for





example, for shifts in salinity (Shen et al., 2018) and different environmental stressors (Markussen et al., 2018). For benthic communities, an increase in areas with hypoxic or anoxic bottom waters may not only lead to declining microbial diversities at the sediment surface but also to faster depletion of sulfate, an upwards migration of the SMTZ, and higher abundances and activities of methanogens.

## 2.8 Interactions between biogeochemical processes and chemical contaminants

Organic contaminants and toxic metals emitted through human activities have become ubiquitous and unwelcome entities in the environment. Their transport and fate are closely associated with biogeochemical cycles, mainly due to the tendency of many substances to sorb to organic matter (Nizzetto et al., 2010). For dissociating organic contaminants and metals, environmental factors such as pH and redox conditions influence speciation and sorption to organic matrices and mineral surfaces (Jones and Tiller, 1999; Pohl and Hennings., 1999). The influence of anthropogenic contaminants on biogeochemical processes is less studied, and many knowledge gaps remain.

### 2.8.1 Catchment characteristics and biogeochemistry impact contaminant transport to the Baltic Sea

*Organic contaminants*. Rivers collect contaminants emitted from point sources (e.g. waste water treatment plants) or deposited on land and subsequently transported via runoff. Hydrological conditions are thereby important for contaminant river transport and retention. Contaminated sediment and soil particles are typically mobilized and eroded during high discharge events (Rügner et al., 2019; Schwientek et al., 2013). Contaminants are also released from the snow-pack during spring floods (Josefsson et al., 2016; Meyer et al., 2011) or mobilized from the soil when snow-melt displaces shallow groundwater (Filipovic et al., 2015). Concentrations of organic contaminants in boreal catchment rivers, and thus the transport to the Baltic Sea, are influenced by vertical gradients of OM and contaminants in soil, and season and land-type dependent pathways for water at different depths in the soil (e.g., discharge from deep soil levels during snow-cover and overland flow in areas with frozen top layer during snowmelt). Another important factor that affects this transport pathway is the quality of suspended OM, which influences the sorption capacity for various compounds and helps explain varying retention of atmospherically deposited contaminants observed in forests and mire-landscapes (Bergknut et al., 2010, 2011; Josefsson et al., 2011, 2016).

Stable flow conditions reduce catchment runoff and therefore transport to rivers, and promote sedimentation of contaminants in rivers and lakes, a process that reduces water concentrations. During low flow conditions, however, concentrations of contaminants emitted from human activities may increase if emissions occur at a constant rate (Urbaniak et al., 2019). Export of contaminants via rivers is counteracted by degradation processes. Field and flume studies of pharmaceuticals in rivers show that biodegradation at the sediment-water interface efficiently eliminates substances due to the relatively long residence time in sediment pore water and diverse microbial community, and is influenced by e.g., sediment composition, redox conditions and DOC concentrations in the water (Lewandowski et al., 2011; Schaper et al., 2018).

Shifts in biological productivity in lakes and marine surface water impact the air-water exchange of many organic contaminants with a mechanism similar to the "biological pump" sequestering $CO_2$ from the atmosphere (Dachs et al., 2002). Hydrophobic





and stable compounds sorb to OM and are transported downwards, thereby potentially depleting contaminant concentrations
in the surface water and enhancing diffusive air-water exchange, as has been observed in the field (Berrojalbiz et al., 2011;
Galbán-Malagón et al., 2012; Josefsson et al., 2011).

*Metals*. Trace metals in freshwater are carried in dissolved, colloidal, or particulate form, mainly in association with OM, Fe-
Mn (oxyhydr)oxides and clay particles. Metal speciation is more important than total concentration with respect to effects on
organisms and export. Metals bound to silicate minerals are generally immobile and non-bioavailable (Tuzen, 2003), while
their ionic species have high mobility and bioavailability (Luoma, 1983; Sunda and Lewis, 1978). Bioavailable trace metals
in rivers can however rapidly precipitate in estuaries after mixing with fairly small amounts of the alkaline brackish seawater
as observed in an estuary in Western Finland, thereby retaining the metals in coastal sediments (Nystrand et al., 2016). Metal
speciation and distribution in the water column are affected by complex interactions like adsorption-, precipitation-,
desorption- or dissolution processes. Among them, redox conditions and formation of metal sulfides play a key role during
stagnant conditions (Pohl and Hennings., 1999). In sediments, metal cycling is driven by microbial oxidation of OM, leading
to the release of complexed/sorbed metals or dissolution of Fe-Mn (oxyhydr)oxides upon changing redox conditions. Part of
the metal-OM complexes is efficiently recycled in the surface sediments during diagenesis, while a considerable fraction is
permanently buried as refractory metal-OM complexes or incorporated into insoluble sulfides, and excluded from biological
processes, making sediments a sink for metals (Jokinen et al., 2020). Several factors such as water flow, changing redox and
oxygen conditions, resuspension of sediment particles, and release from dumped chemical warfare agents (linked to the
presence of anaerobic sulfate reducing bacteria) contribute to returning of metals to the available marine trace metal pool
(Bełdowski et al., 2016; Cybulska et al., 2020). Metals are also entering Baltic Sea bottom waters via submarine groundwater
discharge (SGD) through porous sediments (Szymczycha et al., 2016; Virtasalo et al., 2019). The impact of SGD on the Baltic
Sea ecosystem is still not well understood; the importance of this entry route depends on the physicochemical properties of
groundwater, sediments and bottom water (Jakobsson et al., 2020; Krall et al., 2017; Schlüter et al., 2004; Szymczycha et al.,
2016; Virtasalo et al., 2019).

### 2.8.2 Impact of contaminants on biogeochemical processes

Current research suggests that contaminants together with other stressors such as shifts in nutrient, DOM and oxygen
concentrations contribute to changes in biogeochemical processes. Much knowledge is however lacking, partly because the
chemical mixture in the Baltic Sea is not well characterized (Sobek et al., 2016; Wang et al., 2020), as illustrated in a modelling
study performed on the Kattegatt and the North Sea (Everaert et al., 2015). Exposure to specific polychlorinated biphenyls
(PCBs) and pesticides measured in the seawater did not limit marine phytoplankton growth, but when compensating for the
presence of unknown contaminants in the model, a growth limitation of 10% was estimated for the Kattegatt (Everaert et al.,
2015), which would be a significant impact on the marine carbon cycle.



*Observed effects on microbial communities in the water column.* Exposure to contaminants can change the microbial community composition (Echeveste et al., 2016; van der Meer, 2006) which in turn may affect the functionality of the community (Allison and Martiny, 2008; Shade et al., 2012). In one of the very few studies in the Baltic Sea, effects of exposure to frequently occurring organic contaminants on pelagic bacterial communities were less pronounced than the effects observed due to increased terrestrial DOM concentrations. Still, exposure to contaminants contributed to an overall reduction of bacterial activity and diversity, particularly at elevated terrestrial DOM (Rodriguez et al., 2018). In the Mediterranean (Cerro-Galvez et al., 2019), nutrient conditions were found to be drivers of both microbial growth and extracellular enzymatic activities, but similar to the findings in the Baltic Sea, organic contaminants contributed to the change of both endpoints. Further, a model bacterium culture relevant for the Baltic Sea (*Rheinheimera* sp. BAL341) exposed to a contaminant mixture composed of polycyclic aromatic hydrocarbons (PAHs), alkanes and organophosphate esters had a significant decrease (9-18%) in abundance and production (when in the exponential growth phase) (Karlsson et al., 2019). When exposed to a mixture composed of perfluoroalkylated substances (PFAS), there was no significant effect on growth. Genes that responded to contaminant exposure were involved in several distinct cell functions (Karlsson et al., 2019), suggesting that contaminants may influence microbial community composition and function.

*Effects on biogeochemical processes in sediment.* Field observations from the eastern Gulf of Finland demonstrate the effects of metals and oil contamination on sediment microbial activity (Polyak et al., 2017). Experimental research on sediment from the Baltic Sea showed that in sediments exposed to cadmium (Cd) at environmentally relevant concentrations and varying oxygen conditions including hypoxia, Cd affected microbial denitrification (Broman et al., 2019). In contrast, no effects on nitrification rates or the bacterial ammonia oxidizer gene were found in Baltic Sea sediment contaminated by PAHs and PCBs, while significant effects on the microbenthic community structure were observed (Iburg et al., 2020).

*Indirect effects due to reduced bioturbation.* Organic contaminants, such as PCBs and dichlorodiphenyltrichloroethane (DDT), in sediment, can impact sediment reworking and burrowing of benthic invertebrates (Landrum et al., 2004; Mulsow et al., 2002). Contaminants may however indirectly also affect sediment biogeochemical processing, as was observed in a river system where PAH contamination led to reduced bioturbator activity of the worm *Tubifex tubifex*, which in turn negatively impacted aerobic respiration and denitrification rates in the sediment (Mermillod-Blondin et al., 2013).

**3. Knowledge gaps and future research needs**

Marine research has a long history in the Baltic Sea, and a number of methods and concepts globally applied in marine sciences originate from pioneering work here. The Baltic is among the most heavily investigated seas in the world, which is exemplified, for instance, by the large number of publications referenced in this study, though it is limited in its scope to the core aspects of marine biogeochemistry, and in most parts focuses on the most recent papers presenting the current state of knowledge. Still, there are many knowledge gaps, which limit our understanding of the present-day functioning of the Baltic Sea ecosystem and its possible development in the future.



The future status of the Baltic Sea will be greatly influenced by both socioeconomic and climate drivers. However, there are considerable uncertainties related to the importance of both these drivers and their interactions. As the current downscaling approaches, in particular for socioeconomics, remain poorly constrained, these uncertainties arise in particular from a lack of knowledge of how global drivers will play out at the regional scale. There is a need to develop socioeconomic scenarios that align with current policy trends at both regional and global scales, and to explore how this meets the global sustainability targets (e.g UN SDGs) and the regional environmental targets for the Baltic Sea Basin as defined in the European Union's Marine Strategy Framework Directive (EU MSFD) and the BSAP. Defining the latter, including localized criteria for a good environmental status, also remains challenging. The historical nutrient loads to the Baltic Sea before the 1970s are poorly quantified and uncertain, which limits the understanding of the previous biogeochemical and ecological interactions in the sea and makes it difficult to provide a reliable baseline for modelling its ecological status. To reduce shortcomings associated with this uncertainty, better quantification of the nutrient loads and their geographical distribution is required, combined with a comprehensive assessment of changes occurring in the catchment.

The main biogeochemical processes governing the elemental cycling of C, N, P, and O are generally well known for the Baltic Sea. However, dedicated research is needed to unravel the variations in C:N:P stoichiometry during parts of the spring and the summer blooms, which have been revealed in particular by carbon observations. Furthermore, the environmental and biological factors stimulating certain biogeochemical pathways over others in the complex mosaic of interacting and competing processes in the Baltic Sea are poorly understood. Quantification of the regulatory effects that salinity, temperature, light, availability of oxygen, concentrations of nutrients and trace elements (including pollutants), as well as biological community composition have on rates of important biogeochemical processes is fundamental for proper calibrating coupled hydrodynamic-biogeochemical models, which are the only tools to track the large scale changes in the ecosystem and predict its future development. This refers to the whole Baltic Sea ecosystem, but especially to the highly dynamic coastal zone, acting as a biogeochemical filter for the C, N and P loads from land.

Great knowledge gaps remain also with respect to the role of dissolved organic matter and the variability of its compounds. DOM is still poorly characterized with the measurement techniques available today. They make up 81% of the pelagic nitrogen and 30% of the pelagic phosphorus pools, also fuelling the microbial loop (Savchuk, 2018). This makes DOM an important regulatory factor for many processes in the ecosystem, including primary production. Apart from DOM produced in situ, large loads of organic matter are delivered to the Baltic Sea from land. The fate of this terrigenous fraction of DOM remains, however, poorly understood and recognized. Thus, it is important to understand how the quantity and quality of DOM change across the land-ocean aquatic continuum (LOAC), and how DOM input and transformation in the coastal zone will change in the future.

Additionally, several important knowledge gaps addressing the marine $CO_2$ system (and carbon cycling in general) in the Baltic Sea call for future research actions. The reasons for the observed trends in the alkalinity/salinity relations for the different basins need to be identified, which requires an integrated approach addressing changes in runoff from land, sedimentary





diagenetic processes, and water carbon chemistry. The current inability to close the mass balance for $A_T$, as one of the key biogeochemical parameters and the main variable describing the buffer capacity concerning changes in the acid-base system (i.e. acidification), needs to be overcome. This research need is directly linked to the even larger topic of a consistent carbon inventory for the Baltic Sea. The role of coastal systems in the LOAC as a transformation system and possible sink for carbon entering the marine system from land is currently one of the hot topics of marine biogeochemistry. Yet, an integrated

assessment and mass balance for carbon in the Baltic Sea has not been attempted since the work by Kuliński and Pempkowiak (2011). Without this knowledge, important questions such as the vulnerability to acidification or the current and future role of the Baltic as a source/sink for atmospheric carbon cannot be answered.

There are also significant gaps in our understanding of the oxygen dynamics in the Baltic Sea. Although oxygen deficiency is one of the most important and intensively monitored phenomena in the Baltic Sea, the insufficient amount of data causes that

the sizes of hypoxic and euxinic areas of available map products calculated from measured oxygen and hydrogen sulfide concentration profiles still differ by up to 20% (Meier et al., 2019b). The sources and sinks of oxygen in the deepwater layers still require better identification and quantification. This refers, in particular, to the deficient knowledge about dynamics of small and large saltwater inflows from the North Sea, the role of mixing processes in the ventilation of deep waters and oxygen consumption rates both in water column and sediments.

The lack of data also limits the correct estimation of the role of sediments in the biogeochemical functioning of the Baltic Sea. Organic matter remineralization and burial rates are highly heterogeneous, introducing problems for upscaling the results from a small number of study locations to the whole Baltic Sea or even its sub-basins. Improvements in sediment typology (e.g. Tauber, 2014) and coupled water column–sediment biogeochemical modelling (Radtke et al., 2019) show the way forward towards a more accurate depiction of spatio-temporal variability, but these tools are not yet viable in all locations due to gaps

in data and expertise. A particular problem concerns estimates of process rates, and especially carbon and nutrient burial, in shallow coastal areas of the Baltic Sea. These regions have traditionally been considered to be dominated by sediment erosion and hence omitted from estimates of basin-wide remineralization and burial of organic matter. However, an increasing number of studies has demonstrated the importance of coastal and shallow areas as potential traps of carbon and nutrients (Section 2.2), implying that a subset of these environments should be considered as accumulation zones and are therefore important for

carbon and nutrient cycling. Another important aspect requiring further study is the role of benthic fauna in carbon and nutrient cycling, in particular turnover rates of elements within the faunal biomass, as well as the long-term effects of bioturbation and bioirrigation on rates of microbial processes in a system with changing benthic communities. Also, the role of terrestrial organic matter in diagenetic processes in the Baltic Sea is highly understudied, considering the importance of this material to the total sedimentary carbon and nutrients pools and its variable input rates.

Although recent years brought significant improvements in our understanding of microbial processes, we still lack a complete overview of prokaryotic taxonomic distribution and functionality in the different subsystems of the Baltic Sea. For example, it is not known whether seasonal coastal hypoxia triggers the development of communities comparable to the oxygen-deficient



deep basins. For benthic systems, the database is even weaker, and it is not clear how the different types of sediment biogeochemistry across the Baltic Sea (section 2.5) are reflected in different prokaryotic communities. More importantly, it is currently not understood whether the documented shifts in phylogenetic composition of microbial communities along the main environmental gradients of the Baltic Sea have significant implications for some biogeochemical processes, or whether a high functional redundancy buffers these community shifts. Furthermore, the key players for a couple of important biochemical transformations have not been identified yet, such as methane production and consumption, and metal cycling (Fe and Mn oxidizers and reducers). For some processes (e.g., sulfur oxidation) where pelagic (*Sulfurimonas*, SUP05) and benthic (*Beggiatoa*, cable bacteria) key players are known, niche differentiation and the specific ecological impacts are not sufficiently clear to incorporate the performance of these specific taxa into biogeochemical models. This knowledge gap is due to the lack of representative isolates that can be studied in lab experiments as well as the lack of in situ methods that can quantify the contribution of specific taxa to a particular biogeochemical process.

More knowledge is also needed on the mechanism and importance of interactions between contaminants and biogeochemical processes. It is still unclear how biogeochemical processes affect the transport and degradation of contaminants on their way from catchment to open sea. *Vice versa*, research regarding the quantitative impact of exposure to environmental contaminant mixtures on biogeochemical processes in the Baltic Sea is still in its early stages. One key question is whether exposure to environmental contaminant mixtures makes environmental systems, such as microbial communities, more vulnerable to other environmental stressors, such as climate change or shifts in pH, salinity, redox conditions or oxygen availability.

All the knowledge gaps and research questions addressed above can be directly translated into future research needs. The complexity of the Baltic Sea ecosystem often requires to venture beyond the traditionally understood field of marine biogeochemistry and to pursue a more holistic approach, linking different scientific disciplines. This implies that the multifaceted approaches to be further developed for the Baltic Sea have the potential to lead the way for integrated coastal research in other European and international coastal regions, in line with the historical role of the Baltic Sea for marine research.

## 4. Conclusions and key messages

The Baltic Sea is a complex and highly heterogeneous ecosystem. The specific hydrographic setting (large river runoff, rare and occasional inflows of saltier waters, and permanent water column stratification) as well as strong pressure from external nutrient loads are the main determinants of the ecological status and biogeochemical functioning of the basin. Although measures to reduce nutrient loads to the sea have succeeded in significantly lower nitrogen and phosphorus inputs during the past decades, the Baltic Sea still suffers from eutrophication and hypoxia. In fact, oxygen depletion has recently amplified, and the size of the hypoxic area of >80,000 km$^2$ in 2019 was one of the three largest on record (Hansson et al., 2019). Spreading of hypoxia is observed in the absence of a statistically significant trend in saltwater inflows ventilating the deep water layers in the Baltic Proper, though considerable multi-decadal variability (including stagnation periods) occurs. It appears that the lack of significant improvement in oxygen availability and the slow response time of the system to the reduction in nutrient



loads are mostly due to the vicious circle, a positive feedback mechanism self-supporting eutrophication (section 2.3), and the increase in oxygen consumption observed during the recent decades, especially in the deep water layers. In the range of different developed and explored socio-economic scenarios related to the nutrient loadings, only very few of them, and only those that focus on sustainability with less agricultural land use and targeted technologies for reducing nutrient inputs from land use and wastewater, can comply with the BSAP targets.

The nutrient dynamics in the Baltic Sea is driven by their inputs and modification by biogeochemical processes, mostly organic matter production and remineralization, while nitrogen fixation plays an important role to balance differences in N- and P-availability. The nitrogen and phosphorus loads to the Baltic Sea are shaped by both climate change and socio-economic factors. Modelling shows, however, that changes in societal factors can outweigh the effects of changes in climate. Once in the sea, the nutrient loads undergo significant transformations in the shallow coastal regions. However, coastal ecosystems
around the Baltic Sea are highly diverse in their hydromorphology and physical-chemical conditions as well as the magnitude of received nutrient and organic matter loads, which results in a broad span for the coastal filter efficiency, both today and in the future. Generally, the most important removal process for nitrogen in the coastal zone is denitrification, with the highest rates observed in lagoons and estuaries receiving high inputs of nitrate and organic matter from land. In contrast, phosphorus is removed through permanent burial, but the burial forms have different stabilities that depend on salinity, availability of iron,
and oxygen. Additionally, the composition of dissolved organic matter changes drastically during the passage of the coastal zone through processes of heterotrophic consumption, photochemical degradation, flocculation and burial.

        In addition to eutrophication-driven long-term increase, pelagic nitrogen and phosphorus pools in the Baltic Sea fluctuate with changing redox conditions. Expanding in recent decades, hypoxia causes the nitrogen pool to decline, whereas the phosphorus inventory increases. Furthermore, climate change has also led to important shifts in the seasonality of organic matter
production, with a prolonged phytoplankton growth season, and an earlier start of the spring bloom and a delayed autumn bloom.

        Interestingly, the non-stoichiometric uptake of C, N and P during production has been identified for both the central and northern basins, with strong potential implications on the link between primary production, organic matter export, and deep water oxygen demand. This was possible due to the recent advances in marine $CO_2$ system studies. The latter has also revealed
important changes occurring in the marine acid-base system. The range of pH variability in the Baltic Sea is large, both on temporal and spatial scales, and past and future pH trends are determined by a complex interplay of different drivers. As an example, it was found that the Baltic Sea, with its gradient from the open ocean to freshwater salinity has (at least) two very distinct freshwater total alkalinity endmembers with measurable contributions from unusual components of the acid-base system. Clear trends in alkalinity have been observed over the last 25 years, partially mitigating acidification. In the long run,
however, it is expected that the long-term increase of mean surface $pCO_2$ due to the anthropogenic emissions of carbon dioxide and its rise in the atmosphere, in combination with strong seasonal oscillations in productivity and remineralization with partial separation of both processes in the water column, will play the key roles for acidification trends and extrema.





Overall, about 1% and 4% of the annual nitrogen and phosphorus loads, respectively, accumulate in the Baltic Sea, while the remainder is either exported to the North Sea or lost by biogeochemical processes. Both denitrification and burial remove about 87% of the annual nitrogen inputs, whereas for phosphorus only 69% of the annual load is lost in sediments. Both nitrogen and phosphorus, but also carbon, are deposited in sediments principally as organic matter. However, the organic matter content in the Baltic Sea sediments is highly diverse. Physical redistribution processes lead to the accumulation of fine-grained, OM-rich sediments in deeper areas, whereas the shallow regions are typically characterized by erosion and transport. Still, bathymetry may favour local accumulation in these areas as well. Although autochthonous OM is the dominant source, terrestrial OM may contribute even up to 30%. In surface sediments, OM is intensively recycled. Remineralization returns >90% of the deposited organic carbon to the water column in the form of dissolved inorganic carbon. Both N and P are also released during OM remineralization, but their ultimate fate depends on redox conditions at the sediment-water interface. Both the north-south salinity gradient in the Baltic Sea, as well as the redox gradient between shallow and deep areas, influence relative and absolute rates of sediment biogeochemical processes. These environmental gradients, both in the water column and surface sediments, also strongly shape the microbial community composition, diversity, and function (e.g., sulfate reduction and sulfide oxidation). Biogeochemical processes in the redoxcline of the deep oxygen-deficient basins are carried out by relatively few taxa which control transformations within the carbon, nitrogen and sulphur cycles with ecosystem-wide impact. Deeper sediment microbial communities, including those driving important processes like sulfate reduction and methanogenesis, assemble close to the sediment surface (just below the bioturbation zone) by environmental filtering of the founding surface communities.

Although it is well known that the Baltic Sea is under high anthropogenic pressure, the role of contaminants in shaping biogeochemical processes (and vice-versa) remains still highly unclear. The contaminant export from land to sea is heavily influenced by organic matter cycling due to the tendency of many potentially harmful substances to sorb to OM. The range of environmental conditions accompanying biogeochemical processes has a large impact on the speciation of metals, which in turn affects their bioavailability (and thus also toxicity). Exposure to contaminants at environmental concentrations can also contribute to shifts in e.g. microbial community composition and enzyme activity; these mechanisms, however, remain poorly understood so far.

Both eutrophication-related research and chemical integrity and contaminant research will more and more need to address the multiple drivers resulting from anthropogenic and climate forcing, and controlling processes in an interdisciplinary way. Simultaneous changes, including, but not limited to, temperature, salinity, redox conditions, pH, nutrient loads and contaminant burden, do not only alter rates of biogeochemical reactions, but might induce further changes also on the prevailing key players of (micro)-biological controlled processes. Thus, Baltic Sea biogeochemistry will more and more turn into a field of multi stressor (or multiple driver) research (see e.g. Reckermann et al., 2021, this volume).



**Author contribution**

The manuscript is a result of a joint effort of all co-authors and its final wording is an effect of numerous iterations. More specifically: introduction was drafted by K.K. and G.H., section 2.1. by B.G., J.E.O., A.B., C.H., O.S. and M.S., section 2.2 by J.C. and E.A., section 2.3 by B.M-K., O.S. and B.G. by section 2.4 by H.E.M.M., M.N. and J.C., 2.5 by T.J., P.O.J.H., and C.P.S., section 2.6 by G.R. and K.K., section 2.7 by K.J. and A.Sch., section 2.8 by B.S., E.U. and A.Sob., sections 3 and 4 are the joint effort of all the co-authors. The final manuscript has been compiled by K.K. and G.R with contribution of all co-authors.

**Competing interest**

All the authors declare that they have no conflict of interest.

**Acknowledgements**

The knowledge assessed and summarized in this study was acquired within the Baltic Earth program (Earth System Science for the Baltic Sea region, see http://www.baltic.earth) with significant support from the BalticAPP, BLUEPRINT, COCOA, INTEGRAL, MIRACLE and SOILS2SEA projects funded by BONUS (Art 185) jointly from the European Union's Seventh Framework Programme for research, technological development and demonstration and from the Innovation Fund Denmark, the Swedish Environmental Protection Agency (Naturvådsverket), the Polish National Centre for Research and Development, the German Federal Ministry of Education and Research (BMBF), the Russian Foundation for Basic Research (RFBR), the Latvian Ministry of Education and Science, the Research Council of Lithuania, the Estonian Research Council, the Academy of Finland, and the Swedish Research Council for Environment, Agricultural Sciences and Spatial Planning (FORMAS). Additionally, the contribution of the following researchers in this project has been supported by:

K.K. – the Polish National Science Centre (grants no. 2015/19/B/ST10/02120 and 2019/34/E/ST10/00167) and IO PAN statutory activity II.7,

G.R. – Integrated Carbon Observation Project (ICOS, funded by BMBF) and the project SPECTROPHABS, funded by the German Bundesamt für Seeschiffahrt und Hydrographie (BSH),

B.G., B.M.-K., E.U. and O.S. – The Swedish Agency for Marine and Water Management through the grant 1:11 - Measures for marine and water environment,

T.J. – the Academy of Finland, grants 317684 and 319956,

P.O.J.H. – the Swedish Research Council (VR, grant no. 2015-03717) and the Swedish Agency for Marine and Water Management,

A.S. – the Danish National Research Foundation (DNRF104 and DNRF136),

B.S. - the Norway Grants 2014-2021 operated by National Science Centre under Project Contract 2019/34/H/ST10/00645 and grant no. 2019/34/E/ST10/00217 funded by the Polish National Science Centre.





## Glossary

| | | |
|---|---|---|
| | AB | Arkona Basin |
| | AMO | anaerobic methane oxidation |
| 1540 | AT | alkalinity |
| | BACC I | Assessment of Climate Change for the Baltic Sea Basin |
| | BACC II | Second Assessment of Climate Change for the Baltic Sea Basin |
| | BALTEX | The Baltic Sea Experiment |
| | BALTSEM | Baltic Sea Long-Term Large Scale Eutrophication Model |
| 1545 | BARM | Baltic Sea Reference Metagenome |
| | BEAR | Baltic Earth Assessment Report |
| | BD | Bornholm Deep |
| | BONUS | funding mechanism for the Baltic Sea region, www.bonusportal.org |
| | BP | Baltic Proper |
| 1550 | BS | Baltic Sea |
| | BSAP | Baltic Sea Action Plan |
| | BSB | Baltic Sea sub-basins |
| | BSDB | Baltic Sea Drainage Basin |
| | $^{137}Cs$ | caesium isotope |
| 1555 | C | carbon |
| | Ca | calcium |
| | $CaCO_3$ | calcium carbonate |
| | Cd | cadmium |
| | CDOM | colored dissolved organic matter |
| 1560 | CFA | carbonate fluorapatite |
| | CMIP5 | Coupled Model Intercomparison Project |
| | $CO_2$ | carbon dioxide |
| | $CO_2^*$ | sum of dissolved $CO_2$ and undissociated carbonic acid |
| | $CO_3^{2-}$ | carbonate ion |
| 1565 | $C_{org}$ | organic carbon |
| | $C_{org-T}$ | terrestrial organic matter |
| | CSIM | Community Sea Ice Model |
| | CT | dissolved inorganic carbon |
| | CTD | Conductivity Temperature Depth measuring device |
| 1570 | DDT | Dichlorodiphenyltrichloroethane |
| | DIC | dissolved inorganic carbon |
| | DIN | dissolved inorganic nitrogen |
| | DIP | dissolved inorganic phosphorus |
| | DNRA | dissimilatory nitrate reduction to ammonium |
| 1575 | DOC | dissolved organic carbon |
| | DOM | dissolved organic matter |
| | DON | dissolved organic nitrogen |
| | DOP | dissolved organic phosphorus |
| | EA | electron acceptors |
| 1580 | EU | European Union |





| | | |
|---|---|---|
| | E-HYPE | hydrological prediction of the environment Model |
| | EMEP | source-receptor matrices |
| | Fe | iron |
| | FeS | iron sulfide |
| 1585 | $fCO_2$ | fugacity of carbon dioxide |
| | FREF | fast repetition rate flourometry |
| | GCM | global climate model |
| | GD | Gotland Deep |
| | $HCO_3^-$ | bicarbonate ion |
| 1590 | ICES | International council for the exploration of the sea |
| | IPCC | Intergovernmental Panel on Climate Change |
| | HELCOM | Baltic Marine Environment Protection Commission |
| | HELCOM-PLC | HELCOM pollution load compilation |
| | $H_2S$ | hydrogen sulfide |
| 1595 | LOAC | land-ocean aquatic continuum |
| | MAG | Metagenome-assembled genomes |
| | MBI | Major Baltic Inflow |
| | Mn | manganese |
| | MSFD | Marine Strategy Framework Directive |
| 1600 | N | nitrogen |
| | $N_2O$ | nitrous oxide |
| | NCP | net community production |
| | NECA | Nitrogen Emission Control Area |
| | NHx | sum of ammonia and ammonium |
| 1605 | NOx | nitrogen oxides |
| | O | oxygen |
| | OA | ocean acidification |
| | OM | organic matter |
| | P | phosphorus |
| 1610 | PAH | polycyclic aromatic hydrocarbon |
| | PCB | polychlorinated biphenyl |
| | $pCO_2$ | partial pressure of carbon dioxide |
| | PFAS | perfluoroalkylated substances |
| | pH | negative decadic logarithm of protons |
| 1615 | $pH_T$ | pH expressed in the total scale |
| | POM | particular organic matter |
| | $P_{org}$ | organic phosphorus |
| | RCM | regional climate model |
| | RCP | Representative Concentration Pathways |
| 1620 | S | salinity gradient |
| | S | sulfur |
| | SAG | single amplified genomes |
| | SDG | sustainability development goals |
| | SGD | submarine groundwater discharge |





| 1625 | Si | silicon |
| | SMTZ | sulfate-methane transition zone |
| | SRM | sulfate-reducing microorganisms |
| | SSP | Shared Socioeconomic Pathways |
| | SSP1 | Shared Socioeconomic Pathways (sustainable development) |
| 1630 | SSP5 | Shared Socioeconomic Pathways (fossil fuel development) |
| | SST | sea surfaces temperature |
| | SWIM | Soil and Water Integrated Model |
| | TB | total boron |
| | tDOC | terrestrial dissolved organic carbon |
| 1635 | TDS | total dissolved solids |
| | TIC | total inorganic carbon |
| | TN | total nitrogen |
| | TOC | total organic carbon |
| | TP | total phosphorus |
| 1640 | UN | United Nations |
| | VEMALA | finish national nutrient load model |
| | WFD | Water Framework Directive |

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
