# Peer review of "Biogeochemical functioning of the Baltic Sea"

_Earth System Dynamics, 2021_

## Author Comment (AC1)

Earth Syst. Dynam. Discuss., referee comment RC1
https://doi.org/10.5194/esd-2021-33-RC1, 2021
**Comment on esd-2021-33**

Donald Boesch (Referee)

Referee comment on "Baltic Earth Assessment Report on the biogeochemistry of the Baltic
Sea" by Karol Kuliński et al., Earth Syst. Dynam. Discuss.,
https://doi.org/10.5194/esd-2021-33-RC1, 2021

Responses to Reviewer#1, Donald Boesch, (in blue)

**General Comments**

This is an excellent assessment on the current state of biogeochemistry of the Baltic Sea.
It is comprehensive, thorough, up-to-date, and effectively synthetic. The assessment
considers past, present and future biogeochemical dynamics and how they are affected
by human activities and the changing climate. It will be a standard, stock-taking
reference for future research and science applications in the Baltic Sea, but will also be of
significant interest outside of the region because of the diversity of environmental
conditions and biogeochemical processes in the Baltic.

Thank you. We appreciate the thorough review and suggested changes.

**Specific Comments**

I have no substantive criticisms of the sections summarizing and synthesizing the current
state of knowledge. They are uniformly thorough and sound. The section of knowledge
gaps and future research could be sharpened if it is meant to be useful beyond a wish list
of scientists. This section is replete with characterizations of poorly quantified and
uncertain, poorly understood, poorly characterized, great knowledge gaps, hot topics,
significant gaps, correct estimate, not understood, and not sufficiently clear. But, do
these unknowns and uncertainties matter equally, either in achieving robust
understanding or for practical utility in sustainable development in the face of climate
change. Some level of prioritization as to the most critical knowledge gaps and research
opportunities to narrow them would be helpful.

We agree with this opinion. We will prioritize the knowledge gaps and research needs in
the revised version.

All scientists want more data and the next critical experiments and refined models, but
what difference will these make. For example, i is not at all clear how better
quantification of the nutrient loads and their geographical distribution and changes in the
catchment can better quantify baseline conditions before 1970 (lines 1349-1353). Is a
baseline even a relevant concept for characterizing good ecological status when, because
of changing climatic and other conditions, it is no longer an achievable state?

It is indeed true that we will not return to the past and we agree that the statement does
not make proper sense in its current formulation. However, the past is de facto used to
establish targets/thresholds on indicators of eutrophication, but most importantly, by
validation of models for periods long enough to encompass the transition from oligotrophic
to eutrophic is a prerequisite for gaining any confidence in projections into a less eutrophic
future. We will alter the paragraph in the manuscript to reflect these aspects.

**Technical Corrections**

Perhaps give a more complete name for BONUS such as BONUS: Science for a Better
Future of the Baltic Sea Region (line 49).

We will correct that.

It would be helpful, for context, to mention the other Grand Challenges that Baltic Earth is dealing with (line 60).

We will do that.

A reference is required here to support the statement about the decrease in nutrient loads and briefly some indication of the proportional amount of the decrease would be helpful. I realize this is discussed in more detail later, but the statement needs some support on first mention (lines 99-100).

We will add the reference Gustafsson et al., (2012)

Misplaced, open parenthesis (line 150).

We will correct that.

The RCPs are pathways greenhouse gas concentrations and their radiative forcing, not global warming, per se. (line 158) The authors should make clear that the CMIP5 projections are those used in the IPCC AR5. Updated projections will be included in AR6 that will be released this year.

We will clarify that.

Clarify the sentence: "The results generally show a greater variation among climate models . . . for projections until the mid-21st century, but greater variation among RCPs towards the end of this century". (lines 161-164)

We will rephrase that to show that the variation concerns projected temperature and precipitation changes.

Make clear that these projections are for air temperature. (lines 165-167)

We will correct this sentence to make it clear that the temperature projections concern air temperature.

Are the SSPs for climate derived from CMIP6 model ensembles? (lines 172-173)

In the IPCC methodology, SSPs are different from the RCPs, although there are clear links between which SSPs are consistent with which RCPs. The CMIP6 model ensembles do no directly specify SSPs, but align most of the simulation results to the relevant RCPs (and to some extent SSPs). The SSPs mentioned here were from adapted for the Baltic Sea Region from the overall concept of the SSPs, with consideration of which SSPs fit with relevant RCPs.

Doesn't all carbon enter the Baltic Sea in either inorganic or organic form? (line 221)

Yes. We will rephrase that sentence.

Is this TOC increase for the northern Baltic or an average for the entire Baltic? (line 228)

This refers to the northern Baltic only. We will clarify that.

The caption for Figure 2 should explain the color coding in the histograms. (lines 344-345).

We will add this information to the Fig. 2 caption.

Clarify what is meant by "71% (89%) of the phytoplankton nitrogen and phosphorus uptake." (line 520)

We will clarify that. In principle, we wanted to underline in this sentence the importance of dissolved organic matter in N and P cycling. The numbers refer to the shares of N and P pools used for primary production, which go through the dissolved organic matter pool first.

Misplaced, open parenthesis (line 728).

We will correct that.

Do mid-80th and late 90th refer to the mid 1980s and late 1990s, respectively? (lines 1052 and 1054)

Yes. We will correct that.

The three long sentences on lines 1059-1066 are confusing and should be more clearly stated.

We will rephrase this section.

Non-stoichiometric uptake may not be clear to the non-specialist reader. I suggest that this should be reworded more plainly. (line 1461)

We will rephrase that and explain the meaning of the non-stoichiometric uptake.

---

## Author Comment (AC2)

Earth Syst. Dynam. Discuss., referee comment RC2
https://doi.org/10.5194/esd-2021-33-RC2, 2021
**Comment on esd-2021-33**

Anonymous Referee #2

Referee comment on "Baltic Earth Assessment Report on the biogeochemistry of the Baltic
Sea" by Karol Kuliński et al., Earth Syst. Dynam. Discuss.,
https://doi.org/10.5194/esd-2021-33-RC2, 2021

Review of Baltic Earth Assessment Report on the biogeochemistry of the Baltic Sea
by K. Kulinsky et al.

Responses to Reviewer#2 (in blue)

The authors wrote a long report on the functioning and evolution of several major Baltic
Sea biogeochemical cycles, in particular of water, nutrients (including primary
production/eutrophication and oxygen), carbon (mainly focusing on inorganic carbon), and
some pollutants. Also included is an overview of advances in research on microorganisms
that are involved in and mediate some of the cycles. The author team aims to synthesise
published knowledge in analogy to the IPCC reporting process, in line with past publications
(BACC Reports 2008 and 2015) mandated by BALTEX and its successor, Baltic Earth, on
climate and environmental change in this unusual and interesting shelf sea. The stated aim
is to represent new knowledge that emerged since the last reports.

Thank you. We appreciate the thorough review and suggested changes.
Indeed, we tried to focus in our study mostly on the new knowledge in the field, which
have been published after BACC II (2015) report. However, we did not limit our study only
to this time frame as the aim was to review and assess comprehensively published reports
to present the current state of knowledge in the field.

This iteration of the manuscript review process for a dedicated Special Issue of the journal
Earth System Dynamics apparently follows a first review that asked for minor revisions.
On that background, the manuscript is in a state that can be published. The effort is
ambitious and most of the contents indeed provide some novel insights from observations
and numerical modeling. The paper is well written, and although there are inconsistencies
in structure and imbalances in thematic lines, there are no real corrections that need to be
made.

Thank you for this opinion.

The manuscript being new to me, on the other hand, some general and specific statements
may be in order – perhaps as challenges for a future status report.

Thank you. As in the following quite comprehensive and detailed suggestions for change
of the manuscript, some of which result from a different perception of the scope of the
paper from the reviewer's and the authors' side, here we present some general statement
on how we took the reviewer's suggestion into account, balancing between the statement
that the paper is in a state that could be published (above) and fundamental structural
changes suggested (for future work) below:

- we tried to incorporate all suggestions to improve clarity in individual sentences and
statements
- we will consider to incorporate an IPCC-like judgment on the level of consensus in our
future work, but not in this review. We have to keep in mind that the paper is a natural
scientific review, and has an "audience" and impact quite different from an IPCC report.

Still, a debate on our level of knowledge on certain aspects is an interesting approach for the future.
- we did not refrain from referring also to work published before the BACC II report, as the scope of the paper was to give a review of our state of knowledge on the Baltic Sea Biogeochemistry as a stand-alone paper. As such, it is important to integrate basic findings from the pre-BACC II era, as well as some fundamental descriptions of biogeochemical or element cycling.
- in that regard, the reviewer noted the referencing of knowledge on some sub-chapters as "old", though the large part of the references is after the reporting period of BACC II. In a more general way, a look on the reference list supports that the authors put a strong – though not exclusive – focus on references published after the reference frame of BACC II, i.e. 2013.

This is a consensus report and it is difficult to argue when many eminent researchers and co-authors agree on issues. What I missed is a representation of diverging opinions – what is the point of citing the IPCC process when their method of expressing consensus is not adopted? How certain are authors that changes can be attributed to specific drivers? This in particular is the case in the long sections on nutrients, eutrophication and oxygen deficiency, where according to my knowledge there are diverging opinions that in part may be caused by authors´ different working areas and processes pertinent there. Have rigorous attribution exercises been performed?

Although we cite the IPCC reports in our study, we do not refer directly to the IPCC reporting process itself. Nevertheless, when preparing our report we paid a lot of attention to maintaining objectivity in our work. We based on already published knowledge only, which has been carefully reviewed and the entire content of the manuscript has been agreed in the large group of co-authors during multiple iterations.
The great basis for the sections on nutrient loads, eutrophication and oxygen deficiency are datasets obtained in the monitoring programs, which describe past and present ecological status of the Baltic Sea ecosystem and do not leave space for over- or mis-interpretation. However, for attributing the changes to specific processes/mechanisms and for describing potential future changes we have presented diverging opinions whenever it was necessary and tried to select appropriate phrases in the manuscript to avoid misinterpretation.

In many instances the Baltic Sea biogeochemical cycles are described in their basic functioning (which often are not different from other marine and coastal areas). The claim is to update information provided by previous BACC reports and syntheses, but that intended novelty is often swamped by general statements and descriptions in several of the chapters. For example, there is a substantial amount of text dedicated to nutrients and eutrophication effects, including oxygen deficiency – this offers little in terms of new data/ideas. In line with this impression is the enormous list of references, which for the most part have been published before the reporting period. The authors should have been more rigorous in presenting advances that emerged over the last years!

The main goal of our study was to present the current state of knowledge about the biogeochemical functioning of the Baltic Sea including the most recent (after BACC II) developments in the field. Since BACC II the approach in Baltic Earth and in the large group of scientists contributing to Baltic Earth Assessment Reports (BEARs) has changed. Earlier (in BACC I and BACC II) the individual topics were presented as chapters, which in the end comprised a book. Now, the thematic reports are going to be presented as the individual review papers. This approach requires that each individual topic will be presented as a self-standing publication. This, in our opinion, cannot be done fragmentary (for instance focusing only on the most recent developments), but it requires to present the context in a broader way, sometimes referring to general knowledge and important developments made not necessarily in the most recent past. We believe we managed to balance all these needs in our manuscript, which in the end forms a comprehensive assessment presenting

the current state of knowledge in the field. Trying to keep the manuscript in the format of a self-standing publication understandable for a broader community (also from outside the Baltic Sea and from outside the field of marine biogeochemistry), we still made a thorough assessment of the most recent studies and reviewed in total about 260 publications and reports which have been published after BACC II was concluded.

We cannot share the reviewer's opinion that our manuscript "offers little in terms of new data/ideas" as exemplified by the Reviewer with topics of nutrient cycling, eutrophication and oxygen deficiency. Each of these sections reports most recent data and/or findings. For details, please see for instance Fig. 2, 4, 5, 6 and the corresponding text. All of them present the most recent developments and data and refer to the recently published reports and/or datasets.

Although the structure of individual chapters (basic principles, past, present, what do models say about the future) makes sense, the outlook to future changes often is speculative (signaled by choice of words such as "would be, may be, possibly etc."). How certain are you (see above)? Furthermore, the structure is not rigorously maintained throughout the paper. Some of these projections are based on models of varying complexity, but they apparently have not been rigorously evaluated against each other, although it appears that results of different models differ. Many of the outlooks thus are conjecture or reflect opinions.

We agree with the Reviewer that some text passages related to the future projections sound speculatively. We do not see this, however, as a fault. In our report we have thoroughly reviewed available and published knowledge in this respect. Our goal was not to judge which model is better or aim for a intercomparison, but to demonstrate the spectrum of the projections published in the peer-reviewed papers. In cases where we found consensus, we have used unambiguous wording. However, when the model outcomes were diverging we have intentionally used phrases that indicate uncertainty.

The microbiology part is essentially a status report on the state of research on "omics", which may be timely. The chapter on pollutants on the other hand does not offer any new information and treats "traditional" pollutants only, although there have been reports on some of the large spread of novel pollutants in the Baltic Sea.

We agree that specific chemical contaminants mentioned in this section manuscript encompasses the well-studied classical/traditional/legacy pollutants, and it is certainly true that many novel so called "chemicals of emerging concern" have been observed and studied in the Baltic Sea. However, it is not our ambition here to provide an overview of the current status of the Baltic Sea with respect to presence and levels of chemical contaminants nor ecotoxicological effects. Since the aim of this BEAR report is to describe the available knowledge (both old and new) on the biogeochemical functioning of the Baltic Sea, we focus specifically on the interactions between biogeochemical processes and chemical contaminants. These relationships have previously been, and are still, largely studied by analysing traditional contaminants with a high detection frequency in the environment, well-known properties and behaviour in the environment. We introduce fundamental interaction mechanisms using key-publications irrespective of study area, and review studies conducted in the Baltic Sea region that demonstrate their functioning in this specific region. We admit that due to the limited space dedicated to chemical contaminants in this broad review paper, the descriptions of the different studies are brief, however provide an overview of what type of contaminants – biogeochemical process interactions that have been studied in this sea.

To guide the reader looking for recent reviews of current knowledge regarding contaminant levels and effects in the Baltic Sea, we will add a sentence with references in the beginning of the section "2.8 Interactions between biogeochemical processes and chemical contaminants"

The following are some further comments and suggestions keyed to line numbers:
64: Marine biogeochemistry ... deals with the transport. It is not really a new discipline.

We will rephrase this sentence.

65: ..particular C, N, P, Si....

We will correct that.

68: delete On top of that,

We will correct that.

69: with its periodicity?

We had in mind for instance natural seasonal variability
We will rephrase that.

79: wording and comparative: drained by many smaller rivers, is not that densely populated

We will rephrase that.

103: comment: There appears to be controversy on the role of reactive nitrogen, which is not really addressed in the manuscript

We will address this in the revised manuscript.

111 ff: Further down you state that the present extent of anoxia is the maximum possible extent.

This is correct. This paragraph refers to the knowledge presented in BACC II. The modelling studies cited there show that keeping nutrient loads unchanged (business-as-usual scenario) will significantly increase the anoxic and hypoxic areas. While the implementation of the BSAP has the potential to decrease the extent of hypoxic and anoxic areas despite the counteracting influence of climate change. Later on in the manuscript we discuss these findings against recent observations (e.g. Hansson et al., 2019), which showed that the size of the hypoxic area in 2019 was one of the three largest on record. Based on that we concluded that the lack of significant improvement in oxygen availability and the slow response time of the system to the reduction in nutrient loads are mostly due to the vicious circle, a positive feedback mechanism self-supporting eutrophication (section 2.3), and the increase in oxygen consumption observed during the recent decades, especially in the deep water layers.

118 ff: comment: Here you state the objective of reporting progress since the last assessment. The following text is a general description of biogeochemical processes in the Baltic, which makes for a very long text.

This paragraph presents not only the goal of our study but also its scope. For such a broad review we see it is necessary to frame the extent of our review and communicate it to the reader already in the introduction.

Importantly, our study does not focus only on reviewing the most recent works published after BACC II, which is claimed by the Reviewer. We clearly define the goal of our study in the manuscript (lines: 118-121):
*"Since the work on the last assessment (BACC II, 2015) was carried out, intensive research on the biogeochemical cycling in the Baltic Sea has been conducted, including studies on*

*past, present and future changes. This paper not only summarizes the results of these recent studies but comprehensively assesses currently available, published knowledge on the biogeochemical functioning of the Baltic Sea, while pointing out knowledge gaps and future research needs."*

137: This either is the case for any marine ecosystem, or needs to be extended to include pronounced internal cycling.

In this chapter, as the title informs, we focus only on changes in the external drivers. The internal cycling of nutrients is presented in the subsequent chapters: 2.2, 2.3, 2.4 and 2.5.

155 ff. This is a background on RCPs and some (regionalised) application (in terms of nutrient inputs) for the Baltic Sea and the likelyhood that targets of the Baltic Sea Action Plan will be attained under the assumptions made (from line 170). This should be put into the chapter dealing with probable and plausible nutrient inputs. The RCP´s are a way to prescribe model boundary conditions – I would have expected that the results of model projections thus forced would be treated in all subchapters on future developments.

This section 2.1.1 deals largely with the methodologies that have been used to derive the future projections (scenarios) of environmental (climate and nutrient) drivers for the state of the Baltic Sea. This methodology includes the use of SSPs (described from lines 170-179). In lines 181-183 we mention that other methodologies have also been used. The only result that we mention here is in lines 179-181, and this is only to exemplify differences between SSPs. We have taken the approach to have a section on methodologies in section 2.1.1 to avoid repeating this in the following sections and subchapters.

185: playing - river flow or matter? This refers only to model projections – are there any observation?

We had in mind that riverine input of matter plays an important role in shaping biogeochemical conditions in the Baltic Sea. We will rephrase that sentence.
In this chapter we discuss hydrological regime only. The importance of riverine input is exemplified for instance in Fig. 2 (and discussion around it) which in large extent is based on observations.

194: How certain is this? Anything can potentially happen.

We intentionally used here a wording that implies uncertainty. This is due to the ambiguous projections related to future changes in salinity.
For details about salinity changes please see the following BEARs:
- Salinity dynamics of the Baltic Sea by Lehmann et al. (https://esd.copernicus.org/preprints/esd-2021-15/)
- Oceanographic regional climate projections for the Baltic Sea until 2100 by Meier et al. (https://esd.copernicus.org/preprints/esd-2021-68/)

201: Is this observation-based? The preceding text is on models only, isn´t it?

Yes, this is observation-based. For details please see the report by Klavins et al. (2009) which we refer to in our manuscript.

210: What does however mean here?

Linguistic mistake. We will correct that.

213: This phrase is difficult to understand

We agree and we will rephrase that.

222: Further down you say that 30% of sedimentary TOC is land-derived - does that match this statement? What is the nature of the positive feedback? Higher CO2 = more terrestrial TOC = more respiration in the Baltic?

We do not see these two things contradicting. Here we say that total organic carbon (sum of particulate and dissolved) constitutes about 37% of total carbon load entering the Baltic Sea from land and that this terrestrial organic carbon is to large extent respired in the marine ecosystem. Based on that only, it is impossible to judge if terrestrial carbon may (or may not) constitute 30% of sedimentary OC.

The word 'feedback' was misused here. We had in mind that respiration of terrestrial organic carbon in the sea influences in the end atmospheric CO2. We will rephrase that.

231: What is the role of increasing pH here? pH of river water or of rain?

Higher pH in the catchment increase the solubility of DOC and thus can increase export. A clarification will be added to the text.

232: What does however mean here?

Nothing. It will be removed and the sentence corrected.

239: As stated before, the manuscript would benefit from a more stringent structure of a) observations and b) modeling scenarios

We prefer to not differentiate between both these data sources. In our opinion they both are complementary and discussing them at once is an adding value to the manuscript.

249: What does even mean here?

We will rephrase that.

257: What I get from the entire paragraph is that climate and human effects are important, but that regional differences exist. Is this surprising or new?

In our opinion this paragraph provides more than stated above. It shows the agreement between different studies that the socioeconomic factors play a significant role for shaping nutrient loads and may in some cases outweigh or even reverse the climate impacts which indicates the potential for effective mitigation strategies in the region.

283: lowering pace?

It will be reformulated to "despite that emissions are not decreasing as fast as they used to".

Paragraphs 2.1.5 and 2.1.7 both deal with inputs from the catchment – shouldn´t they be treated together (observations, past, likely future? And include atmospheric inputs as well?

In the interest of clarity we would prefer to keep the structure as it is now. Each of these three sections reports different issues: (i) climate influence on loads, (ii) atmospheric deposition and (iii) time-series of nutrient inputs from catchment.

345 ff: The entire chapter (2.2) on the coastal filter is very long and repetitive/redundant

with other parts. It does not really serve the purpose of updating information published since the last assessment.

The decision to separate the chapter on transformations of C,N,P in the coastal zone was made by the authors after a thorough review of the available literature. We noticed a number of studies published recently which underline the importance of the coastal regions acting as a biogeochemical filter between land and open Baltic Sea. In total 18 out of 25 papers cited in this chapter have been published since BACC II was concluded.
The length of this chapter reflects this added information from literature. However, we will carefully examine the revised manuscript for any potential redundancy and remove paragraphs and sentences that are redundant.

It would be helpful to define the extent and properties of the coastal filter, because in a sense the entire Baltic Sea is an estuary. Much of the data here relate to the northern Baltic Sea, whereas data for the southern coastal areas is not even mentioned (e.g., those raised in national and European projects).

This is a valid point and it would be good to have a formal operational definition of the coastal zone in the Baltic Sea. However, it is beyond the scope of our paper to propose such general definition, but we have provided the definition employed in Asmala et al. (2017), when referring to specific estimates of the efficiency of the coastal filter.

356: This is doubtful: much more <N> is removed in southern coastal areas based on delta15N data! Phosphorus is effectively shuttled to depositional basins from these areas. See, for example, papers by Radtke et al. (modeling) and Voss et al. (N-isotopes).

We agree that N is effectively removed in southern coastal areas, particularly the lagoons that are located in this part of the Baltic Sea and receive high nutrient inputs. This is also evident from Figure 3 and stated in the manuscript. Thus, we are unsure what the reviewer mean when stating that our statement is doubtful, since we seem to agree!

377: This may the case in northern sub-systems with high humic matter input - it is not the case in southern river discharge areas.

River water rich in humic-like substances is not specific for the northern part of the Baltic Sea catchment only. High concentrations have been reported also for the continental rivers draining into the Baltic Sea, for instance for the Vistula River (Pempkowiak & Kupryszewski, 1980)
Pempkowiak J., Kupryszewski G., 1980, The input of organic matter to the Baltic from the Vistula River. Oceanologia 12, 79-98.

386: This view appears to be very much skewed to northern Baltic.

We do not see this sentence is skewed to the northern Baltic Sea. This impression may arise as the coastal zone in the northern Baltic is much more diverse than in the south. Still, we see this sentence shows well the diversity of the entire Baltic coasts.

436: The river plumes and open coasts in the southern Baltic experience significant changes in nutrient ratios from a significant N-surplus to N-deficit.

We will add this information to the revised manuscript.

451: This entire paragraph is very speculative (may happen, can happen etc) - are there any constraints on coastal functioning in the warmer/more humid future from coupled models?

Coastal regions are very diverse ecosystems, often with high spatial and temporal variability of environmental conditions. This is due to the direct influence of land (including nutrient and organic matter input) and strong benthic-pelagic coupling. All these make those regions difficult to model and thus also to predict future changes.
Depending on the certainty of future changes, we chose appropriate wording. For cases, where the direction and/or scale of changes is difficult to predict, we intentionally used wording suggesting uncertainty.

489: This paragraph draws on older data - are there no new data that are relevant for the reporting period since BACC II? As it is, this recaps already published information with limited novelty.

We disagree with this opinion. This paragraph presents the actual state of knowledge in the field. It brings together both older data (published before BACC II) and the more recent one. This can be exemplified for instance in Fig. 4 and/or Table 1 and references therein.

529: What causes the different transport patterns of N and P? Sounds like diffusion, which is unlikely?

We will clarify that in the revised manuscript.
Nitrogen and phosphorus are exchanged intensely between different basins of the Baltic Sea. The net results of inter-basin imports and exports are determined by the water circulation pattern and the spatial distribution of N and P concentrations, which in turn are formed by the regional balances between nutrient inputs and nutrient sinks. Both net transports of N and P are directed westward from the Baltic Proper towards the Danish Straits, while phosphorus is also transported northwards because its concentrations are successively declining from the Baltic Proper to the Bothnian Sea and the Bothnian Bay (e.g. Savchuk, 2018 and references therein). In fact, most budgets estimate that the highest phosphorus removal takes place in the Bothnian Sea. Nitrogen, in contrast, is transported southward from the Bothnian Bay and the Bothnian Sea into the Baltic Proper, where most of the nitrogen removal takes place (see Table 1 and references therein). The net nutrient exchange of the Baltic Proper with the Gulf of Finland and the Gulf of Riga is directed towards or away from the Baltic Proper, depending on budget calculation method and period covered.

545 ff: These are general statements/observations – what is their purpose here?

This paragraph describes the actual state of measurements of primary production in the Baltic Sea. We believe it contains valuable information and would like to keep it in our manuscript.

The data in Table 2 are really old – how can they be relevant to an assessment paper for the present situation?

We do not generate any new knowledge in our study but only assess the knowledge, which has been published so far. Table 2 summarizes all available results on primary production in different regions of the Baltic Sea.
To make sure, we have reviewed again the literature and found the report by Zdun et al (2021), which was published already after this manuscript was submitted to the journal. We will include it in Table 2 in the revised manuscript.

Zdun, A., Stoń-Egiert, J., Ficek, D., Ostrowska, M., 2021. Seasonal and Spatial Changes of Primary Production in the Baltic Sea (Europe) Based on in situ Measurements in the Period of 1993–2018. Front. Mar. Sci. 7, 604532. https://doi.org/10.3389/fmars.2020.604532

591: One might mention somewhere that N2-fixation has been a consistent feature in the

Baltic over the last 8000 years or so (papers by Bianchi, Struck) and that the so-called vicious cycle has been a natural phenomenon since the Baltic Sea developed into a brackish system.

We will add this information into the revised manuscript, and also point to the anthropogenic reinforcement of some of the inherent phenomena

612: What can possibly cause that lag of 20 years?

This lag of 20 years describes the time between increasing nutrient loads and shaping favourable conditions for cyanobacteria blooms.
First, the increase in nutrient loads enhanced the production of N-limited phytoplankton. Higher production (eutrophication) contributed to the increase in the size of hypoxic and anoxic areas. Consequently, this intensified both denitrification (N loss) and recycling of P (reduced P burial efficiency) and led to the stoichiometric surplus of P making favourable conditions for P-limited $N_2$-fixing cyanobacteria.

624: Decline in benthic production due to an increasingly anoxic Baltic Proper?

The decrease in benthic production is expected to occur as a result of higher respiration in the water column and thus less food available for benthic organisms.
We will clarify that in the manuscript.

626: might decrease – anything might happen.....

As already stated above, in cases when the future predictions are uncertain we intentionally use wording showing this uncertainty. The anticipated increase in terrestrial DOC input to the Gulf of Bothnia will decrease the water transparency, which will negatively influence primary production. But primary production depends on many more factors, which in total make the scale of future change less certain.

638: How is that if P surplus is a main driver?

The word "counteract" was misused here. We will rephrase that in the manuscript.
This paragraph discusses sinks and sources of the bioavailable N in the future Baltic Sea. Simulations suggest that both denitrification and N2 fixation will increase in the Baltic. Our intention was to say that the future nitrogen loss through denitrification can largely balance nitrogen fixation and external inputs.

655: Lefébure (?) et al., 2013

Yes, spelling of the name and citation details are correct.

658 ff: There is no mention of change - only general principles are described

We disagree with this opinion. The entire chapter on organic matter remineralization and oxygen availability presents the current state of knowledge in the field. This includes the discussion of most recent changes, which occurred during and after the period of intense inflow activity in years 2014-2017.

676: deeper....weaker than what?

The average salinity, depth of the halocline and strength of the stratification oscillate in time. When we say lower, deeper or weaker we refer to the periods when those properties were found in the lower ranges

701: old reference, what is new?

We present in our study the current state of knowledge based on the available and published literature. We do not see the paper by Carstensen et al. (2014) "old", especially as it was not included in the previous assessment (BACC II, 2015).

705-712: These are contradictory statements?

No, they are not contradictory. First paragraph presents the results by Schneider and Otto (2019), who found the organic carbon remineralization rate relatively constant (irrespective of the redox conditions) during the period 2004-2014. While the second paragraph refers to the study by Meier et al. (2018), who found that oxygen consumption rate in the deep waters increased since 1970s.

710: How is that possible, when increased nutrient loads are responsible (2 paragraphs up)? Does that mean that zooplankton respiration was increased in deep water after inflows, or that zooplankton necromass caused a respiration increase? I have a problem with the mass balance - how much zooplankton in terms of ratio to primary production and sinking flux is needed to make a difference?

This paragraph describes the increased oxygen consumption rates observed after MBIs in the deep water layers since 1970s. The explanation derived from the model study suggests that this may be due to the increased POM concentration in the inflowing water. This includes both detritus and zooplankton. Thus, the effect comes from both zooplankton respiration and remineralization of detrital organic matter.

717: What does that mean for the entire system? That GoF and northern areas function like a lake?

The observation mentioned in this paragraph shows the specificity of the Gulf of Finland. This basin has a weaker saline stratification than the Baltic Proper, which can be broken during winter. Thus, the authors of the cited publication (Stoicescu et al. 2019) suggest that "oxygen debt" indicator should be based on data from the stratified season only. This finding based on high-frequency data obtained from the fixed automated station shows the importance of this type of sampling. However, due to the specificity of the Gulf of Finland hydrological setting, it does not have to be directly applicable to other regions.

736: What does stagnant mean – no movement? Above you mention that small but frequent inflows supply more oxygen than MBIs?

We will clarify this sentence. Our intention was to say that in the deep water layers in the central basins oxygen conditions are less dynamic and linked to Major Baltic Inflows.

Small inflows can provide oxygen for the Arkona Basin and partially also for the Bornholm Basin. But the amount of oxygen they transport is usually too low to re-oxygenate deep regions in the Gotland Basin.

747: stagnant?

We will clarify that. We wanted to say that after the latest period of intense inflow activity during 2014 to 2017 and several ventilation events of the central deep water, the environmental status switched back to conditions typical for the stagnation periods

776-780: Are these two statements not contradictory, because the lignin data are raised on sedimentary POC?

We see no contradiction in these statements.

In the first part, we described the mechanism of organic matter transport in which resuspension and resedimentation play an important role. This causes that sediments in the central deep basins are rich in organic carbon (concentrations of about 12-16% of the dry mass of sediment).

Further on, we refer to the study by Miltner and Emeis (2001), who found based on lignin biomarker analysis that 10-30% of the sedimentary organic carbon is terrigenous.

811: Do sulfate concentration indeed persist below the disappearance of methane?

Yes. Sulfate-methane transition zone (SMTZ) is usually several centimetres thick. For details please see for instance the study by Jilbert et al. (2018).

Jilbert, T., Asmala, E., Schröder, C., Tiihonen, R., Myllykangas, J.-P., Virtasalo, J. J., Kotilainen, A., Peltola, P., Ekholm, P., and Hietanen, S.: Impacts of flocculation on the distribution and diagenesis of iron in boreal estuarine sediments, Biogeosciences, 15, 1243-1271, https://doi.org/10.5194/bg-15-1243-2018, 2018.

864: Why should the rates in permeable sediments of the shallow Baltic Sea be different from rates in permeable sediments of the North Sea (e.g., Hüttl-papers; Neumann et al., 2021)? There high rates due to advective transports into ripples are considerably higher than those for diffusive transports in impermeable sediments. Can you suggest any reason?

We believe that a comparison of rates for permeable sediments in the Baltic to those in other systems lies outside the scope of this manuscript. If indeed rates in the Baltic Sea are higher than in the North Sea, this could be explained by the more eutrophic nature of the Baltic Sea and associated higher rates of organic matter supply to the sediment.

877: What does that mean: Are there hidden depocenters in shallow parts of the Baltic? Hard to believe that they have gone unnoticed or are large enough to make a difference....

So far, the studies on organic carbon burial focused mostly on the deep depositional areas in the Baltic Sea. Here, by bringing together information about sediment mass accumulation rates and organic carbon concentration in shallow regions we suggest studies on organic carbon burial in sediments should be extended also to these regions, which have been omitted in this respect so far.

939: The following paragraph is a general introduction to the carbonate system - is this needed?

We believe that this introductory paragraph is essential to understanding the specificities of the marine $CO_2$ system in the Baltic Sea, which are described later in this chapter. Recent novel findings such as on increasing alkalinity and deviations of the Baltic Sea acid-base system cannot be understood without this introduction. (see also general comment on scope of the paper).

976: announcement - put in outlook!

This sentence is not meant as an announcement, but the fact that, as described in the 3 papers cited, the technological hurdles have been overcome. We will replace "will allow" by "now allows" to avoid the misunderstanding

987-988 lower...higher: compared to what?

Compared to salinity and alkalinity observed in the central Baltic Sea.
We believe this sentence is correctly formulated.

Fig. 8 caption: Give full names in legend

We will add the full names of the basins in the caption of Fig. 8.

999: Is this relevant information in the context of this review?

This paragraph describes the recently detected peculiarities of the marine $CO_2$ system in the Baltic Sea. In our opinion, all the information mentioned there is an important contribution for studying the issues like for instance ocean acidification in the Baltic.

1032: Unit?

This is a typo. It should be µmol/kg. We will correct that.

1033: Is this last sentence important?

Yes. In this sentence, we would like to emphasize that the recent findings regarding the structure and variability of the marine $CO_2$ system in the Baltic Sea with all its peculiarities may contribute to a better understanding of the coastal processes in general (also outside the Baltic Sea).

1051: Due to enhanced CO2 assimilation?

Yes, assimilation and respiration. We will clarify that in the revised manuscript.

1054: 1990´s

We will correct that.

1062: atmospheric

We will extend atm. to atmospheric

1064: ...various pH-reducing drivers work in the same direction, with natural weathering potentially counteracting (what? pH decrease?)

Yes, pH decrease. We will clarify that in the revised manuscript.

1064: large

We will correct that.

Caption figure 9: It would be interesting to state the water depths of measurements.

These are surface data (0-10m). We will add that information to the caption in Fig. 9.

1084: non-Redfield

We will correct that.

1087-1088: on instead of at

We will correct that.

Paragraph 2.7.1.: unclear how this related to biogeochemistry? Most references are older than period under investigation.

Microorganisms were previously not included in the Baltic assessment reports such as BACC II. Due to the advancement in technology, microbial data on composition and functionality can provide substantial new insights and amendments to the biogeochemical studies in the Baltic Sea, which mostly focused on quantifying the transport and transformations of the macro-elements. In this section (2.7) we provide a state-of-the art compressed review on microorganisms involved in major biogeochemical processes in the Baltic Sea and their regulating factors. We think that for a better understanding of the recent developments in this field, it is important briefly referring to the most relevant studies that introduced the new approaches (e.g., high-throughput sequencing and omics) to the Baltic Sea and which are mainly from the years 2011-2015. Then main part of section 2.7, however, focuses on the role of microbial key players in biogeochemical cycles in the Baltic Sea. In the revised version, we have also added two more recent references (Rasigraf et al., 2017, 2019).

Again, we would like to emphasize that the goal of our study was not only to summarize the recent results (published after BACC II was concluded), but also to comprehensively assess the currently available, published knowledge on the microbially mediated biogeochemical functioning of the Baltic Sea, and the microorganisms involved, to present the current state of knowledge in the field.

1012: What role do they play in matter cycling?

The role of microorganisms in matter cycling is described in the following subchapters: 2.7.2 – 2.7.4.

1165 ff: references are quite old

As stated above, we wanted to highlight the studies that provided for the first time the new picture of Baltic Sea microbiomes by using the techniques that became available by then. Besides, these studies have not been replaced yet by more recent and more comprehensive studies. This paragraph presents the actual state of knowledge, which is supported with the references used. Therefore we do not see the reason to exclude some references, and thus also the knowledge, just because they were not published very recently. In total, there are 11 papers cited in this paragraph. Out of that, 9 were published in the last 10 years.

1199: How does the recent review paper differ from this one?

As it is said in our manuscript, the cited publication (Jørgensen et al., 2020) is an excellent review on specific aspects of the sub-seafloor biogeochemical processes and microbial life. It does, however, focus mainly on deeper sediments (e.g., based on an IODP expedition), and as such it is narrower in terms of scope than our study which encompasses the whole Baltic Sea, with water column and (surface) sediments.

Section 2.8 (pollutants): This entire section is vague and holds no pertinent information specific to the Baltic Sea. Metals and "old" organic contaminants are at the focus - wouldn´t it be appropriate to include data on novel pollutants (I believe such data are available)? Section 2.8.1: This again is in large parts quite general and not specific for the Baltic Sea, neither is the focus on progress in recent years....

Please see our detailed response related to that chapter above.

Part 3: There has been a lot of planning for future research on national and international levels in the past years - how are the points raised here integrated and interlinked?

Indeed, there are some initiatives being taken regarding biogeochemical research in the Baltic Sea. However, our ambition was not to review research plans of the individual consortia or scientific teams. We have focused in our manuscript only on reviewing the

published and available knowledge. Based on that we listed knowledge gaps and future research needs. However, we will re-structure Chapter 3 to represent current knowledge gaps in a more overarching way and with some degree of prioritization, as has been requested by Reviewer 1. It is beyond the scope of this paper to foreseen in how far some of the recent initiatives and projects will success in closing some of these knowledge gaps.

1349: This has been the case for the last 20 years - how can this shortcoming be remedied above all the data mining and modeling that has already gone into this?

This is still the case. Indeed, there were some attempts to model historical nutrient loads to the Baltic Sea before the 1970s, but the obtained results are uncertain. This still limits defining a reliable baseline for modelling the ecological status of the Baltic Sea. The sentence will be somewhat reformulated due to comments from the other reviewer.

1355: This as well has been a theme of intense research over decades already, hasn´t it?

This is true. However, the recent studies (for instance Fransner et al., 2018) in the Baltic Sea revealed that common Redfield-based stoichiometry may not explain nutrient and $pCO_2$ dynamics during spring and summer blooms in some parts of the Baltic Sea.

Fransner, F., Gustafsson, E., Tedesco, L., Vichi, V., Hordoir, R., Roquet, F., Spilling, K., Kuznetsov, I., Eilola, K., Magnus-Mörth, M., Humborg, C., and Nycander, J.: Non-Redfieldian Dynamics Explain Seasonal $pCO2$ Drawdown in the Gulf of Bothnia, Journal of Geophysical Research: Oceans, 123(1), 166-188, 2018.

1368: There have been quite a lot of recent studies on DOM, if I am not mistaken?

Yes, we agree that recent years brought many important findings about the role of DOM in the biogeochemical functioning of the Baltic Sea. Still, however, important knowledge gaps remain in the present understanding. This refers especially to parametrization of the terrestrial DOM transformations (including processes occurring in the land-sea aquatic continuum.

1399: Accumulation or turnover? It seems unlikely that there will be significant depocenters, but rather intense turnover is very likely in sandy southern shallow-water areas and seems to be indicated by both observations and models.

Both. As already mentioned above, in some of the shallow regions relatively high sediment mass accumulation rates have been identified. This, in connection with carbon concentrations observed there, makes those regions important to add them to the burial estimations.

1419 ff: This is certainly true, and not only in the Baltic Sea

Yes, we agree with this opinion. We see the topic of interactions between biogeochemical processes and chemical contaminants especially important in the Baltic Sea, which due to its location and hydrological setting is under high anthropogenic pressure. Our ambition was to extend the common scope of marine biogeochemical studies as focusing only on quantifying the transport and transformations of the macroelements in the marine environment and to bring to the discussion also this important for the Baltic Sea aspect.

1435: The Baltic Sea has "suffered" from anoxia over the last 8000 years. An eminent researcher once remarked: If you don´t want the Baltic to be anoxic, well, build a dam!

We fully agree with this opinion. We will correct the wording.

1057: Who is G.H.? Not in author list?

This is a typo. It should be G.R. We will correct that.